# Lie Group Decompositions for Equivariant Neural Networks

**Mircea Mironenco**
AI4Science Lab, AMLab
Informatics Institute
University of Amsterdam
mircea.mironenco@gmail.com

**Patrick Forré**
AI4Science Lab, AMLab
Informatics Institute
University of Amsterdam
p.d.forre@uva.nl

## Abstract

Invariance and equivariance to geometrical transformations have proven to be very useful inductive biases when training (convolutional) neural network models, especially in the low-data regime. Much work has focused on the case where the symmetry group employed is compact or abelian, or both. Recent work has explored enlarging the class of transformations used to the case of Lie groups, principally through the use of their Lie algebra, as well as the group exponential and logarithm maps. The applicability of such methods is limited by the fact that depending on the group of interest $G$, the exponential map may not be surjective. Further limitations are encountered when $G$ is neither compact nor abelian. Using the structure and geometry of Lie groups and their homogeneous spaces, we present a framework by which it is possible to work with such groups primarily focusing on the groups $G = \mathrm{GL}^+(n, \mathbb{R})$ and $G = \mathrm{SL}(n, \mathbb{R})$, as well as their representation as affine transformations $\mathbb{R}^n \rtimes G$. Invariant integration as well as a global parametrization is realized by a decomposition into subgroups and submanifolds which can be handled individually. Under this framework, we show how convolution kernels can be parametrized to build models equivariant with respect to affine transformations[1]. We evaluate the robustness and out-of-distribution generalisation capability of our model on the benchmark affine-invariant classification task, outperforming previous proposals.

## 1 Introduction

Symmetry constraints in the form of invariance or equivariance to geometric transformations have shown to be widely applicable inductive biases in the context of deep learning (Bronstein et al., 2021). Group-theoretic methods for imposing such constraints have led to numerous breakthroughs across a variety of data modalities. CNNs (LeCun et al., 1995) which make use of translation equivariance while operating on image data have been generalized in several directions. Group-equivariant convolutional neural networks (GCNNs) represent one such generalization Cohen & Welling (2016). GCNNs make use of group convolution operators to construct layers that produce representations which transform in a predictable manner whenever the input signal is transformed by an a-priori chosen symmetry group $G$. These models have been shown to exhibit increased generalization capabilities, while being less sensitive to $G$-perturbations of the input data. For these reasons, equivariant architectures have been proposed for signals in a variety of domains such as graphs (Han et al., 2022), sets (Zaheer et al., 2017) or point clouds data (Thomas et al., 2018). Constructing equivariant networks entails first choosing a group $G$, a representation for the signal space in which our data lives and a description of the way this space transforms when the group *acts* on it. Choosing a particular group $G$ entails making a modelling assumption about the underlying (geometrical) structure of the data that should be preserved. Early work has focused on the case where $G$ is finite, with later work largely concentrated on the Euclidean group $\mathrm{E}(n)$, and its subgroups $\mathrm{SE}(n, \mathbb{R})$ or $\mathrm{SO}(n)$. Working with continuous groups is much more challenging, and the vast majority of equivariant models focus on the case where the group $G$ has a set of desirable topological and structural properties, namely $G$ is either compact or abelian, or both.

---

[1]Our code is publicly available at `https://github.com/mirceamironenco/rgenn`.

Recent work (Bekkers, 2019; Finzi et al., 2020) explores the possibility of building equivariant networks for Lie groups - continuous groups with a smooth structure. This research direction is promising since it allows for the modelling of symmetry groups beyond Euclidean geometry. Affine and projective geometry, respectively affine and homography transformations are ubiquitous within computer vision, robotics and computer graphics (Zacur et al., 2014). Accounting for a larger degree of geometric variation has the promise of making (vision) architectures more robust to real-world data shifts. When working with non-compact and non-abelian Lie groups, for which the group exponential is not surjective, standard harmonic analysis tools cannot be employed directly. Our contribution is a framework making it possible to work with such groups.

**Contributions** We present a procedure by which invariant integration with respect to the Haar measure can be done in a principled manner, allowing for an efficient numerical integration scheme to be realized. We then construct *global* parametrization maps which allow us to map elements back and forth between the Lie algebra and the group, addressing the non-surjectivity of the group exponential. We apply our framework to the groups $G = \mathrm{GL}^+(n, \mathbb{R})$ and $G = \mathrm{SL}(n, \mathbb{R})$, and more broadly the family of affine matrix Lie groups $\mathrm{Aff}(G) \coloneqq \mathbb{R}^n \rtimes G, G \leq \mathrm{GL}(n, \mathbb{R})$. The methodology and tools are generally applicable to any Lie group with finitely many connected components, and we explain how our approach can be seen as a generalization of previous proposals for constructing equviariant layers when working with the regular representation of a topological group.

## 2 RELATED WORK

Recent proposals for Lie group equivariance (Bekkers, 2019; Finzi et al., 2020) focus on the infinite-dimensional regular representation of a group and rely on the group exponential map to allow convolution kernels to be defined analytically in the Lie algebra of the group. Working with the regular representation entails dealing with an intractable convolution integral over the group, and a (Monte Carlo) numerical integration procedure approximating the integral needs to be employed, which requires sampling group elements with respect to the *Haar* measure of the group. Unfortunately, the applicability of these methods is limited to Lie groups for which the group exponential map is surjective, which is not the case for the affine group $\mathrm{Aff}(\mathrm{GL}(n, \mathbb{R}))$. These methods also rely on the fact that for compact and abelian groups sampling with respect to the Haar measure of the group is straightforward, which is not the case for the affine groups of interest. MacDonald et al. (2022) propose a framework which can be applied to arbitrary Lie groups, aiming to address such limitations while still relying on the group exponential. This proposal is more closely reviewed in Sec. A.3, together with other related equivariant models.

## 3 BACKGROUND

**Continuous group equivariance** A Lie group $G$ is a group as well as a smooth manifold, such that $\forall g, h \in G$ the group operation $(g, h) \mapsto gh$ and the inversion map $g \mapsto g^{-1}$ are smooth. $\mathrm{GL}(n, \mathbb{R})$ denotes the Lie group consisting of all invertible $n \times n$ matrices. A *linear* or *matrix* Lie group refers to a Lie subgroup of $\mathrm{GL}(n, \mathbb{R})$. $\mathrm{GL}(n, \mathbb{R})$, the translation group $(\mathbb{R}^n, +)$ and the family of affine groups $\mathrm{Aff}(G)$, $G \leq \mathrm{GL}(n, \mathbb{R})$ are our primary interest, with $G$ usually being one of $\mathrm{GL}^+(n, \mathbb{R})$, $\mathrm{SL}(n, \mathbb{R}) \leq \mathrm{GL}^+(n, \mathbb{R})$ or $\mathrm{SO}(n)$. Equivariance with respect to the action of a locally compact group $G$ can be realized by constructing layers using the cross-correlation/convolution operators. We recall that in the continuous setting we model our signals as functions $f : X \to \mathbb{R}^K$ defined on some underlying domain $X$. For example, images and feature maps can be defined as $K$-channel functions $f \in L^2_\mu(\mathbb{R}^2, \mathbb{R}^K)$ which are square-integrable (with respect to the measure $\mu$), and which have bounded support in practice, e.g. $f : [-1, 1]^2 \subseteq \mathbb{R}^2 \to \mathbb{R}^K$. $\mathcal{L}_g$ denotes the left-regular representation of $G$, encoding the action of $G$ on function spaces. For any continuous $f \in C(X)$:

$$[\mathcal{L}_g f](x) \coloneqq f(g^{-1}x), \quad \forall g \in G, \ x \in X \tag{1}$$

Every locally compact group $G$ has a left (right) *invariant* Radon measure $\mu_G$ called the left (right) *Haar measure* of $G$. The Haar measure allows for $G$-invariant integration to be realized, and for the group convolution to be defined. To state the invariance property of $\mu_G$, define the functional $\lambda_{\mu_G}$:

$$\lambda_{\mu_G} : L^1(G) \to \mathbb{R}, \quad \lambda_{\mu_G}(f) = \int_G f(g) \mathrm{d}\mu_G(g), \ \forall f \in L^1(G) \tag{2}$$

Then, a left Haar measure respects $\lambda_{\mu_G}(\mathcal{L}_g f) = \lambda_{\mu_G}(f)$ for any $g \in G$ and $f \in L^1(G)$. Additional details on convolution operators are provided in Sec. A.1, with Lie groups reviewed in Sec. B.1.

**Convolution operators** For a group $G$ acting transitively on locally compact spaces $X$ and $Y$ we then seek to construct an operator $\mathcal{K} : L^2(X) \to L^2(Y)$ satisfying the *equivariance constraint* $\mathcal{L}_g \circ \mathcal{K} = \mathcal{K} \circ \mathcal{L}_g$. We formalize two scenarios, when $X$ is a homogeneous space of $G$ (not necessarily a group) and $Y = G$, and the case where $X = Y = G$. Focusing on the second case, if $\mathrm{d}\mu_X = \mathrm{d}\mu_G$ is the Haar measure on $G$, the integral operator $\mathcal{K}$ can be defined was the standard convolution/cross-correlation. Let $k : Y \times X \to \mathbb{R}$ be a kernel that is invariant to the left action of $G$ in both arguments, such that $k(gx, gy) = k(x, y)$ for any $(x, y) \in Y \times X$ and $g \in G$. Let $\mu_X$ be a $G$-invariant Radon measure on $X$, and define $\mathcal{K} := C_k : L^p(X) \to L^p(G)$ ($p \in \{1, 2\}$) such that $\forall f \in L^p(X)$:

$$C_k : f \mapsto C_k f(y) = \int_X f(x) k(x, y) \mathrm{d}\mu_X(x), \quad \forall y \in Y \tag{3}$$

$C_k$ is $G$-equivariant: $\mathcal{L}_g \circ C_k = C_k \circ \mathcal{L}_g, \ \forall g \in G$ (A.2). Since $X = Y = G$ are homogeneous spaces of $G$ we can easily define a bi-invariant kernel by projection $k(x, y) = \tilde{k}(g_y^{-1} x)$ ($\tilde{k} : G \to \mathbb{R}$) for any $(x, y) \in Y \times X$, where $y = g_y y_0$ for some fixed $y_0$. The kernel is bi-invariant:

$$k(hx, hy) = \tilde{k}((hg_y)^{-1} hx) = \tilde{k}(g_y^{-1} h^{-1} hx) = \tilde{k}(g_y^{-1} x) = k(x, y), \quad \forall h \in G \tag{4}$$

For the case $Y = G$ and $g_y = y$ ($y_0 = e$, the identity of $G$) this corresponds to a cross-correlation. For a convolution operator, we would analogously define $k(x, y) = \tilde{k}(g_x^{-1} y)$ where $x = g_x x_0$ for $x_0 \in X$. In this case the *essential* component needed for equivariance of the operator $C_k$ is the $G$-invariant measure $\mathrm{d}\mu_X$, which is the Haar measure when $X = Y = G$. When $X$ is a homogeneous space of $G$, but not necessarily $G$ itself, we have to work with an operator which takes in a signal in $L^p(X)$ and produces a signal $L^p(G)$ on the group. This encompasses the case of the *lifting* layers, which are commonly employed when working with the regular representation of a group (Cohen & Welling, 2016; Kondor & Trivedi, 2018). The kernel $k(\cdot)$ in this case can be derived through an equivariance constraint as in Bekkers (2019); Cohen et al. (2019). It can also be shown (A.2) that an equivariant lifting cross-correlation can be defined as an operator $C_k^{\uparrow}$ such that for any $f \in L^p(X)$:

$$C_k^{\uparrow} : f \mapsto C_k^{\uparrow} f, \quad C_k^{\uparrow} f : g \mapsto \int_X f(x) k(g^{-1} x) \delta(g^{-1}) \mathrm{d}\mu_X(x), \ \forall g \in G \tag{5}$$

where $\delta : G \to \mathbb{R}_{>0}^{\times}$ records the change of variables by the action of $G$ (see A.2). Group cross-correlation $C_k^{\star} := C_k$ and convolution $C_k^{*}$ operators will be defined for any $f \in L^p(G)$:

$$C_k : f \mapsto C_k f, \quad C_k f : g \mapsto \int_G f(\tilde{g}) k(g^{-1} \tilde{g}) \mathrm{d}\mu_G(\tilde{g}), \ \forall g \in G \tag{6}$$

$$C_k^{*} : f \mapsto C_k^{*} f, \quad C_k^{*} f : g \mapsto \int_G f(\tilde{g}) k(\tilde{g}^{-1} g) \mathrm{d}\mu_G(\tilde{g}), \ \forall g \in G \tag{7}$$

**Lie algebra parametrization** The tangent space at the identity of a Lie group $G$ is denoted by $\mathfrak{g}$ and called the Lie algebra of $G$. A Lie algebra is a vector space equipped with a bilinear map $[\cdot, \cdot] : \mathfrak{g} \times \mathfrak{g} \to \mathfrak{g}$ called the Lie bracket. To construct an equivariant layer using the Lie algebra of the group, one defines the kernels $k(\cdot)$ in (6) or (7) as functions which take in Lie algebra elements. This requires a map $\xi : \mathfrak{g} \to G$ which is (at least locally) a diffeomorphism, with an inverse that can be easily calculated, preferably in closed-form. This allows us to rewrite the kernel $k : G \to \mathbb{R}$ as:

$$k(g^{-1} \tilde{g}) = k(\xi(\xi^{-1}(g^{-1} \tilde{g}))) = \tilde{k}_\theta(\xi^{-1}(g^{-1} \tilde{g})) \tag{8}$$

$\tilde{k}_\theta(\cdot)$ is effectively an approximation of $k(\cdot)$ of the form $\tilde{k}_\theta \cong k \circ \xi : \mathfrak{g} \to \mathbb{R}$ with learnable parameters $\theta$. Using the inverse map $\xi^{-1}(g^{-1} \tilde{g})$, $\tilde{k}_\theta$ maps the Lie algebra coordinates of the 'offset' group element $g^{-1} \tilde{g}$ (for cross-correlations) to real values corresponding to the evaluation $k(g^{-1} \tilde{g})$. Our kernels are now maps $\tilde{k}_\theta \circ \xi^{-1} : G \to \mathbb{R}$, requiring the implementation of $\xi^{-1}(\cdot)$ and a particular choice for the Lie algebra kernel $\tilde{k}_\theta$. This description encompasses recent proposals for Lie group equivariant layers. In Bekkers (2019) the kernels are implemented by modelling $\tilde{k}_\theta$ via B-splines, while Finzi et al. (2020) choose to parametrize $\tilde{k}_\theta$ as small MLPs. Once $\tilde{k}_\theta$ and $\xi$ are chosen, we can approximate e.g. the cross-correlation using Monte Carlo integration:

$$\int_G f(\tilde{g}) \tilde{k}_\theta(\xi^{-1}(g^{-1} \tilde{g})) \mathrm{d}\mu_G(\tilde{g}) \approx \frac{\mu_G(G)}{N} \sum_{i=1}^{N} f(\tilde{g}_i) \tilde{k}_\theta(\xi^{-1}(g^{-1} \tilde{g}_i)), \ \tilde{g}_i \sim \mu_G \tag{9}$$

where $\mu_G(G)$ denotes the volume of the integration space $G$ and $\tilde{g}_i \sim \mu_G$ indicates that $\tilde{g}_i$ is sampled (uniformly) with respect to the Haar measure. This allows one to obtain equivariance (in expectation) with respect to $G$. For compact groups, $\mu_G$ can be normalized such that $\mu_G(G) = 1$. To summarize, we record the components of the framework which are needed for (9) to realize an equivariant operator. Namely, we require (1) a *parametrization* map $\xi^{-1} : G \to \mathfrak{g}$, as well as (2) the implementation of an efficient *sampling* scheme with respect to the Haar measure $\mu_G$ such that numerical integration is feasible in practice.

## 4 LIE GROUP DECOMPOSITIONS FOR CONTINUOUS EQUIVARIANCE

**Limitations of the group exponential** For every Lie group we can define the Lie group exponential map expm : $\mathfrak{g} \to G$, which is a diffeomorphism locally around $0 \in \mathfrak{g}$. Since we are interested in $\mathrm{GL}(n, \mathbb{R})$ and its subgroups, we can make things more concrete as follows. $\mathrm{M}_{nn}(\mathbb{R}) := \mathrm{M}_n(\mathbb{R})$ (the vector space of $n \times n$ real matrices) is the Lie algebra of $\mathrm{GL}(n, \mathbb{R})$ (Sec. B.1). The notation $\mathfrak{gl}(n, \mathbb{R}) = \mathrm{M}_n(\mathbb{R})$ is used for this identification. For $G = \mathrm{GL}(n, \mathbb{R})$ with $\mathfrak{g} = \mathfrak{gl}(n, \mathbb{R})$, the group exponential is the matrix exponential expm : $\mathfrak{gl}(n, \mathbb{R}) \to \mathrm{GL}(n, \mathbb{R})$, with the power series expression $X \mapsto e^X = \sum_{k=0}^{\infty} \frac{1}{k!} X^k$. The map $\xi$ in (8) is most commonly implemented as the group exponential $\xi := $ expm. Given a subgroup $G \leq \mathrm{GL}(n, \mathbb{R})$ for which expm is surjective, every element $g \in G$ can be expressed as $g = \mathrm{expm}(X) = e^X$ for $X \in \mathfrak{g}$, and fast routines for calculating $\mathrm{expm}(\cdot)$ are available. In this case, the inverse map $\xi^{-1}$ is given by the matrix logarithm, giving us:

$$\xi^{-1}(g^{-1}\tilde{g}) = \mathrm{logm}(g^{-1}\tilde{g}), \quad \mathrm{logm} : G \to \mathfrak{g} \tag{10}$$

In general, we need to consider if both $\xi$ and $\xi^{-1}$ need to be implemented, and whether these maps are available in closed form. Assuming there exist $X$ and $Y$ such that $e^X = g^{-1}$ and $e^Y = \tilde{g}$, (10) can be rewritten as $\mathrm{logm}(g^{-1}\tilde{g}) = \mathrm{logm}(e^X e^Y)$. A key optimization underlying this framework is enabled by employing the BCH formula (B.1), which tells us that for *abelian* Lie groups $\mathrm{logm}(e^X e^Y) = X + Y$. This simplifies calculations considerably and allows one to work primarily at the level of the Lie algebra, bypassing the need to calculate and sample the kernel inputs $g^{-1}\tilde{g}$ at the group level. Considering the affine Lie groups $\mathrm{Aff}(G)$, $G \leq \mathrm{GL}(n, \mathbb{R})$, this simplification can be used for example for the abelian groups $G = \mathrm{SO}(2)$ and $G = \mathbb{R}^\times(2) \times \mathrm{SO}(2)$, consisting of rotations and scaling. Bekkers (2019); Finzi et al. (2020) primarily work with these groups, and choose $\xi$ and $\xi^{-1}$ to be the matrix exponential and logarithm, respectively. If the group is non-abelian but the exponential remains surjective (such as with compact groups like $\mathrm{SO}(3)$), $\mathrm{expm}(\cdot)$ remains a generally valid choice for $\xi$ as long as $\xi^{-1}$ can be accurately calculated in closed-form. For the non-abelian, non-compact groups $\mathrm{SL}(n, \mathbb{R})$ or $\mathrm{GL}^+(n, \mathbb{R})$ the non-surjectivity of the exponential map limits the applicability of the matrix logarithm outside of a neighborhood around the identity (Prop. B.2). The class of equivariant networks that can be implemented with this framework is then firstly limited by the parametrization maps $\xi$ and $\xi^{-1}$, motivating the search for an alternative.

Another key limitation is that for (9) to realize an equivariant estimator when numerically approximating the convolution/cross-correlation integral, sampling needs to be realised with respect to the Haar measure of the group $G$. Techniques for sampling with respect to the Haar measure on the groups $\mathrm{SO}(n)$ or $\mathbb{R}^\times(n) \times \mathrm{SO}(n)$ are known, and generally reduce to working with uniform measures on Euclidean spaces or unit quaternions in the case of $\mathrm{SO}(3)$. We aim to address these limitations, allowing the previously described framework to be generalized to arbitrary Lie groups $G \leq \mathrm{GL}(n, \mathbb{R})$. We further seek a solution that places minimal limitations on the class of 'Lie algebra kernels' $k_\theta : \mathfrak{g} \to \mathbb{R}$ that can be used, and one should be able to employ any $k_\theta$ that uses the coordinates of tangent vectors in $\mathfrak{g}$ expressed in some basis. In the following we present a set of generally applicable tools while considering $\mathrm{SL}(n, \mathbb{R})$ and $\mathrm{GL}^+(n, \mathbb{R})$ as working examples, since these groups require more consideration and represent our primary application.

### 4.1 LIE GROUP DECOMPOSITION THEORY

We exploit the fact that the groups $\mathrm{GL}^+(n, \mathbb{R})$ and $\mathrm{SL}(n, \mathbb{R})$ have an underlying product structure that allows them to be decomposed into subgroups and submanifolds which are easier to work with individually. More precisely, $G \in \{\mathrm{GL}^+(n, \mathbb{R}), \mathrm{SL}(n, \mathbb{R})\}$ can be decomposed as a product $P \times H$, where $H \leq G$ is the maximal compact subgroup of $G$ and $P \subseteq G$ is a submanifold which is diffeomorphic to $\mathbb{R}^k$, for some $k \geq 0$, and we have a diffeomorphism $\varphi : P \times H \to G$.

Similar decompositions are available for a larger class of groups $G \leq \mathrm{GL}(n, \mathbb{R})$ (Abbaspour & Moskowitz, 2007, Ch. 6). It can be shown that if the map $\varphi$ is chosen correctly the Haar measure $\mu_G$ can be expressed as the pushforward measure $\varphi_*(\mu_P \otimes \mu_H)$, where $\mu_P$ is a $G$-invariant measure on $P$ and $\mu_H$ is the Haar measure on $H$. In some cases the group decomposition presents a corresponding Lie algebra decomposition, which we can leverage to build the parametrization map $\xi^{-1} : G \to \mathfrak{g}$.

**Factorizing the Haar measure** Let $G$ be a locally compact group of interest (e.g. $\mathrm{GL}^+(n, \mathbb{R})$), with (left) Haar measure $\mu_G$. Assume there exist a set of subspaces or subgroups $P \subseteq G$, $K \subseteq G$, such that $G = PK$, and a homeomorphism $\varphi : P \times K \to G$. Further assume that $\mu_P$ and $\mu_K$ are (left) $G$-invariant Radon measures on the corresponding spaces. We look to express (up to multiplicative coefficients) the Haar measure $\mu_G$ as the pushforward of the product measure $\mu_P \otimes \mu_K$ under the map $\varphi$. This allows for the following change of variables for any $f \in L^1(G)$:

$$\int_G f(g) \mathrm{d}\mu_G(g) = \int_{P \times K} f(\varphi(p,k)) \mathrm{d}(\mu_P \otimes \mu_K)(p,k) = \int_P \int_K f(\varphi(p,k)) \mathrm{d}\mu_K(k) \mathrm{d}\mu_P(p) \quad (11)$$

In the context of Monte Carlo simulation this will enable us to produce random samples distributed according the measure $\mu_G$ by sampling on the *independent* factor spaces $P$ and $K$ and constructing a sample on $P \times K$ and respectively on $G$ using the map $\varphi$. The space $P$ will either be another closed subgroup, or a measurable subset $P \subseteq G$ that is homeomorphic to the quotient space $G/K$. In particular, if $P$ is not a subgroup, we will focus on the case where $P$ is a homogeneous space of $G$ with stabilizer $K$ such that $P \cong G/K$. When the left and right Haar measure of a group coincide, the group is called *unimodular*. The groups $\mathrm{GL}^+(n, \mathbb{R})$, $\mathrm{SL}(n, \mathbb{R})$ are unimodular, however this is not true for all affine groups $\mathrm{Aff}(G)$. For groups which are volume-preserving, this is not as much of an issue in practice. However, $\mathrm{GL}^+(n, \mathbb{R})$ is not volume-preserving, and we also desire that our framework be general enough to deal with the non-unimodular case as well. If $G$ is not unimodular and $\mu_G$ is its left Haar measure, there exists a continuous group homomorphism $\Delta_G : G \to \mathbb{R}^\times_{>0}$, called the *modular function* of $G$, which records the degree to which $\mu_G$ fails to be right-invariant. We now have the tools necessary to record two possible integral decomposition methods.

**Theorem 4.1.** *(1) Let $G$ be a locally compact group, $H \leq G$ a closed subgroup, with left Haar measures $\mu_G$ and $\mu_H$ respectively. There is a $G$-invariant Radon measure $\mu_{G/H}$ on $G/H$ if and only if $\Delta_G|_H = \Delta_H$. The measure $\mu_{G/H}$ is unique up to a scalar factor and if suitably normalized:*

$$\int_G f(g) \mathrm{d}\mu_G(g) = \int_{G/H} \int_H f(gh) \mathrm{d}\mu_H(h) \mathrm{d}\mu_{G/H}(gH), \; \forall f \in L^1(G) \quad (12)$$

*(2) Let $P \leq G$, $K \leq G$ closed subgroups such that $G = PK$. Assume that $P \cap K$ is compact, and $Z_0$ denotes the stabilizer of the transitive left action of $P \times K$ on $G$ given by $(p,k) \cdot g = pgk^{-1}$, for any $(p,k) \in P \times K$ and $g \in G$. Let $G$, $P$ and $K$ be $\sigma$-compact (which holds for matrix Lie groups), $\mu_G$, $\mu_P$ and $\mu_K$ left Haar measures on $G$, $P$, and $K$ respectively and $\Delta_G|_K = \Lambda$ is the modular function of $G$ restricted to $K$. Then $\mu_G$ is given by $\mu_G = \pi_*(\mu_P \otimes \Lambda^{-1}\mu_K)$, where $\pi : P \times K \to (P \times K)/Z_0$ is the canonical projection. In integral form we have:*

$$\int_G f(g) \mathrm{d}\mu_G(g) = \int_P \int_K f(pk) \frac{\Delta_G(k)}{\Delta_K(k)} \mathrm{d}\mu_K(k) \mathrm{d}\mu_P(p), \quad \forall f \in L^1(G) \quad (13)$$

*Proof.* Folland (2016, Theorem 2.51) and Wijsman (1990, Proposition 7.6.1). $\square$

The existence and range of the convolution operators (for arbitrary Lie groups $G$) are described in Sec. A.2.1, with the non-unimodular case being covered by Prop. A.3. When going to the Lie group setting, we can already deal with semi-direct products of groups of the form $N \rtimes G$. The modular function on $N \rtimes G$ is $\Delta_{N \rtimes G}(n,g) = \Delta_N(n)\Delta_G(g)\delta(g)^{-1}$ (Kaniuth & Taylor, 2013). The term $\delta : G \to \mathbb{R}^\times_{>0}$ records the effect of the action of $G$ on $N$, and it coincides with the term $\delta(\cdot)$ used in the lifting layer definition (27). Concretely, take the affine groups $\mathrm{Aff}(G) = \mathbb{R}^n \rtimes G$, $G \leq \mathrm{GL}(n, \mathbb{R})$, defined under the semi-direct product structure:

$$\mathrm{Aff}(G) = \mathbb{R}^n \rtimes G = \{(x, A) \mid x \in \mathbb{R}^n, \ A \in G\} \quad (14)$$

$G$ acts on $\mathbb{R}^n$ by matrix multiplication and for $(x, A), (y, B) \in \mathrm{Aff}(G)$, the product and inverse are:

$$(x, A)(y, B) = (x + Ay, AB), \quad (x, A)^{-1} = (-A^{-1}x, A^{-1}) \quad (15)$$

Elements of $\mathbb{R}^n$ are concretely represented as column vectors. Viewing $(\mathbb{R}^n, +)$ as the additive group, we have $\delta : G \to \mathbb{R}^{\times}_{>0}$ given by $\delta(A) = |\det(A)|$ for any $A \in G$. Applying Thm. 4.1, gives:

$$\int_{\text{Aff}(G)} f(g) \, \mathrm{d}\mu_{\text{Aff}(G)}(g) = \int_G \int_{\mathbb{R}^n} f((x, A)) \frac{\mathrm{d}x \mathrm{d}\mu_G(A)}{|\det(A)|}, \; \forall f \in C_c(\text{Aff}(G)) \tag{16}$$

Expressing the cross-correlation $C_k f$ of (7) in this product space we have for $f \in L^2(\text{Aff}(G))$:

$$C_k f : (x, A) \mapsto \int_G \int_{\mathbb{R}^n} f(\tilde{x}, \tilde{A}) k((x, A)^{-1}(\tilde{x}, \tilde{A})) \delta(\tilde{A}^{-1}) \mathrm{d}\tilde{x} \mathrm{d}\mu_G(\tilde{A}) \tag{17}$$

For the affine groups $\text{Aff}(G) = \mathbb{R}^n \rtimes G$ a parametrization map $\xi_{\text{Aff}(G)} : \mathbb{R}^n \oplus \mathfrak{g} \to \text{Aff}(G)$ will simply be the identity on the first factor, since the Lie algebra of $\text{Aff}(\text{GL}(n, \mathbb{R}))$ decomposes as $\mathbb{R}^n \oplus \mathfrak{gl}(n, \mathbb{R})$ when represented as a Lie subalgebra of $\mathfrak{gl}(n + 1, \mathbb{R})$. We are then left with the parametrization and invariant integration of the $G$-factor of $\text{Aff}(G)$. We provide a solution for the cases $G = \text{SL}(n, \mathbb{R})$ and $G = \text{GL}^+(n, \mathbb{R})$, while remarking that a solution for $\text{GL}^+(n, \mathbb{R})$ can be immediately extended to $\text{GL}(n, \mathbb{R})$ (Sec. B.4). Our approach is based on a generalized Polar decomposition of matrices, which is applicable in the case of reductive ($\text{GL}^+(n, \mathbb{R})$) or semi-simple ($\text{SL}(n, \mathbb{R})$) Lie groups. An alternative decomposition is discussed in Sec. B.6.2.

**Manifold splitting via Cartan/Polar decomposition** Let $\text{Sym}(n, \mathbb{R})$ be the vector space of $n \times n$ real symmetric matrices and $\text{Pos}(n, \mathbb{R})$ the subset of $\text{Sym}(n, \mathbb{R})$ of symmetric positive definite (SPD) matrices. Denote by $\text{SPos}(n, \mathbb{R})$ the subset of $\text{Pos}(n, \mathbb{R})$ consisting of SPD matrices with unit determinant, and by $\text{Sym}_0(n, \mathbb{R})$ the subspace of $\text{Sym}(n, \mathbb{R})$ of traceless real symmetric matrices. Any matrix $A \in \text{GL}(n, \mathbb{R})$ can be uniquely decomposed via the left polar decomposition as $A = PR$ where $P \in \text{Pos}(n, \mathbb{R})$ and $R \in \text{O}(n)$ (B.4). The factors of this decomposition are uniquely determined and we have a bijection $\text{GL}(n, \mathbb{R}) \to \text{Pos}(n, \mathbb{R}) \times \text{O}(n)$ given by:

$$A \mapsto (\sqrt{AA^T}, \sqrt{AA^T}^{-1} A), \quad \forall A \in \text{GL}(n, \mathbb{R}) \tag{18}$$

For the reader unfamiliar with Lie group structure theory, the following results can simply be understood in terms of matrix factorizations commonly used in numerical linear algebra. The polar decomposition splits the manifold $\text{GL}^+(n, \mathbb{R})$ into the product $\text{Pos}(n, \mathbb{R}) \times \text{SO}(n)$, and $\text{SL}(n, \mathbb{R})$ into $\text{SPos}(n, \mathbb{R}) \times \text{SO}(n)$. We use the notation $G \to M \times H$ to cover both cases. This decomposition can be generalized, as the spaces $\text{Pos}(n, \mathbb{R}) = \text{GL}^+(n, \mathbb{R})/\text{SO}(n)$ and $\text{SPos}(n, \mathbb{R}) = \text{SL}(n, \mathbb{R})/\text{SO}(n)$ are actually *symmetric spaces*, and a *Cartan decomposition* is available in this case (B.4). The Cartan decomposition tells us how to decompose not only at the level of the Lie group, but also at the level of the Lie algebra. In fact, using this decomposition we can also obtain a factorization of the measure on these groups. Let $(G/H, M, \mathfrak{m})$ define our 'Lie group data', corresponding to $(\text{GL}^+(n, \mathbb{R})/\text{SO}(n), \text{Pos}(n, \mathbb{R}), \text{Sym}(n, \mathbb{R}))$ or $(\text{SL}(n, \mathbb{R})/\text{SO}(n), \text{SPos}(n, \mathbb{R}), \text{Sym}_0(n, \mathbb{R}))$.

**Theorem 4.2.** *Let $(G/H, M, \mathfrak{m})$ be as above, and denote by $\mathfrak{g}, \mathfrak{h}$ the Lie algebras of $G$ and $H$.*

1. *The matrix exponential and logarithm are diffeomorphisms between $\mathfrak{m}$ and $M$, respectively. For any $P \in M$ and $\alpha \in \mathbb{R}$, the power map $P \mapsto P^\alpha$ is smooth and can be expressed as:*
$$P^\alpha = \text{expm}(\alpha \text{logm}(P)), \quad \forall P \in \text{Pos}(n, \mathbb{R}) \tag{19}$$

2. *$G \cong M \times H$ and $G \cong \mathfrak{m} \times H$. We have group-level diffeomorphisms:*
$$\chi : M \times H \to G, \quad \chi(P, R) \mapsto PR \tag{20}$$
$$\Phi : \mathfrak{m} \times H \to G, \quad \Phi : (X, R) \mapsto \text{expm}(X)R = e^X R \tag{21}$$

3. *The above maps can be inverted in closed-form:*
$$\chi^{-1} : G \to M \times H, \; \chi^{-1} : A \mapsto (\sqrt{AA^T}, \sqrt{AA^T}^{-1} A) \tag{22}$$
$$\Phi^{-1} : G \to \mathfrak{m} \times H, \quad \Phi^{-1} : A \mapsto (\frac{1}{2} \text{logm}(AA^T), \text{expm}(-\frac{1}{2} \text{logm}(AA^T))A) \tag{23}$$

A proof and further references for Theorem 4.2 can be found in Sec. B.5. At the level of the Lie algebra, we have the decomposition $\mathfrak{gl}(n, \mathbb{R}) = \mathfrak{so}(n) \oplus \text{Sym}(n, \mathbb{R})$. The Lie algebra of $\text{SL}(n, \mathbb{R})$ is $\mathfrak{sl}(n, \mathbb{R}) = \{X \in \mathfrak{gl}(n, \mathbb{R}) \mid \text{tr}(X) = 0\}$. It decomposes similarly $\mathfrak{sl}(n, \mathbb{R}) = \mathfrak{so}(n) \oplus \text{Sym}_0(n, \mathbb{R})$. The Cartan decomposition of $\mathfrak{g}$ is therefore expressed as $\mathfrak{g} = \mathfrak{h} \oplus \mathfrak{m}$ where $\mathfrak{h} = \mathfrak{so}(n)$ with $\mathfrak{m} = \text{Sym}(n, \mathbb{R})$ if $G = \text{GL}^+(n, \mathbb{R})$ and $\mathfrak{m} = \text{Sym}_0(n, \mathbb{R})$ if $G = \text{SL}(n, \mathbb{R})$.

## 4.2 A PARAMETRIZATION BASED ON THE CARTAN DECOMPOSITION

Consider again the notation $(G/H, M, \mathfrak{m})$ as in Theorem 4.2 ($G = \mathrm{GL}^+(n, \mathbb{R})$ or $G = \mathrm{SL}(n, \mathbb{R})$).

**Concrete integral decompositions**   From Theorem 4.2 and the fact that symmetric matrices have a unique square root, we actually have equivalent decompositions for $A \in G$ as $A = PR$ or $A = S^{1/2}R$ for $S, P \in M$, $R \in H$ and $P = S^{1/2}$. For $\mathrm{GL}(n, \mathbb{R})$, the decomposition $A = S^{1/2}R$, has a factorization of the Haar measure of $\mathrm{GL}(n, \mathbb{R})$ as a product of invariant measures on $\mathrm{Pos}(n, \mathbb{R})$ (shortened $\mathrm{Pos}(n)$) and $\mathrm{O}(n)$. Let $\mu_{\mathrm{Pos}(n)}$ denote the $\mathrm{GL}(n, \mathbb{R})$ invariant measure on $\mathrm{Pos}(n)$.

**Theorem 4.3.** *Denote $G = \mathrm{GL}(n, \mathbb{R})$, $H = \mathrm{O}(n)$, and let $\mu_G$ be the Haar measure on $G$ and $\mu_H$ the Haar measure on $H$ normalized by $\mathrm{Vol}(H) = 1$. For $A \in G$, under the decomposition $A = S^{1/2}R$, $S \in \mathrm{Pos}(n)$, $R \in H$, the measure on $G$ splits as $d\mu_G(A) = \beta_n d\mu_{\mathrm{Pos}(n)}(S) d\mu_H(R)$, where $\beta_n = \frac{\mathrm{Vol}(\mathrm{O}(n))}{2^n}$ is a normalizing constant. Restricting to $G = \mathrm{GL}^+(n, \mathbb{R})$ and $H = \mathrm{SO}(n)$ and ignoring constants, we have:*

$$f \mapsto \int_G f(A) \mathrm{d}\mu_G(A) = \int_{\mathrm{Pos}(n)} \int_H f(S^{1/2}R) \mathrm{d}\mu_H(R) \mathrm{d}\mu_{\mathrm{Pos}(n)}(S), \; \forall f \in C_c(G) \quad (24)$$

The Haar measure of $\mathrm{GL}(n, \mathbb{R})$ is $d\mu_{\mathrm{GL}(n,\mathbb{R})}(A) = |\det(A)|^{-n} dA$, with $dA$ the Lebesgue measure on $\mathbb{R}^{n^2}$. We now describe how to sample on the individual factors to obtain $\mathrm{GL}(n, \mathbb{R})$ samples.

**Theorem 4.4.** *If a random matrix $A \in \mathrm{GL}(n, \mathbb{R})$ has a left-$\mathrm{O}(n)$ invariant density function relative to $|AA^T|^{-n/2}dA$, then $(AA^T)^{1/2} = S^{1/2}$ and $R = (AA^T)^{-1/2}A$ are independent random matrices and $R$ has a uniform probability distribution on $\mathrm{O}(n)$. The uniform distribution on $\mathrm{O}(n)$ will be the normalized Haar measure $\mu_{\mathrm{O}(n)}$. Conversely, if $S \in \mathrm{Pos}(n)$ has a density function $f : \mathrm{Pos}(n) \to \mathbb{R}_{\geq 0}$ relative to $\mu_{\mathrm{Pos}(n)}$ and $R \in \mathrm{O}(n)$ is uniformly distributed with respect to the Haar measure $\mu_{\mathrm{O}(n)}$, then $A = S^{1/2}R$ has a density function $\beta_n^{-1} f(AA^T)|\det(A)|^{-n}$ relative to $dA$.*

Theorems 4.3 and 4.4 are known results that appear in the random matrix theory literature, but have not seen recent application in the context of deep learning. In (B.6) we provide more details and references. Using the decomposition $A = S^{1/2}R$ invariant integration problems on $G$ can be transferred to the product space $M \times H$, and we can express up to normalization the invariant measure $\mu_G$ as $\varphi_*(\mu_M \otimes \mu_H)$. To construct samples $\{\mathbf{A}_1, \ldots, \mathbf{A}_n\} \sim \mu_G$ one produces samples $\{\mathbf{R}_1, \ldots, \mathbf{R}_n\} \sim \mu_H$ where $\mu_H$ will be the uniform distribution on $H$, and samples $\{\mathbf{M}_1, \ldots, \mathbf{M}_n\} \sim \mu_M$. Then $\mu_G$-distributed random values are obtained by $\{\mathbf{A}_1, \ldots, \mathbf{A}_n\} = \{\varphi(\mathbf{M}_1, \mathbf{R}_1), \ldots, \varphi(\mathbf{M}_n, \mathbf{R}_n)\}$, where again $\varphi : M \times H \to G$ is given by $\varphi : (S, R) \mapsto S^{1/2}R$.

**Mapping to the Lie algebra and back**   Any $A \in G$ can be expressed uniquely as $A = e^X R$ for $X \in \mathfrak{m}$ and $R \in H$. Since $H = \mathrm{SO}(n)$ in both cases, the fact that $\mathrm{expm} : \mathfrak{so}(n) \to \mathrm{SO}(n)$ is surjective, allows us to write it $A = e^X e^Y$, $Y \in \mathfrak{so}(n)$. The factors $X$ and $R = e^Y$ are obtained using $\Phi^{-1}$ (22). Then by taking the principal branch of the matrix logarithm on $H = \mathrm{SO}(n)$, $Y = \mathrm{logm}(R)$. A map $\xi^{-1} : G \to \mathfrak{g}$ as described in (8) and (9) is constructed as $\xi^{-1} = (\mathrm{id}_{\mathfrak{m}} \times \mathrm{logm}) \circ \Phi^{-1}$. More precisely, for any $A = e^X e^Y \in G$, using $\xi^{-1}$ we obtain the tangent vectors $(Y, X) \in \mathfrak{so}(n) \times \mathfrak{m}$ and since $\mathfrak{g} = \mathfrak{so}(n) \oplus \mathfrak{m}$ we have a unique $Z = X + Y \in \mathfrak{g}$. Details are given in (B.7).

Define $\tilde{K}_\theta := k_\theta \circ \xi^{-1} : G \to \mathbb{R}$ as our Lie algebra kernel. A Monte Carlo approximation of a cross-correlation operator $C_k : L^2(G) \to L^2(G)$ as in (9) will be of the form:

$$C_k f : g \mapsto \frac{1}{N} \sum_{i=1}^N f(\tilde{g}_i) \tilde{K}_\theta(\tilde{g}_i^{-1} g), \; \tilde{g}_i \sim \mu_G, \quad \forall g \in G \quad (25)$$

For affine groups, every element $(x, A)$ of $\mathbb{R}^n \rtimes G$, can be uniquely decomposed as $(x, I)(0, A)$, with $I$ the $n \times n$ identity matrix. One can use the fact that $\mathcal{L}_{(x,A)} = \mathcal{L}_{(x,I)}\mathcal{L}_{(0,A)}$ to write:

$$k((x, A)^{-1}(\tilde{x}, \tilde{A})) = \mathcal{L}_{(x,A)} k(\tilde{x}, \tilde{A}) = \mathcal{L}_{(x,I)}[\mathcal{L}_{(0,A)} k(\tilde{x}, \tilde{A})] = \mathcal{L}_x[k(A^{-1}\tilde{x}, A^{-1}\tilde{A})] \quad (26)$$

An efficient implementation of a convolutional layer can be realised in practice for $n \in \{2, 3\}$ by first obtaining the transformed kernel $k(A^{-1}\tilde{x}, A^{-1}\tilde{A})$ and then applying the translation $\mathcal{L}_x$ using an efficient convolution routine, as done for example in Cohen & Welling (2016); Bekkers (2019).

In practice, the exact discretization of the translation factor $\mathbb{R}^n$ will depend on the support of the input data. For example, if our input signals are defined compactly on a grid (e.g. 2D images), we can approximate a continuous convolution (Finzi et al., 2020) by sampling the translation factor in a uniform grid of coordinates $\tilde{x} \sim [-1, 1]^n \subset \mathbb{R}^n$ as the parametrization $\xi_{\text{Aff}(G)} : \mathbb{R}^n \times \mathfrak{gl}(n, \mathbb{R}) \to G$ is the identity map for the first factor. We can then approximate a lifting cross-correlation layer by:

$$C_k^\uparrow f : (x, A) \mapsto \int_{\mathbb{R}^n} f(\tilde{x}) \mathcal{L}_x k(A^{-1}\tilde{x}) \delta(A^{-1}) \mathrm{d}\tilde{x} \tag{27}$$

$$\approx \frac{1}{N} \sum_{i=1}^N f(\tilde{x}_i) \mathcal{L}_x [k_\theta(A^{-1}\tilde{x}_i) \delta(A^{-1})], \ x_i \sim [-1, 1]^n \tag{28}$$

For the non-lifting layers, starting from (17), denoting $\mathrm{d}\tilde{A} = \mathrm{d}\mu_G(\tilde{A})$ and applying (26) we have:

$$[C_k f](x, A) = \int_{\mathbb{R}^n} \int_G f(\tilde{x}, \tilde{A}) \mathcal{L}_x k(A^{-1}\tilde{x}, A^{-1}\tilde{A}) \delta(\tilde{A}^{-1}) \mathrm{d}\tilde{x} \mathrm{d}\tilde{A} \tag{29}$$

Using Theorem 4.3, denote the invariant measures $\mu_M$ and $\mu_H$ by $\mathrm{d}S$ and $\mathrm{d}R$, we obtain:

$$[C_k f](x, A) = \beta_n \int_{\mathbb{R}^n} \int_H \int_M f(\tilde{x}, S^{1/2}R) \mathcal{L}_x k(A^{-1}\tilde{x}, A^{-1}S^{1/2}R) \delta(S^{-1/2}) \mathrm{d}S \mathrm{d}R \mathrm{d}\tilde{x} \tag{30}$$

The kernel in (25) is now of the form $K_\theta : \mathbb{R}^n \rtimes G \to \mathbb{R}$, giving us:

$$[C_k f](x, A) \approx \frac{V}{N} \sum_{i=1}^N f(\tilde{x}_i, S_i^{1/2}R_i) \mathcal{L}_x [K_\theta(A^{-1}\tilde{x}_i, \xi^{-1}(A^{-1}S_i^{1/2}R_i)) \delta(S_i^{-1/2})] \tag{31}$$

where $\tilde{x}_i \sim [-1, 1]^n$, $S_i \times R_i \sim (\mu_M \otimes \mu_H)$, and $R_i$ sampled uniformly with respect $\mu_H$. $V$ records both the volume of the integration space from the MC approximation as well as the constant $\beta_n$.

## 5 EXPERIMENTS

For all experiments we use a ResNet-style architecture, replacing convolutional layers with cross-correlations that are equivariant (in expectation) with respect to the groups $\mathbb{R}^2 \rtimes \mathrm{GL}^+(2, \mathbb{R})$ and $\mathbb{R}^2 \rtimes \mathrm{SL}(2, \mathbb{R})$. Details regarding the network architecture and training are given in Appendix D.

**Affine-transformation invariance**   We evaluate our model on a benchmark affine-invariant image classification task employing the affNIST dataset[2]. The main works we compare with are the affine-equivariant model of MacDonald et al. (2022) and the Capsule Networks Ribeiro et al. (2020a;b) which are state of the art for this task. The experimental setup involves training on the standard set of 50000 non-transformed MNIST images (padded to $40 \times 40$), and evaluating on the affNIST test set, which consists of 320000 affine-transformed MNIST images. The model never sees the transformed affNIST images during training, and we do not use any data augmentation techniques. In this case, robustness with respect to the larger groups of the affine family of transformations is needed. For a fair comparison we roughly equalize the number of parameters with the referenced models.

Table 1: affNIST classification accuracy, after training on MNIST.

| Model | affNIST Acc. | MNIST Acc. | Parameters | MC. Samples |
|---|---|---|---|---|
| $\mathbb{R}^2 \rtimes \mathrm{SL}(2, \mathbb{R})$ | 98.2($\pm$0.1) | 99.55($\pm$0.1) | 370K | 10 |
| VB CapsNet (Ribeiro et al., 2020b) | 98.1 | 99.7 | 175K | — |
| RU CapsNet (Ribeiro et al., 2020a) | 97.69 | 99.72 | $> 580$K | — |
| $\mathbb{R}^2 \rtimes \mathrm{GL}^+(2, \mathbb{R})$ | 97.4($\pm$0.2) | 99.5($\pm$0.1) | 395K | 10 |
| affConv (MacDonald et al., 2022) | 95.08 | 98.7 | 374K | 100 |
| affine CapsNet (Gu & Tresp, 2020) | 93.21 | 99.23 | — | — |
| Equivariant CapsNet (Lenssen et al., 2018) | 89.1 | 98.42 | 235K | — |

---

[2]http://www.cs.toronto.edu/tijmen/affNIST

Table 1 reports the average test performance of our model at the final epoch, over five training runs with different initialisations. We observe that our equivariant models are robust and generalize well, with the $\mathbb{R}^2 \rtimes \mathrm{SL}(2, \mathbb{R})$ model outperforming all previous equivariant models and Capsule Networks. Note that, compared to MacDonald et al. (2022), our sampling scheme requires 10 times less samples to realize an accurate Monte Carlo approximation of the convolution. The $\mathbb{R}^2 \rtimes \mathrm{GL}^+(2, \mathbb{R})$ model performs slightly worse than the volume-preserving affine group $\mathbb{R}^2 \rtimes \mathrm{SL}(2, \mathbb{R})$. This can be explained by considering that the affNIST dataset contains only a small degree of scaling.

**Homography transformations** We further evaluate and report in Table 2 the performance of the same model evaluate on the homNIST dataset of MacDonald et al. (2022). The setup is identical to the affNIST case, with the images now being transformed by random homographies. We observe a similar degree of robustness in this case, again outperforming previous methods applied to this task.

Table 2: homNIST classification.

| Model | homNIST Acc. | MC. Samples |
|---|---|---|
| $\mathbb{R}^2 \rtimes \mathrm{SL}(2, \mathbb{R})$ | $98.3(\pm0.1)$ | 10 |
| $\mathbb{R}^2 \rtimes \mathrm{GL}^+(2, \mathbb{R})$ | $97.71(\pm0.1)$ | 10 |
| affConv (MacDonald et al., 2022) | 95.71 | 100 |

As our models are only equivariant in expectation, we analyze numerically in Sec. D the degree to which the *equivariance error* is dependent on the number of Monte Carlo samples used to approximate the convolution/cross-correlation integral.

## 6 CONCLUSION

We have built a framework for constructing equivariant networks when working with matrix Lie groups that are not necessarily compact or abelian. Using the structure theory of semisimple/reductive Lie groups we have shown one possible avenue for constructing invariant/equivariant (convolutional) layers primarily relying on tools which allow us to decompose larger groups into smaller ones. In our preliminary experiments, the robustness and out-of-distribution capabilities of the equivariant models were shown to outperform previous proposals on tasks where the symmetry group of relevance is one of $\mathrm{GL}^+(n, \mathbb{R})$ or $\mathrm{SL}(n, \mathbb{R})$.

Our contribution is largely theoretical, providing a framework by which equivariance/invariance to complex symmetry groups can be obtained. Further experiments will look to validate the applicability of our method to other data modalities, such as point clouds or molecules, as in Finzi et al. (2020).

While we have primarily focused on convolution operators, we remark that the tools explored here are immediately applicable to closely-related machine learning models which employ Lie groups and their regular representation for invariance/equivariance. For example, the 'LieTransformer' architecture proposed in Hutchinson et al. (2021) opts to replace convolutional layers with self-attention layers, while still using the Lie algebra of the group as a mechanism for incorporating positional information. They face the same challenge in that their parametrization is dependent on the mapping elements back and forth between a chosen Lie group and its Lie algebra, and they require a mechanism for sampling on the desired group. The methods presented here are directly applicable in this case.

Future work will explore expanding the class of Lie groups employed by such models using the tools presented here. Another potential avenue to explore is the applicability of the presented tools to the problem of 'partial' and 'learned' invariance/equivariance (Benton et al., 2020). The sampling mechanism of the product decomposition allows one to specify a probability distribution for the non-orthogonal factor, which could be learned from data.

ACKNOWLEDGMENTS

The presentation of this paper at the conference was financially supported by the Amsterdam ELLIS Unit and Qualcomm.

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

# Appendix

## Table of Contents

## A ADDITIONAL BACKGROUND

A symmetry group refers to a set of transformations which preserve some underlying structure present in the data. Formally, a group $G$ is a set together with an associative binary operation $G \times G \to G$ which tells us how group elements can be composed to form another. Every element $g \in G$ has an inverse $g^{-1} \in G$, and the group has an identity element $e \in G$ such that $gg^{-1} = e$.

### A.1 TOPOLOGICAL GROUPS AND THE HAAR MEASURE

A Hausdorff[3] topological space $M$ is **locally compact** if each point $m \in M$ has a compact neighborhood. A **topological group** $G$ is a group as well as a Hausdorff topological space such that the group operation $(g, h) \mapsto gh$ and inversion map $g \mapsto g^{-1}$ are continuous. For the following and more details on Radon measures see Folland (1999); Lang (2012).

**Radon measures** Let $(X, \mathcal{B}_X, \mu)$ be a measure space with Hausdorff topological space $X$, $\mathcal{B}_X$ the $\sigma$-algebra of Borel sets and $\mu : \mathcal{B}_X \to [0, \infty]$ any measure on $\mathcal{B}_X$ (referred to as a Borel measure). The measure $\mu$ is called **locally finite** if every point $x \in X$ has an open neighborhood $U \ni x$ for which $\mu(U) < \infty$. A Borel set $E \subseteq X$ ($E \in B_X$) is called **inner regular** if:

$$\mu(E) = \sup\{\mu(K) \mid K \subseteq E, \ K \text{ compact}\} \tag{32}$$

Respectively, a Borel set $E \subseteq X$ is called **outer regular** if:

$$\mu(E) = \inf\{\mu(V) \mid V \supseteq E, \ \text{V open}\} \tag{33}$$

The measure $\mu$ is called **inner** (**outer**) regular if all Borel sets are inner (outer) regular. It is called **regular** if it is both inner and outer regular. $\mu$ is a **Radon measure** if it is locally finite, inner regular on open sets and outer regular. $\mu$ is $\sigma$**-finite** if there exists a countable family of Borel sets $\{B_n\}_{n \in \mathbb{N}}$, where $\mu(B_n) < \infty$, $\forall n \in N$ and $\bigcup_{n \in N} B_n = X$. If $\mu$ is a Borel measure on a Hausdorff topological space $X$, local finiteness will imply that $\mu$ is finite on compact subsets of $X$.

---

[3]A topological space where any two distinct points have disjoint neighbourhoods is called Hausdorff.

**Theorem A.1.** [4] *Every locally compact group $G$ has a left (right) nonzero Radon measure $\mu_G$ such that $\mu_G(gB) = \mu_G(B)$ (respectively $\mu_G(Bg) = \mu_G(B)$) for any Borel subset $B \subseteq G$ and any $g \in G$. The measure $\mu_G$ is called the left (right) Haar measure of $G$ and if $\nu_G$ is another Haar measure on $G$, then $\mu_G = c \cdot \nu_G$ for some $c \in \mathbb{R}_{>0}$.*

When integrating with respect to the left Haar measure $\mu_G$ we have for any $f \in C_c(G)$:

$$\int_G f(yx) \mathrm{d}\mu_G = \int_G f(x) \mathrm{d}\mu_G, \ \forall y \in G \tag{34}$$

All of the groups and topological spaces in the main text will be $\sigma$-compact. A locally compact space $X$ is $\sigma$-compact (or countable at infinity) if it is a countable union of compact subsets. We will use the notation $\mu_G$ to refer to the left Haar measure and when needed the notation $\mu_L(\cdot)$ and $\mu_R(\cdot)$ will be used to differentiate left and right Haar measures.

**Remark A.2.** *If $X$ is a homogeneous space of $G$ (but not $G$ itself), then a $G$-invariant Radon measure $\mathrm{d}\mu_X$ on $X$ (if it exists) respects the same invariance property presented in Thm. A.1 and (34), and we simply refer to $\mathrm{d}\mu_X$ as a $G$-invariant measure. For a review of such measures, see (Folland, 2016, Chapter 2.6).*

**Function spaces** Suppose $(X, \mathcal{B}_X, \mu)$ is a measure space where $X$ is locally compact Hausdorff space and $\mu$ a Radon measure on $X$. $L^p_\mu(X, \mathbb{R}) := L^p(X)$ for $1 <= p < \infty$ denotes the space of equivalence classes of functions $\{f : X \to \mathbb{R} \mid f \text{ Borel measurable and } \int_X |f|^p \mathrm{d}\mu < \infty\}$ that agree $\mu$-almost everywhere. Equipped with the norm $\|f\|_p = (\int_X |f|^p \mathrm{d}\mu)^{1/p}$, $L^p(X)$ is a Banach space. $C(X) := C(X, \mathbb{R})$ denotes the space of continuous real-valued functions on $X$. The support of $f \in C(X)$ is defined as $\mathrm{supp}(f) = \overline{\{x \in X \mid f(x) \neq 0\}}$. We state that a function $f$ has compact support whenever $\mathrm{supp}(f)$ is compact. $C_c(X)$ denotes the subspace of $C(X)$ of continuous functions with compact support.

### A.2 EQUIVARIANT CONVOLUTIONAL OPERATORS

In this section we show that the convolution/cross-correlation operators (6), (7) as well as the lifting cross-correlation (5) are equivariant. We then clarify the existence and range of these operators. For a non-lifting convolution/cross-correlation operator, recall that $X = Y = G$ and $k : Y \times X \to \mathbb{R}$ is a bi-invariant kernel:

$$k(gx, gy) = k(x, y), \quad \forall (x, y) \in Y \times X, \ \forall g \in G \tag{35}$$

$\mu_X$ is a $G$-invariant Radon measure on $X$. The operator $C_k$ was defined for any $f \in L^1_{\mu_X}(X)$:

$$C_k : f \mapsto C_k f(y) = \int_X f(x) k(x, y) \mathrm{d}\mu_X(x), \quad \forall y \in Y \tag{36}$$

$C_k : f \mapsto C_k f$ maps an element of $L^1(X)$ to $L^1(G)$, respecting $\mathcal{L}_g \circ C_k = C_k \circ \mathcal{L}_g$ for any $g \in G$:

$$\mathcal{L}_g(C_k f)(y) = C_k f(g^{-1}y) = \int_X f(x) k(x, g^{-1}y) \mathrm{d}\mu_X(x) \tag{37}$$

$$(\text{Haar invariance}) \ = \int_X f(g^{-1}x) k(g^{-1}x, g^{-1}y) \mathrm{d}\mu_X(x) \tag{38}$$

$$(\text{Kernel bi-invariance}) = \int_X f(g^{-1}x) k(x, y) \mathrm{d}\mu_X(x) = C_k(\mathcal{L}_g[f])(y) \tag{39}$$

A lifting cross-correlation operator $C_k^\uparrow$ was defined for $f \in L^1(X)$ by:

$$C_k^\uparrow : f \mapsto C_k^\uparrow f, \quad C_k^\uparrow f : g \mapsto \int_X f(x) k(g^{-1}x) \delta(g^{-1}) \mathrm{d}\mu_X(x), \ \forall g \in G \tag{40}$$

---

[4] See Proposition 2.9 and Theorems 2.10 & 2.20 of Folland (2016).

where $X$ is a homogeneous $G$-space with Radon measure $\mathrm{d}\mu_X$ and $\delta : G \to \mathbb{R}^{\times}_{>0}$ records the change of variables produced by the action of $G$:

$$\int_X f(x)\mathrm{d}\mu_X(x) = \int_X f(gx)\delta(g)\mathrm{d}\mu_X(x) \tag{41}$$

In the case where $X = \mathbb{R}^n$ and $Y = \mathrm{Aff}(G)$, for $\mathrm{Aff}(G)$ the semi-direct product $\mathrm{Aff}(G) = \mathbb{R}^n \rtimes G$, $G \leq \mathrm{GL}(n, \mathbb{R})$, we have $\mathrm{Aff}(G)$ acting transitively on $X$, and we can make the identification $X \cong \mathrm{Aff}(G)/G$. Each element $g \in \mathrm{Aff}(G)$ can be represented as $g = (x, h)$ with $x \in \mathbb{R}^n, h \in G$. In this case, we have $\delta(g) = |\det(g)| = |\det(h)|$. Note that the form of the kernel can equivalently be derived through an equivariance constraint relation as in Bekkers (2019, Theorem 1). The lifting layer is then more concretely of the form $C_k^{\uparrow} f = f \star k$ where:

$$(f \star k)(g) = \int_{\mathbb{R}^n} f(x)k(g^{-1}x)\frac{\mathrm{d}x}{|\det(h)|} \tag{42}$$

The lifting layer is equivariant ($\mathcal{L}_{\tilde{g}}[f \star k] = [\mathcal{L}_{\tilde{g}}f] \star k$), since for any $\tilde{g} = (\tilde{x}, \tilde{h}) \in \mathrm{Aff}(G)$ we have:

$$\mathcal{L}_{\tilde{g}}[f \star k](g) = (f \star k)(\tilde{g}^{-1}g) = \int_{\mathbb{R}^n} f(x)k(g^{-1}\tilde{g}x)\frac{\mathrm{d}x}{|\det(\tilde{h}^{-1}h)|} \tag{43}$$

$$(x \mapsto \tilde{g}^{-1}x) = \int_{\mathbb{R}^n} f(\tilde{g}^{-1}x)k(g^{-1}x)|\det(\tilde{h}^{-1})|\frac{\mathrm{d}x}{|\det(\tilde{h}^{-1}h)|} \tag{44}$$

$$= \int_{\mathbb{R}^n} \mathcal{L}_{\tilde{g}}f(x)k(g^{-1}x)\frac{\mathrm{d}x}{|\det(h)|} = ([\mathcal{L}_{\tilde{g}}f] \star k)(g) \tag{45}$$

A similar derivation appears in MacDonald et al. (2022, Theorem 4.2). The lifting layer equivariance here was derived for the case where $X$ is the homogeneous space $\mathbb{R}^n = \mathrm{Aff}(G)/G$, however a similar process is available for more general homogeneous spaces. One only has to identify the appropriate *relatively* invariant measure $\delta(\cdot)^{-1}\mathrm{d}\mu_X$ which appears in (5). For more details on relatively invariant measures see Hewitt & Ross (2012).

### A.2.1 EXISTENCE AND RANGE OF CONVOLUTION OPERATORS

If $G$ is a locally compact Hausdorff group, $C_c(G)$ is dense in $L^p(G)$ for $1 \leq p < \infty$[5]. We can therefore approximate functions in $L^p(G)$ using functions in $C_c(G)$. While some results hold in a more general setting, we assume that all topological groups are (locally) compact Hausdorff and second countable (and therefore $\sigma$-compact), as the Lie groups of interest satisfy these properties.

**Proposition A.3.** [6] *We record the following results concerning the existence and range of the convolution operators for a locally compact group $G$.*

1. *If $f \in L^1(G)$ and for any $k \in L^p(G)$ ($1 <= p <= \infty$) then $\int_G f(\tilde{g})k(\tilde{g}^{-1}g)\mathrm{d}\mu_G(\tilde{g})$ converges absolutely, $f * k \in L^p(G)$ and $\|f * k\|_p \leq \|f\|_1\|k\|_p$.*

2. *If $G$ is not unimodular, $f \in L^p(G)$ and $k \in L^1(G) \cap C_c(G)$ then $f * k \in L^p(G)$.*

3. *If $f, k \in L^2(G)$ then $f * k \in C_0(G)$, see also (Hewitt & Ross, 2012, Theorem 20.16).*

For the cross-correlation operator, if we define the involution $f^*(g) = f(g^{-1})$, we can write $f \star k$ as $f * k^*$, and reuse the results of Proposition A.3. In the case of a nonunimodular semi-direct product group $G = N \rtimes H$, where $H$ *is* unimodular, the Haar measure is of the form $\mu_G = \Delta_G \cdot \mu_N \otimes \mu_H$[7]. Since $\Delta_G : G \to (0, \infty)$ is unbounded, we therefore always make the assumption that the support of $fk$ is a compact set where $\Delta_G$ is bounded.

**Assumption.** *We assume $f, k \in L^1(G) \cap L^2(G)$ and additionally $k \in C_c(G)$. When working with image data, one can also take $f \in C_c(G)$ directly. Going forward we therefore establish for $p \in \{1, 2\}$, $C_k^{\uparrow} : L^p(X) \to L^p(G)$ defined by (5) and $C_k^*$ or $C_k$ are similarly operators $L^p(G) \to L^p(G)$ given by (6) and (7).*

---

[5](Folland, 1999, Proposition 7.9).

[6](Folland, 2016, Propositions 2.40 & 2.41).

[7]Corresponds to Thm. 4.1(2) in integral form. A more precise reference is Wijsman (1990, Corollary 7.6.3).

The restriction to a compact *subset* of $G$ can also be motivated if we wish to employ a Monte Carlo approximation of the integral using a uniform distribution, since the Haar measure is finite on compact subsets. However, the restriction will not tied to the injectivity/surjectivity radius of the group exponential map, as we will use a different parametrization as shown in the main text.

## A.3 RELATED WORK

The theory behind constructing equivariant convolutional layers when our input space is a homogeneous space of some locally compact topological group is covered in Cohen & Welling (2016); Kondor & Trivedi (2018); Cohen et al. (2019); Bekkers (2019). A differential geometric formulation not necessarily limited to homogeneous spaces is given in Weiler et al. (2021; 2023) and a review focused on the application of induced representations in the context of neural networks can be found in Kondor & Trivedi (2018); Cohen et al. (2019).

Employing Monte Carlo integration to approximate convolution integrals has also been proposed e.g. in Hermosilla et al. (2018); Finzi et al. (2020); Romero et al. (2022); Knigge et al. (2022). In Hutchinson et al. (2021) the Lie algebra parametrization is employed with convolution operators being replaced with self-attention layers. Steerable CNNs make use of group representation theory to parametrise convolution kernels by solving a kernel steerability constraint Cohen & Welling (2017); Weiler & Cesa (2019); Lang & Weiler (2021); Cesa et al. (2022).

In the context of finite-dimensional group representation Finzi et al. (2021) present a general solution for constructing equivariant MLPs. They present a framework for solving the equivariance constraint by making use of the generators of the Lie algebra of a group. The resulting linear system is solved for finite dimensional representations by using the singular value decomposition. More general solutions are developed in Bogatskiy et al. (2020); Batatia et al. (2023) for a wider class of Lie groups and representations.

**MacDonald et al. (2022)** The limitations of previous Lie algebra methods (as reviewed in Sec. 4) are also discussed in MacDonald et al. (2022), which proposes a possible solution while still working with the group exponential. To overcome its lack of surjectivity and be able to sample with respect to the Haar measure of $\mathrm{Aff}(\mathrm{GL}(n,\mathbb{R})) = \mathbb{R}^n \rtimes \mathrm{GL}(n,\mathbb{R})$, the domain of integration itself is restricted and the convolution integral is reformulated, ensuring that the group elements $\tilde{g}^{-1}g$ (using the convolution operator notation) are within the injectivity radius of the exponential map. This is done by first changing the convolution operator after the lifting layer to work with the right Haar measure[8]:

$$(f * k)(g) = \int_G f(\tilde{g})k(\tilde{g}^{-1}g)\mathrm{d}\mu_L(\tilde{g}) = \int_G f(g\tilde{g}^{-1})k(\tilde{g})\mathrm{d}\mu_R(\tilde{g}) \tag{46}$$

For non-lifting layers instead of the kernel $k(\cdot)$, the feature map $f(\cdot)$ is now evaluated at $f(g\tilde{g}^{-1})$.

**Proposition A.4.** [9] *Let $G$ be a Lie group with Lie algebra $\mathfrak{g}$ and suppose $U \subseteq \mathfrak{g}$ is a neighborhood of $0 \in \mathfrak{g}$ and $\mathrm{expm}(U)$ a neighborhood in $e \in G$ such that the group exponential $\mathrm{expm} : \mathfrak{g} \to G$ is a diffeomorphism of $U$ onto $\mathrm{expm}(U)$. For $f \in C_c(G)$ with support in $\mathrm{expm}(U)$ we have:*

$$\int_G f(g)\mathrm{d}\mu_G(g) = \int_\mathfrak{g} f(\mathrm{expm}(X)) \det\left(\frac{1 - e^{-ad_{-X}}}{ad_{-X}}\right)\mathrm{d}X \tag{47}$$

*where $\det\left(\frac{1-e^{-ad_{-X}}}{ad_{-X}}\right)$ is the Jacobian determinant of the differential of $\mathrm{expm}$ and $\mathrm{d}X$ is the Lebesgue measure on $\mathfrak{g}$.*

The change of variables of Proposition A.4 and the expression (46) allow MacDonald et al. (2022) to define the non-lifting convolutional layers to be of the form:

$$(f * k)(g) = \int_G f(g\tilde{g}^{-1})k(\tilde{g})\mathrm{d}\mu_R(\tilde{g}) = \int_\mathfrak{g} f(ge^{-X})\tilde{k}_\theta(X) \det\left(\frac{1 - e^{-ad_{-X}}}{ad_{-X}}\right)\mathrm{d}X \tag{48}$$

---

[8]See (MacDonald et al., 2022, Theorem 4.3) or (Folland, 2016, (2.32) & (2.36)). Here, we used the convolution operator to be consistent with the derivation of MacDonald et al. (2022).

[9](Helgason, 1984, Theorem 1.14), (MacDonald et al., 2022, Theorem 4.4).

$\tilde{k}_\theta$ is again a learnable map that takes in Lie algebra elements approximating $k \circ \text{expm} : \mathfrak{g} \to \mathbb{R}$. The difference here lies in the fact that one does not need to use the inverse map $\xi^{-1}$ to map back to $\mathfrak{g}$. By treating $\det\left(\frac{1-e^{-\text{ad}-X}}{\text{ad}-X}\right)$ as (proportional to) a density function, sampling is realised directly on the Lie algebra using standard MCMC methods. One starts the sampling process from the outer-most convolution integral and rejects samples that lie outside of the support of the exponential map.

While the approach can be applied to any matrix Lie group, in practice because every possible $\tilde{g}^{-1}g$ element must be precalculated and kept in memory before the forward pass, the scalability of the method is greatly limited due to its memory requirements. Further numerical errors are introduced due to the fact that $\det\left(\frac{1-e^{-\text{ad}-X}}{\text{ad}-X}\right)$ is approximated with limited precision by the power series expression $\left(\frac{1-e^{-\text{ad}-X}}{\text{ad}-X}\right) = \sum_{k=0}^{\infty} \frac{(-1)^k}{(k+1)!}(\text{ad}_X)^k$. Additionally, while a discretization of the integral is always necessary, this approach is limited in that the domain of integration must still be restricted to the injectivity/surjectivity radius of the group exponential for the change of variables to apply. In this paper, we also employ a change of variables in the context of invariant integration with respect to the Haar measure. However, rather than working with the tangent space of the group as in Prop. A.4, we make use of group-level decompositions into independent factor spaces, and show that the Haar measure decomposes as a product of invariant measures, allowing us to construct a sampling scheme on the lower-dimensional subcomponents, as explained in Sections 4.1 and B.6.

# B LIE GROUP DECOMPOSITIONS

## B.1 LIE GROUPS

A **Lie group** $G$ is a group as well as a smooth manifold, such that both the group operation and the inversion map are smooth. Lie groups are therefore second countable Hausdorff topological spaces. An abelian Lie group $G$ is a Lie group and an abelian group, i.e. a group for which the order of the group operation does not matter $gh = hg$, $\forall g, h \in G$. $\text{M}_{nn}(\mathbb{R}) := \text{M}_n(\mathbb{R})$ denotes the vector space of $n \times n$ matrices. It is canonically isomorphic to $\mathbb{R}^{n^2}$, which is locally compact.

Closed or open subsets of $\text{M}_n(\mathbb{R})$ will be locally compact with respect to the induced topology[10]. One such open subset is $\text{GL}(n, \mathbb{R})$, the Lie group of invertible matrices:

$$\text{GL}(n, \mathbb{R}) = \{X \in \text{M}_n(\mathbb{R}) \mid \det(X) \neq 0\} \tag{49}$$

The notation $H \leq G$ ($H < G$) is used to indicate that $H$ is a (proper) subgroup of $G$, rather than just a (proper) subset $H \subseteq G$ ($H \subset G$). We are only interested in closed Lie subgroups of $\text{GL}(n, \mathbb{R})$.

A (closed) *Lie subgroup*[11] $H$ of a Lie group $G$ will refer to a closed subgroup and a submanifold of $G$ (with the induced topology). A *linear* or **matrix Lie group** is defined to be a Lie subgroup of $\text{GL}(n, \mathbb{R})$, and will therefore be locally compact and second countable. $\text{GL}(n, \mathbb{R})$, the translation group $(\mathbb{R}^n, +)$ and the family of affine groups $\mathbb{R}^n \rtimes H$, $H \leq \text{GL}(n, \mathbb{R})$ are our primary interest, with $H$ being one of the groups:

- $\text{GL}^+(n, \mathbb{R}) = \{X \in \text{GL}(n, \mathbb{R}) \mid \det(X) > 0\}$, the identity component of $\text{GL}(n, \mathbb{R})$; It it also referred to as the *positive* general linear group;
- $\text{SL}(n, \mathbb{R}) = \{X \in \text{GL}(n, \mathbb{R}) \mid \det(X) = 1\}$, the special linear group;
- $\text{O}(n) = \{X \in \text{GL}(n, \mathbb{R}) \mid X^T X = \text{I}_n\}$, the orthogonal group;
- $\text{SO}(n) = \{X \in \text{O}(n) \mid \det(X) = 1\}$, the special orthogonal group.

**Proposition B.1.** [12] *The group exponential map* $\text{expm} : \mathfrak{g} \to G$ *is smooth with* $d(\text{expm})_0 = \text{id}$, *making* $\text{expm}$ *a diffeomorphism* $\text{expm}|_U : U \to V$ *of some neighborhood* $U$ *of* $0 \in \mathfrak{g}$ *onto a neighborhood* $V$ *of* $e \in G$.

The notation $\text{expm} : \mathfrak{g} \to G$ for the Lie group exponential is used to differentiate it from the Riemannian exponential, which is mentioned later on. Unless the group $G$ can be equipped with

---

[10](Lee, 2010, Proposition 4.66).

[11]An abstract subgroup $H$ of $G$ is a submanifold iff $H$ is closed with the induced topology ((Gallier & Quaintance, 2020, Theorem 19.18)).

[12](Lee, 2013, Proposition 20.8).

a bi-invariant Riemannian metric, the exponentials do not coincide (see Proposition B.10). $M_n(\mathbb{R})$ equipped with the matrix commutator $[X, Y] = XY - YX$ for $X, Y \in M_n(\mathbb{R})$ is a Lie algebra, and more precisely it is (canonically isomorphic to) the Lie algebra of $GL(n, \mathbb{R})$[13]. We use the notation $\mathfrak{gl}(n, \mathbb{R}) = M_n(\mathbb{R})$ when working with this identification. For $G = GL(n, \mathbb{R})$ with $\mathfrak{g} = \mathfrak{gl}(n, \mathbb{R})$, the group exponential is given by the matrix exponential:

$$\text{expm} : \mathfrak{gl}(n, \mathbb{R}) \to GL(n, \mathbb{R}), \quad X \mapsto e^X = \sum_{k=0}^{\infty} \frac{1}{k!} X^k \tag{50}$$

From Prop. B.1 we can define the inverse of the group exponential $(\text{expm}|_U)^{-1} : V \to U$ which is a diffeomorphism of $V$ onto $U$. For matrix Lie groups this map is the matrix logarithm which we denote by $\text{logm}(\cdot)$. Its power series expression is:

$$\text{logm}(A) = \sum_{i=1}^{\infty} \frac{(-1)^{k+1}}{k} (A - I)^k, \quad A \in GL(n, \mathbb{R}) \tag{51}$$

The existence of the inverse of the matrix exponential is characterized as follows.

**Proposition B.2.** [14] *Let* $B(I, 1) = \{X \in M_n(\mathbb{R}) \mid \|X - I\| < 1\}$ *where* $\|\cdot\|$ *is a norm on* $M_n(\mathbb{R})$ *(e.g. Frobenius norm) and* $I$ *the identity matrix. Note that* $B(I, 1) \subseteq GL(n, \mathbb{R})$. *Then for every* $g \in B(I, 1)$ *we have* $\text{expm}(\text{logm}(g)) = g$ *and for every* $X \in B(0, \log(2))$, *we have* $\text{logm}(\text{expm}(X)) = X$.

Recall from Section 4 that if the parametrization map $\xi : \mathfrak{g} \to G$ is chosen to be the group exponential $\xi = \text{expm}$, then $\xi^{-1}$ is given by the matrix logarithm:

$$\xi^{-1}(g^{-1}\tilde{g}) = \text{logm}(g^{-1}\tilde{g}), \quad \text{logm} : G \to \mathfrak{g} \tag{52}$$

For the following, see Hall (2015, Chapter 5). Assuming there exist $X$ and $Y$ such that $e^X = g^{-1}$ and $e^Y = \tilde{g}$, (10) can be rewritten as $\text{logm}(e^X e^Y)$. The Baker-Campbell-Hausdorff (BCH) formula states that there exists a sufficiently small open subset $0 \in U \subset \mathfrak{g}$ so that $e^X e^Y \in e^U$ and one has:

$$\text{logm}(e^X e^Y) = X + Y + \frac{1}{2}[X, Y] + \frac{1}{12}[X, [X, Y]] - \frac{1}{12}[Y, [X, Y]] + \dots \tag{53}$$

For abelian Lie groups this reduces to:

$$\text{logm}(e^X e^Y) = X + Y \tag{54}$$

For $V$ a finite-dimensional vector space over $\mathbb{R}$, we denote by $GL(V)$ the group of invertible linear maps of $V$ and by $\mathfrak{gl}(V, \mathbb{R}) = \text{End}(V)$ the space of linear maps $V \to V$. $GL(V)$ admits a Lie group structure as it is isomorphic to $GL(n, \mathbb{R})$ once a basis is chosen. The space $\mathfrak{gl}(V, \mathbb{R})$ can be made into a Lie algebra under the commutator bracket and it is isomorphic to $M_n(\mathbb{R}) = \mathfrak{gl}(n, \mathbb{R})$.

**Definition B.3.** *Let* $V$ *be a vector space. A (finite-dimensional real) representation of a Lie group* $G$ *is a Lie group homomorphism* $\rho : G \to GL(V)$. *For* $\mathfrak{g}$ *a (real) Lie algebra, a representation of* $\mathfrak{g}$ *is a Lie algebra homomorphism* $\phi : \mathfrak{g} \to \mathfrak{gl}(V, \mathbb{R})$.

The conjugation (inner automorphism) map $C_g : G \to G$ is defined such that $C_g = L_g \circ R_{g^{-1}}$:

$$C_g : G \to G, \quad C_g : h \mapsto ghg^{-1} \tag{55}$$

The **adjoint representation** $G$, is given by the homomorphism[15]

$$\text{Ad} : G \to GL(\mathfrak{g}), \ g \mapsto \text{Ad}_g \tag{56}$$

where $\text{Ad}_g : \mathfrak{g} \to \mathfrak{g}$, $\text{Ad}_g = d(C_g)_e = (dL_g)_{g^{-1}} \circ (dR_{g^{-1}})_e$. The differential of $\text{Ad}$ is used to define the **adjoint representation** of $\mathfrak{g}$, denoted by $\text{ad}$:

$$\text{ad} : \mathfrak{g} \to \mathfrak{gl}(\mathfrak{g}), \ \text{ad}_X = d(\text{Ad})_e(x) \tag{57}$$

---

[13](Lee, 2013, Example 8.36 & Proposition 8.41).

[14](Faraut, 2008, Theorem 2.2.1).

[15](Lee, 2013, Proposition 20.24).

## B.2 PRIMER ON RIEMANNIAN GEOMETRY

For more details on smooth and Riemannian manifolds see Lee (2013).

**Riemannian manifolds** A Riemannian metric (tensor) $g$ on a smooth manifold $M$ is a covariant 2-tensor field smoothly assigning to each $p \in M$, an inner product $g_p : T_p M \times T_p M \to \mathbb{R}$ at its tangent space $T_p M$:

$$p \mapsto g_p(\cdot, \cdot) = \langle \cdot, \cdot \rangle_p \tag{58}$$

A smooth manifold $M$ with a Riemannian metric $g$ is a *Riemannian manifold* $(M, g)$.

**Definition B.4.** *Let $(M, g)$ and $(N, h)$ be Riemannian manifolds. A map $\phi : M \to N$ is an* ***isometry*** *if $\phi$ is a diffeomorphism and $g = \phi^* h$. Equivalently, $\phi$ is bijective, smooth and $\forall p \in M$, $d\phi_p : T_p M \to T_{\phi(p)} N$ is a linear isometry:*

$$g_p(u, v) = h_{\phi(p)}(d\phi_p(u), d\phi_p(v)), \ \forall u, v \in T_p M \tag{59}$$

An *affine connection* is a bilinear map $\nabla$ that maps a pair of vector fields $X, Y$ to another vector field $\nabla_X Y$, which is the covariant derivative of $Y$ with respect to $X$. The affine connection allows us to define the notion of a parallel vector field. If $M$ is a smooth manifold and $\nabla$ a connection on $M$, then for $\gamma : I \to M$ a smooth curve, any vector field $X$ along $\gamma$ is called parallel if $\nabla_{\dot{\gamma}(t)} X = 0$ for any $t \in I$, where $\dot{\gamma}(t_0) := d\gamma_{t_0}\left(\frac{d}{dt}\big|_{t_0}\right)$. A smooth curve $\gamma : I \to M$ is a *geodesic* (with respect to $\nabla$) iff $\dot{\gamma}(t)$ is parallel along $\gamma$, that is $\nabla_{\dot{\gamma}(t)} \dot{\gamma}(t) = 0, \ \forall t \in I$. For every point $p \in M$ and every tangent vector $v \in T_p M$ there exists some interval $I = (-\eta, \eta), \eta > 0$ around 0 and a unique geodesic $\gamma : I \to M$ satisfying:

$$\gamma(0) = p, \ \text{and} \ \gamma'(0) = v \tag{60}$$

There exists a unique geodesic $\gamma$ satisfying these conditions and for which the domain $I$ cannot be extended. In this case, $\gamma$ is the unique **maximal geodesic** satisfying the initial conditions (60). We denote it by $\gamma_{p,v}$, and say that $\gamma_{p,v}$ is a geodesic through $p$ with initial velocity $v$.

**Definition B.5** (Exponential map of connection)**.** *Let $M$ be a manifold and $\nabla$ a connection on $M$. Define for a point $p \in M$ the set $D(p) = \{v \in T_p M \mid \gamma_{p,v}(1) \ \text{defined}\}$.*

*The exponential map $\exp_p : D(p) \to M$ is given by:*

$$\exp_p : v \mapsto \gamma_{p,v}(1) \tag{61}$$

**Proposition B.6.** [16] *The differential of the exponential map $d(\exp_p)$ at 0 is the identity on $T_p M$. For every $p \in M$, the exponential $\exp_p$ is a diffeomorphism from an open subset $U \subseteq T_p M$ centered at 0 such that $\exp_p(U) \subseteq M$ is open.*

It is therefore possible to build a local chart $(\exp_p(U), \exp_p^{-1})$ around every point $p \in M$ using the inverse of the exponential map $\exp_p^{-1}$. The Levi-Civita connection is the unique affine connection that is metric-compatible and torsion free.

**Definition B.7.** [17] *The exponential map of the Levi-Civita connection will be called the* ***Riemannian exponential****. For any $p \in M$, a* ***normal neighborhood*** *of $p$ is an open neighborhood $U_p = \exp_p(B(0, \epsilon))$ where $\exp_p$ is a diffeomorphism from the open ball $B(0, \epsilon) \subseteq T_p M$ onto $U_p$. The* ***injectivity radius*** $\text{inj}_M(p)$ *at $p$ is the least upper bound value $\epsilon > 0$ such that $\exp_p$ is a diffeomorphism on $B(0, \epsilon)$. The chart $(\exp_p(B(0, \text{inj}_M(p))), \exp_p^{-1})$ is called a* ***normal chart*** *and the inverse of the exponential $\exp_p^{-1}$ is the* ***Riemannian logarithm****.*

### B.2.1 LIE GROUPS AS RIEMANNIAN MANIFOLDS

A Riemannian metric on a Lie group $G$ is called *left-invariant* iff the left-translation map is an isometry:

$$\langle u, v \rangle_x = \langle (dL_a)_x u, (dL_a)_x v \rangle_{ax}, \ \forall a, x \in G, \ \forall u, v \in T_x G \tag{62}$$

**Right-invariant** metrics are defined analogously. A *bi-invariant metric* on $G$ is a Riemannian metric that is both left and right invariant. To specify an invariant metric we can use the following.

---

[16](Gallier & Quaintance, 2020, Propositions 16.4 & 16.5).
[17](Gallier & Quaintance, 2020, Definitions 16.8-16.10).

**Proposition B.8.** [18] *For $G$ a Lie group with Lie algebra $\mathfrak{g}$ there is a one-to-one correspondence between inner products on $\mathfrak{g}$ and left (right) invariant metrics on $G$.*

Left (right) invariant metrics can therefore be determined uniquely by specifying an inner product on $\mathfrak{g}$. This can be seen since for any $x \in G$ and $u, v \in T_x G$:

$$\langle u, v \rangle_x = \langle \mathrm{d}(L_{x^{-1}})_x u, \mathrm{d}(L_{x^{-1}})_x v \rangle_e \tag{63}$$

The analogue result for bi-invariant metrics is the following.

**Proposition B.9.** [19] *There is a one-to-one correspondence between* $\mathrm{Ad}$*-invariant inner products on $\mathfrak{g}$ and bi-invariant metrics on $G$. An* $\mathrm{Ad}$*-invariant inner product on $\mathfrak{g}$ is defined such that for any $g \in G$,* $\mathrm{Ad}_g$ *is a linear isometry:*

$$\langle u, v \rangle = \langle \mathrm{Ad}_g(u), \mathrm{Ad}_g(v) \rangle, \ \forall g \in G, \ \forall u, v \in \mathfrak{g} \tag{64}$$

Lie groups with bi-invariant metrics are convenient to work with as the group and Riemannian exponential maps coincide at the identity.

**Proposition B.10.** [20] *If a Lie group $G$ is compact, then it has a bi-invariant metric. If $G$ has a bi-invariant metric, then the Riemannian exponential at the identity* $\exp_e : T_e G \to G$ *coincides with the Lie group exponential* $\mathrm{expm} : \mathfrak{g} \to G$.

The Lie groups of interest $\mathrm{SL}(n, \mathbb{R})$, $\mathrm{GL}(n, \mathbb{R})$ or $\mathrm{SE}(n, \mathbb{R})$ do not admit bi-invariant Riemannian metrics[21]. When only a left (right) invariant metric is available, it is still possible in some cases to obtain closed-form expressions for geodesics such as the Riemannian exponential. Suppose the group $\mathrm{GL}(n, \mathbb{R})$ is endowed with the canonical left-invariant metric determined by the inner product $\langle X, Y \rangle := \mathrm{tr}(X^T Y)$ for $X, Y \in \mathfrak{gl}(n, \mathbb{R})$. Then for any $A \in \mathrm{GL}(n, \mathbb{R})$:

$$g_A(X_1, X_2) = g_I(A^{-1} X_1, A^{-1} X_2) = \langle A^{-1} X_1, A^{-1} X_2 \rangle, \quad \forall X_1, X_2 \in T_A \mathrm{GL}(n, \mathbb{R}) \tag{65}$$

A closed-form expression for the Riemannian exponential map at the identity is given by[22]:

$$\exp_I(X) = e^{X^T} e^{X - X^T}, \quad \forall X \in \mathfrak{gl}(n, \mathbb{R}) \tag{66}$$

On the right-hand side we have used the Lie group exponential, given by the matrix exponential. The same expression holds for $\mathrm{SL}(n, \mathbb{R})$[23] and can be used to define the exponential at any point.

**Remark B.11.** *If we equip* $\mathrm{GL}^+(n, \mathbb{R})$ *or* $\mathrm{SL}(n, \mathbb{R})$ *with their canonical left-invariant metric, the Riemannian exponential is available in closed form given by (66). As opposed to the Lie group exponential, the Riemannian exponential is surjective, and one could see it as a possible choice for the parametrization map* $\xi : \mathfrak{g} \to G$.

*However, as explained in Section 4, if one cannot work only at the level of the Lie algebra, and group elements have to be mapped from the group $G$ to $\mathfrak{g}$, the map $\xi^{-1}$ would also need to be available. We are not aware of a closed-form expression for the Riemannian logarithm on the groups* $\mathrm{GL}^+(n, \mathbb{R})$, $\mathrm{SL}(n, \mathbb{R})$, *corresponding to the canonical left-invariant metric.*

*One could employ a* shooting *or* relaxation *method to compute the Riemannian logarithm[24], as done for example in (Rentmeesters et al., 2013, Chapter 6.2). However, since $\xi^{-1}$ is used at every (lifting) cross-correlation layer, this would add a large computational cost during the forward pass. This issue motivated the search for an alternative solution, as the one proposed in the main text.*

---

[18](Gallier & Quaintance, 2020, Propositions 21.1 & 21.2)

[19](Gallier & Quaintance, 2020, Proposition 21.3).

[20](Gallier & Quaintance, 2020, Propositions 21.6 & 21.20).

[21]A connected Lie group $G$ has a bi-invariant metric iff it is isomorphic to the Cartesian product of a compact group and an additive vector group $(\mathbb{R}^n, +)$ for $n \geq 0$ ((Gallier & Quaintance, 2020, Theorem 21.9)).

[22]See Andruchow et al. (2014); Martin & Neff (2016). Note that in this case, the metric will be left invariant and right-$\mathrm{O}(n)$-invariant.

[23]See (Zacur et al., 2014, Section 8.9) which references Wang (1969); Helgason (1979).

[24]See Zacur et al. (2014, Section 9) for a general discussion on obtaining the Riemannian logarithm in the context of matrix Lie groups.

### B.3 GROUP ACTIONS & HOMOGENEOUS SPACES

Let $G$ be a group and $M$ a set. The left action of $G$ on $M$ is a map $\lambda : G \times M \to M$ satisfying for any $m \in M$ and group elements $h, g \in G$:

$$\lambda(e, m) = m \text{ and } \lambda(h, \lambda(g, m)) = \lambda(hg, m) \tag{67}$$

Right actions are defined analogously. Where it is clear we are referring to a left action we use the notation $g \cdot m$ or $gm$ for $\lambda(g, m)$. Using the action we can define the map $\lambda_g : M \to M$ given by $\lambda_g : x \mapsto g \cdot x$. Since Lie groups are locally compact topological groups some results will be useful if stated more generally. If $G$ is a topological group and $M$ a topological space, then $\lambda$ is continuous and $\lambda_g$ will be a homeomorphism. Whereas when $G$ is a Lie group and $M$ a smooth manifold the action is smooth and $\lambda_g$ will be a diffeomorphism. The action of $G$ on $M$ is **transitive** if:

$$\forall x, y \in M : \exists g \in G : g \cdot x = y \tag{68}$$

If a group $G$ acts transitively on a set $M$, then $M$ is called a homogeneous space of $G$. For any point $x \in M$, the set of group elements that fix $x$ form a subgroup of $G$ called the **isotropy group** or stabilizer of $x$, and denoted by $G_x$. The **orbit** of a point $x \in M$ is denoted by $O_x$:

$$G_x = \{g \in G \mid g \cdot x = x\} \tag{69}$$

$$O_x = G \cdot x = \{g \cdot x \mid g \in G\} \subseteq M \tag{70}$$

Let $G$ also act on a set $N$. A map $f : M \to N$ is **equivariant** if it commutes with the action of $G$:

$$f(g \cdot m) = g \cdot f(m), \quad \forall m \in M, \forall g \in G \tag{71}$$

**Proposition B.12.** [25] *Let $\lambda : G \times M \to M$ be a transitive left action of a group $G$ on a set $M$, and denote by $H = G_x$ the stabilizer of $x \in M$. The map by $\pi : G \to G/H$ denotes the canonical projection $\pi : g \mapsto gH$ on the left cosets for any $g \in G$. For any such $x \in M$ the projection (or orbit) map $\pi_x : G \to M$ is a surjective map defined by:*

$$\pi_x : G \to M, \ \pi_x : g \mapsto \lambda(g, x) = g \cdot x \tag{72}$$

*Since $\pi_x$ is surjective and we have $\pi_x(gH) = gH \cdot x = g \cdot Hx = g \cdot x = \pi_x(g)$ for any $g \in G$, it induces a bijection $\phi_x : G/H \to M$ by passing to the quotient:*

$$\pi_x = \phi_x \circ \pi, \quad \phi_x : \pi(g) \mapsto g \cdot x \tag{73}$$

**Theorem B.13.** [26] *Let $G$ be a locally compact Hausdorff group that is also $\sigma$-compact. Suppose $G$ acts transitively and continuously on a locally compact Hausdorff space $M$. For any $x \in M$, the stabilizer $G_x$ is a closed subgroup of $G$ and the quotient space $G/G_x$ is Hausdorff. The projection $\pi : G \to G/H$ is a continuous open map, and the orbit map $\pi_x : G \to M$ is also continuous. Furthermore, the map $\phi_x : G/G_x \to M$ is a homeomorphism, and it is $G$-equivariant with the action of $G$ on $G/H$ defined as in (74)[27].*

If $M$ and $N$ are smooth manifolds, $\pi : M \to N$ a smooth map, and $d\pi_p : T_p M \to T_{\pi(p)} N$ its differential at $p \in M$. $\pi$ is a **smooth submersion** if $d\pi_p$ is surjective for every $p \in M$. The subset $\pi^{-1}(x)$ for any $x \in N$ is referred to as the **fiber (of $\pi$) over** $x$, and it is a properly embedded submanifold[28]. The 'analogue' of Theorem B.13 in the Lie group setting are the following results.

**Theorem B.14.** [29] *Suppose $H$ is a closed Lie subgroup of a Lie group $G$. There exists a unique smooth structure on the set of left cosets $G/H$ so that the canonical projection $\pi : G \to G/H$ is a smooth submersion. Furthermore, the left action of $G$ on the cosets:*

$$\tau : G \times G/H \to G/H, \ (g_1, g_2 H) \mapsto g_1 g_2 H \tag{74}$$

*is transitive and smooth, i.e. $G/H$ is a homogeneous $G$-space.*

---

[25]See for example (Gallier & Quaintance, 2020, Proposition 5.2).

[26](Gallier & Quaintance, 2020, Proposition 5.7 & Theorems 5.13-5.14).

[27](Abbaspour & Moskowitz, 2007, Theorem 0.4.5).

[28](Lee, 2013, Corollaries 5.13 & 5.14).

[29](Lee, 2013, Theorem 21.17).

$G/H$ is also referred to as a *coset manifold*. Note that $\pi : G \to G/H$ is $G$-equivariant, and we have a diffeomorphism $\tau_h : G/H \to G/H$, $\tau_h : gH \mapsto hgH$ such that:

$$\tau_g \circ \pi = \pi \circ L_g, \quad \tau_{gh} = \tau_g \circ \tau_h, \quad \forall g, h \in G \tag{75}$$

**Theorem B.15.** [30] *Let $\lambda : G \times M \to M$ be a smooth transitive left action of a Lie group $G$ on a smooth manifold $M$. For any $x \in M$, let $G_x = H$ denote its stabilizer. The stabilizer $H$ is a closed Lie subgroup of $G$. $\phi_x : G/H \to M$ defined as in (73):*

$$\phi_x : G/H \to M, \ gH \mapsto \lambda(g, x) = g \cdot x \tag{76}$$

*is a diffeomorphism. Furthermore, $\phi_x$ is equivariant with respect to the action of $G$ on $G/H$ and the action of $G$ on $M$. The projection map $\pi_x : G \to M$, $g \mapsto \lambda(g, x) = g \cdot x$ (which can be expressed as $\pi_x = \phi_x \circ \pi$) is a smooth submersion.*

In the main text, we employ the decomposition of a group $G$ as $G/H \times H$ for $H \leq G$ a closed subgroup. The following sections describe the specific class of homogeneous spaces $G/H$ for which these decompositions are realised. Our starting point is the following general result.

**Proposition B.16.** [31] *Let $G$ be a Lie group, $H$ a closed Lie subgroup, and denote $M = G/H$. If the projection $\pi : G \to M$ has a smooth cross section $\sigma : M \to G$ ($\pi \circ \sigma = \mathrm{id}_M$) then:*

$$\varphi : M \times H \to G, \ \varphi(m, h) = \sigma(m)h \tag{77}$$

*defines a diffeomorphism from the product space $M \times H$ onto $G$.*

*Proof.* The proof is given in (O'Neill, 1983, Lemma 11.16). Here we give a more verbose description of the construction since the inverse of this map is mentioned in the following sections. Let $H = G_x$ be the stabilizer of a point $x \in M$, where $M = G/H$. To show that $\varphi : M \times H \to G$ is a diffeomorphism we define the inverse map $\psi : G \to M \times H$ such that:

$$\psi : g \mapsto (\pi(g), (\sigma(\pi(g)))^{-1}g), \quad \forall g \in G \tag{78}$$

$\psi$ is smooth as it is a composition of smooth maps. To show that $\psi$ is well-defined one shows $\sigma(\pi(g))^{-1}g \in H = G_x$. Recall from Proposition B.12 the projection $\pi_x(g) = g \cdot x$ for any $g \in G$ and that we've assumed $\pi \circ \sigma = \mathrm{id}_M$:

$$\sigma(\pi(g)) \cdot x = \pi_x(\sigma(\pi(g))) = \pi_x(g) = g \cdot x \tag{79}$$

Then $(\sigma(\pi(g)))^{-1}g \cdot x = (\sigma(\pi(g)))^{-1}\sigma(\pi(g)) \cdot x = x$, such that $\sigma(\pi(g))^{-1}g \in H = G_x$. We therefore have $\psi(g) \in M \times H$, and it is clear that $\varphi \circ \psi = \mathrm{id}_G$ and $\psi \circ \varphi = \mathrm{id}_{M \times H}$, the result follows. $\square$

Lemma 11.27 of O'Neill (1983) gives a method for constructing such a map for a class of homogeneous spaces $M = G/H$ called *naturally reductive*. Before reviewing these spaces, we need to define *Riemannian submersions*. These class of submersions will allow us to describe the geometry of $G/H$ using the geometry of $G$. For a comprehensive description one can consult O'Neill (1983, Chapter 7) or (Gallier & Quaintance, 2020, Section 18.3), which serve as our main references.

Suppose $M$ and $N$ are smooth manifolds, and $\pi : M \to N$ a submersion. For any $x \in \pi(M)$, the fiber above $x$ given by $\pi^{-1}(x)$ is a submanifold of $M$. For any $p \in \pi^{-1}(x)$ then $T_p\pi^{-1}(x) = \ker d\pi_p$. Any complement of $\ker d\pi_p = T_p\pi^{-1}(x)$ in $T_pM$ will be isomorphic to $T_{\pi(p)}N$. In the Riemannian case for $(M, g)$ and $(N, h)$ smooth manifolds endowed with metrics, the fibers $\pi^{-1}(x)$ will be Riemannian submanifolds of $M$, and we can define an orthogonal decomposition with respect to the metric. The orthogonal subspaces are referred to as horizontal and vertical subspaces. More precisely, for each $x \in \pi(M) \subseteq N$ and $p \in \pi^{-1}(x)$, the tangent space $T_pM$ can be decomposed into orthogonal subspaces $T_pM = \ker d\pi_p \oplus (\ker d\pi_p)^{\perp} = V_p \oplus H_p$. Tangent vectors $v \in T_pM$, can be written uniquely using horizontal and vertical components:

$$v = v_H + v_V, \quad v_H \in H_p, v_V \in V_p \tag{80}$$

---

[30]See (Lee, 2013, Theorem 21.18) and (O'Neill, 1983, Proposition 11.13).

[31]The same result is obtained in the case of topological groups when $\sigma : G \to M$ is a continuous cross section of $\pi$, in which case $\varphi : M \times H \to G$ is then a homeomorphism.

If $v \in H_p$ ($V_p$), then $v$ is called a **horizontal (vertical) tangent vector**. The differential $d\pi_p$ of a submersion being surjective for any $p \in M$ allows us to construct a vector space isomorphism $d\pi_p|_{H_p} : H_p \to T_{\pi(p)}N$ between horizontal spaces $H_p$ and $T_{\pi(p)}N$. $\pi$ is a **Riemannian submersion** if for all $p \in M$, the differential $d\pi_p$ restricted to $H_p$ is a linear isometry onto $T_{\pi(p)}N$:

$$g_p(u, v) = h_{\pi(p)}(d\pi_p(u), d\pi_p(v)), \quad \forall u, v \in H_p \tag{81}$$

The main utility of Riemannian submersions in our case comes from the next theorem which describes how to express geodesics in $N$ as projections of horizontal geodesics in $M$.

**Theorem B.17.** [32] *Let $\pi : M \to N$ be a Riemannian submersion between Riemannian manifolds $(M, g)$ and $(N, h)$ equipped with the Levi-Civita connection. If $\overline{\gamma} : I \to M$ is a geodesic that starts horizontally, i.e. $\overline{\gamma}'(0)$ is a horizontal vector, then $\overline{\gamma}$ is a **horizontal geodesic** ($\overline{\gamma}'(t)$ is horizontal for all $t \in I$). Furthermore, the projection $\pi \circ \overline{\gamma} = \gamma$ is a geodesic in $N$ of the same length as $\overline{\gamma}$. Conversely, for any $p \in M$, if $\gamma$ is a geodesic in $N$ with $\gamma(0) = \pi(p)$, there exists a unique local horizontal lift $\overline{\gamma}$ of $\gamma$ such that $\overline{\gamma}(0) = p$ and $\overline{\gamma}$ is a geodesic in $M$.*

Theorem B.17 states that if a Riemannian submersion $\pi : M \to N$ is available, horizontal geodesics in $M$ are mapped to geodesics in $N$. As $d\pi$ is an isomorphism when restricted to horizontal spaces, if we have a smooth cross-section $\sigma : N \to M$ we can express the Riemannian exponential on $N$ using the Riemannian exponential on $M$:

$$\exp_x(v) = \pi \circ \exp_{\sigma(x)}(\overline{v}), \quad \forall x \in N, v \in T_x N \tag{82}$$

### B.3.1 Naturally reductive & Symmetric spaces

**Definition B.18.** [33] *Let $G$ be a Lie group, $H \le G$ a closed subgroup $\mathrm{Ad} : G \to \mathrm{GL}(\mathfrak{g})$ be the adjoint representation of $G$. A homogeneous space $G/H$ is **reductive** if there is a subspace $\mathfrak{m}$ of $\mathfrak{g}$ where:*

$$\mathfrak{g} = \mathfrak{h} \oplus \mathfrak{m}, \ and \ \mathrm{Ad}_h(\mathfrak{m}) \subseteq \mathfrak{m}, \ \forall h \in H \tag{83}$$

*That is, $G/H$ is reductive if we can find an $\mathrm{Ad}(H)$-invariant subspace $\mathfrak{m}$ complementary to $\mathfrak{h}$ in $\mathfrak{g}$.*

The following property gives a recipe for constructing a $G$-invariant metric on $G/H$ and extending it to a left-invariant metric on $G$ that is right $H$-invariant such that $\pi : G \to G/H$ is a Riemannian submersion, with $\mathfrak{h}$ and $\mathfrak{m}$ being the vertical and horizontal subspaces at $e \in G$.

**Proposition B.19.** [34] *Let $G$ be a Lie group, $H$ a closed subgroup and $G/H$ a reductive homogeneous space with reductive decomposition $\mathfrak{g} = \mathfrak{h} \oplus \mathfrak{m}$.*

1. *There is a one-to-one correspondence between $G$-invariant metrics on $G/H$ and $\mathrm{Ad}(H)$-invariant inner products on $\mathfrak{m}$. The correspondence can be established by making $d\pi_e|_\mathfrak{m} : \mathfrak{m} \to T_o(G/H)$ into a linear isometry, where $o = \pi(e) = eH$. A $G$-invariant metric on $G/H$ exists iff the closure of $\mathrm{Ad}(H)(\mathfrak{m})$ is compact. If $H$ is compact then $\mathrm{Ad}(H)(\mathfrak{m})$ is compact, so there exists a $G$-invariant metric on $G/H$.*

2. *Let $\mathfrak{m}$ have an $\mathrm{Ad}(H)$-invariant inner product. If we extend it to an inner product on $\mathfrak{g} = \mathfrak{h} \oplus \mathfrak{m}$ such that $\mathfrak{h}^\perp = \mathfrak{m}$, and endow $G$ with the corresponding left-invariant metric then the canonical map $\pi : G \to G/H$ is a Riemannian submersion.*

The reductive homogeneous spaces of interest are the following.

**Definition B.20.** [35] *Let $G$ be a Lie group and $H$ a closed subgroup of $G$. The homogeneous space $G/H$ is **naturally reductive** if it is reductive with decomposition $\mathfrak{g} = \mathfrak{h} \oplus \mathfrak{m}$, has a $G$-invariant metric and satisfies:*

$$\langle [X, Z]_\mathfrak{m}, Y \rangle = \langle X, [Z, Y]_\mathfrak{m} \rangle, \ \forall X, Y, Z \in \mathfrak{m} \tag{84}$$

---

[32]See (Gallier & Quaintance, 2020, Proposition 18.8) or (O'Neill, 1983, Lemma 7.45 & Corollary 7.46).

[33](Gallier & Quaintance, 2020, Definition 23.8).

[34]Corresponds to (Gallier & Quaintance, 2020, Propositions 23.23).

[35](Gallier & Quaintance, 2020, Definition 23.9).

In this case, it is possible to express geodesics in $G/H$ with respect to the Levi-Civita connection as orbits of one-parameter subgroups generated by the tangent vectors in $\mathfrak{m}$.

**Proposition B.21.** [36] *Suppose $G/H$ is a naturally reductive homogeneous space, and we have $\mathfrak{g} = \mathfrak{h} \oplus \mathfrak{m}$. Using the $G$-invariant metric of $G/H$, a left-invariant metric is constructed on $G$, such that its restriction to on $\mathfrak{m}$ is $\mathrm{Ad}(H)$-invariant and we have $\mathfrak{m} = \mathfrak{h}^\perp$, and recall that in this case $\pi : G \to G/H$ is a Riemannian submersion. For every $X \in \mathfrak{m}$ the geodesic starting at $o = \pi(e) = eH$ with initial velocity $d\pi_e(X)$ is given by:*

$$\gamma_{o,d\pi_e(X)}(t) = \mathrm{expm}(tX) \cdot o = \pi \circ \mathrm{expm}(tX), \quad \forall t \in \mathbb{R} \tag{85}$$

Since the one-parameter subgroups $t \mapsto \mathrm{expm}(tX)$ are defined for any $t \in \mathbb{R}$, by the preceding proposition so are maximal geodesics through $o$ and therefore through any point since we are working with a homogeneous space. Naturally reductive homogeneous spaces are therefore complete[37].

**Definition B.22.** *A connected Riemannian manifold $(M, g)$ is a **(Riemannian) symmetric space** if for every point $p \in M$ there exists a unique isometry $s_p : M \to M$ such that $s_p(p) = p$ and $(ds_p)_p = -\mathrm{id}_p$. Equivalently, for every $p \in M$, the map $s_p$ is an involutive isometry ($s_p^2 = \mathrm{id}$) having $p$ as its only fixed point.*

The isometry $s_p : M \to M$ is called the *global symmetry* of $M$ at the $p$. Symmetric spaces can be constructed from 'Lie group data'. The connection can be made clear with a few more definitions.

An **involutive automorphism** of a Lie group $G$ is an automorphism $\sigma : G \to G$ such that $\sigma \neq \mathrm{id}$ and $\sigma^2 = \mathrm{id}$. For $\sigma$ an involutive automorphism $G$, $G^\sigma = \{g \in G \mid \sigma(g) = g\}$ will denote the closed subgroup of fixed points of $\sigma$ and $G_0^\sigma$ its identity component.

**Definition B.23.** [38] *A **symmetric pair** is a triplet $(G, H, \sigma)$ where $G$ is a connected Lie group, $H$ a closed Lie subgroup of $G$, and $\sigma : G \to G$ an involutive automorphism of $G$ such that $G_0^\sigma \subseteq H \subseteq G^\sigma$. If additionally $\mathrm{Ad}(H) \subseteq \mathrm{GL}(\mathfrak{g})$ is compact (where $\mathrm{Ad} : G \to \mathrm{GL}(\mathfrak{g})$ is the adjoint representation of $G$), then $(G, H, \sigma)$ is a **Riemannian symmetric pair**.*

The differential $d\sigma_e : \mathfrak{g} \to \mathfrak{g}$ of an involutive automorphism $\sigma : G \to G$ defines the $\pm 1$ eigenspaces:

$$\mathfrak{h} = \{X \in \mathfrak{g} \mid d\sigma_e(X) = X\}, \quad \mathfrak{m} = \{X \in \mathfrak{g} \mid d\sigma_e(X) = -X\} \tag{86}$$

**Theorem B.24.** [39] *Suppose that $(G, H, \sigma)$ is a symmetric pair. Then the following properties hold. Note that items 1-3 make $G/H$ into a reductive homogeneous space.*

1. *$\mathfrak{h} = \{X \in \mathfrak{g} \mid d\sigma_e(X) = X\}$ is the Lie algebra of $H$.*

2. *$\mathfrak{g} = \mathfrak{h} \oplus \mathfrak{m}$, where $\mathfrak{m} = \{X \in \mathfrak{g} \mid d\sigma_e(X) = -X\}$. The decomposition follows from $d\sigma_e : \mathfrak{g} \to \mathfrak{g}$ also being an involution $d\sigma_e^2 = \mathrm{id}$ and the identity:*

$$X = \frac{1}{2}(X + d\sigma_e(X)) + \frac{1}{2}(X - d\sigma_e(X)), \quad \forall X \in \mathfrak{g} \tag{87}$$

3. *$\mathrm{Ad}_k(\mathfrak{m}) \subseteq \mathfrak{m}, \forall k \in K$.*

4. *$[\mathfrak{h}, \mathfrak{h}] \subseteq \mathfrak{h}, [\mathfrak{h}, \mathfrak{m}] \subseteq \mathfrak{m}, [\mathfrak{m}, \mathfrak{m}] \subseteq \mathfrak{h}$.*

The map $d\sigma_e : \mathfrak{g} \to \mathfrak{g}$ associated to a symmetric pair $(G, H, \sigma)$ is referred to as a **Cartan involution**, with the automorphism $\sigma : G \to G$ being a *global* **Cartan involution**. The decomposition $\mathfrak{g} = \mathfrak{h} \oplus \mathfrak{m}$ given by $d\pi_e$ as in Thm. B.24 is called a **Cartan Decomposition** of $\mathfrak{g}$. If one further assumes that $G_0^\sigma$ and $H$ are compact, then we obtain a symmetric space.

**Theorem B.25.** [40] *Suppose that $(G, H, \sigma)$ is a Riemannian symmetric pair with $G_0^\sigma$ and $H$ compact. Denote by $\mathfrak{g}$ and $\mathfrak{h}$ the Lie algebras of $G$ and $H$ respectively.*

---

[36](Gallier & Quaintance, 2020, Propositions 23.25-23.27).

[37](O'Neill, 1983, pp. 313).

[38](Helgason, 2001, Chapter IV, §3).

[39](Gallier & Quaintance, 2020, Theorem 23.33).

[40]See (Ziller, 2010, Proposition 6.25) or (Helgason, 2001, Proposition 3.4). Corresponds to (Gallier & Quaintance, 2020, Theorem 23.34), (O'Neill, 1983, Theorem 11.29).

1. *Since $H$ is compact, $G/H$ admits a $G$-invariant metric from Proposition B.19 (1). From the previous theorem, $G/H$ has a reductive decomposition $\mathfrak{g} = \mathfrak{h} \oplus \mathfrak{m}$ where $\mathfrak{h}$ and $\mathfrak{m}$ are the $\pm 1$ eigenspaces of $d\sigma_e$. Using the identity $[\mathfrak{m}, \mathfrak{m}] \subseteq \mathfrak{h}$ and assuming a $G$-invariant metric on $G/H$ the natural reductivity condition of B.20 holds trivially (since $\mathfrak{h} \cap \mathfrak{m} = \{0\}$).*

2. *For every $p \in G/H$, there exists a isometry $s_p : G/H \to G/H$ such that $s_p(p) = p$ and $d(s_p)_p = -\mathrm{id}_p$, making $G/H$ a Riemannian symmetric space. For the projection $\pi : G \to G/H$ and $o = \pi(e) = eH$, the symmetry at $o$ is defined such that $s_o : gH \mapsto \sigma(g)H$:*

$$s_o \circ \pi = \pi \circ \sigma \tag{88}$$

*For an arbitrary $p = gH \in G/H$, the geodesic symmetry is given by:*

$$s_p = \tau_g \circ s_o \circ \tau_{g^{-1}} \tag{89}$$

By the preceding theorem symmetric spaces can be given a naturally reductive structure. We can now use Proposition B.16 to construct a global cross section, under which the Lie group $G$ can be identified with the product space $\mathfrak{m} \times H$ or $\mathrm{expm}(\mathfrak{m}) \times H$. Recall from Proposition B.21 that geodesics starting at $o = \pi(e) = eH = H$ with initial velocity $d\pi_e(X)$ for $X \in \mathfrak{m}$ are of the form $\gamma_{o, d\pi_e(X)}(t) = \mathrm{expm}(tX) \cdot o = \pi(\mathrm{expm}(tX))$. In particular, we can obtain the following expression for the Riemannian exponential $\exp_o : T_o M \to M$:

$$\exp_o(d\pi_e|_{\mathfrak{m}}(X)) = \pi(\mathrm{expm}(X)), \quad \forall X \in \mathfrak{m} \tag{90}$$

That is, the following diagram commutes:

$$\begin{array}{ccc} \mathfrak{m} & \xrightarrow{d\pi_e|_{\mathfrak{m}}} & T_o M \\ {\scriptstyle \mathrm{expm}} \downarrow & & \downarrow {\scriptstyle \exp_o} \\ G & \xrightarrow{\pi} & M \end{array} \tag{91}$$

**Proposition B.26.** *Let $M = G/H$ be a naturally reductive homogeneous space and $\pi : G \to G/H$ the canonical projection. If the Riemannian exponential $\exp_o$ at the point $o = \pi(e) = eH \in M$ is a diffeomorphism, we can construct a diffeomorphism of $\mathfrak{m} \times H$ onto $G$ given by:*

$$\Phi : \mathfrak{m} \times H \to G, \quad (X, h) \mapsto \mathrm{expm}(X)h \tag{92}$$

*Proof.* O'Neill (1983, Lemma 11.27). Again, we reproduce the proof as the maps defined are referenced in later sections. The map is built by constructing a cross-section of $\pi : G \to G/H$ using the relation (90) of the Riemannian exponential such that one first defines:

$$\mathrm{Exp}_e := \exp_o \circ d\pi_e|_{\mathfrak{m}} = \pi \circ \mathrm{expm} : \mathfrak{m} \to M \tag{93}$$

By hypothesis $\exp_o : T_o(M) \to M$ is a diffeomorphism, and so is $d\pi_e|_{\mathfrak{m}}$ making $\mathrm{Exp}_e$ a diffeomorphism. We can define the cross-section by $\sigma : M \to G$ by:

$$\sigma := \mathrm{expm} \circ \mathrm{Exp}_e^{-1} \tag{94}$$

Then $\pi \circ \sigma = \pi \circ \mathrm{expm} \circ \mathrm{Exp}_e^{-1} = \mathrm{Exp}_e \circ \mathrm{Exp}_e^{-1} = \mathrm{id}_M$, and by Proposition B.16 we have a diffeomorphism $\varphi : M \times H \to G$ given by (77):

$$\varphi : (m, h) \mapsto \sigma(m)h = \mathrm{expm}(\mathrm{Exp}_e^{-1}(X))h \tag{95}$$

Composing this map with the map $\mathrm{Exp}_e \times \mathrm{id}_H$ we obtain the desired map $\Phi := \varphi \circ (\mathrm{Exp}_e \times \mathrm{id}_H)$:

$$\Phi : \mathfrak{m} \times H \to G, \quad \Phi : (X, h) \mapsto \sigma(\mathrm{Exp}_e(X))h = \mathrm{expm}(X)h \tag{96}$$

$\square$

### B.4 THE CARTAN/POLAR DECOMPOSITION

Define the following subsets of $M_n(\mathbb{R})$:

$$\text{Sym}(n, \mathbb{R}) = \{P \in M_n(\mathbb{R}) \mid P = P^T\} \tag{97}$$

$$\text{Pos}(n, \mathbb{R}) = \{P \in \text{Sym}(n, \mathbb{R}) \mid \forall v \in \mathbb{R}^n, v \neq 0, v^T P v > 0\} \tag{98}$$

$$\text{SPos}(n, \mathbb{R}) = \{P \in \text{Pos}(n, \mathbb{R}) \mid \det(P) = 1\} \tag{99}$$

$$\text{Sym}_0(n, \mathbb{R}) = \{P \in \text{Sym}(n, \mathbb{R}) \mid \text{tr}(P) = 0\} \tag{100}$$

$\text{Sym}(n, \mathbb{R})$ is the vector space of $n \times n$ real symmetric matrices and $\text{Pos}(n, \mathbb{R})$ is the subset of $\text{Sym}(n, \mathbb{R})$ of symmetric positive definite (SPD) matrices. $\text{SPos}(n, \mathbb{R})$ denotes the subset of $\text{Pos}(n, \mathbb{R})$ consisting of SPD matrices with unit determinant, and $\text{Sym}_0(n, \mathbb{R})$ the subspace of $\text{Sym}(n, \mathbb{R})$ of traceless real symmetric matrices. Every SPD matrix $S \in \text{Pos}(n, \mathbb{R})$ has a unique square root[41] $S^{1/2} = P$, $P \in \text{Pos}(n, \mathbb{R})$, which shows the uniqueness of the polar decomposition.

**Proposition B.27** (Polar decomposition). [42] *Any matrix $A \in \text{GL}(n, \mathbb{R})$ can be uniquely decomposed as $A = PR$ or $A = \tilde{R}\tilde{P}$, where $P, \tilde{P} \in \text{Pos}(n, \mathbb{R})$ and $R, \tilde{R} \in O(n)$. We refer to the factorization $A = PR$ as the **left polar decomposition** and to $A = \tilde{R}\tilde{P}$ as the **right polar decomposition**. We choose to work with the left polar decomposition. The factors of this decomposition are uniquely determined and we have a bijection $\text{GL}(n, \mathbb{R}) \to \text{Pos}(n, \mathbb{R}) \times O(n)$ given by:*

$$A \mapsto (\sqrt{AA^T}, \sqrt{AA^T}^{-1} A), \quad \forall A \in \text{GL}(n, \mathbb{R}) \tag{101}$$

As mentioned in the main text, this decomposition can be generalized using the fact that the spaces $\text{Pos}(n, \mathbb{R}) = \text{GL}^+(n, \mathbb{R})/\text{SO}(n)$ and $\text{SPos}(n, \mathbb{R}) = \text{SL}(n, \mathbb{R})/\text{SO}(n)$ are symmetric spaces, and a **Cartan decomposition** is available in this case. We first state some useful properties of $\text{Pos}(n, \mathbb{R})$ and then review its symmetric space and naturally reductive structure.

**Proposition B.28.** [43] *Every real symmetric matrix $X \in \text{Sym}(n, \mathbb{R})$ has a spectral decomposition $X = ODO^T$ where $O \in \text{SO}(n)$ and $D = \text{diag}(d_1, \ldots, d_n)$, $d_i \in \mathbb{R}$ is a diagonal matrix consisting of the eigenvalues of $X$, which are positive iff $X$ is positive-definite. Using this decomposition we have simplified expressions for the matrix exponential $\text{expm} : \text{Sym}(n, \mathbb{R}) \to \text{Pos}(n, \mathbb{R})$ and logarithm $\text{logm} : \text{Pos}(n, \mathbb{R}) \to \text{Sym}(n, \mathbb{R})$:*

$$\text{expm}(X) = O\text{diag}(\exp(d_1), \ldots, \exp(d_n))O^T, \quad \forall X \in \text{Sym}(n, \mathbb{R}) \tag{102}$$

$$\text{logm}(P) = O\text{diag}(\log(d_1), \ldots, \log(d_n))O^T, \quad \forall P \in \text{Pos}(n, \mathbb{R}) \tag{103}$$

$\text{Pos}(n, \mathbb{R})$ *is an open subset of* $\text{Sym}(n, \mathbb{R})$ *and a smooth manifold of dimension* $n(n+1)/2$, *with the tangent space* $T_P\text{Pos}(n, \mathbb{R})$ *at any* $P \in T_P\text{Pos}(n, \mathbb{R})$ *naturally isomorphic (by translation) to* $\text{Sym}(n, \mathbb{R})$. *The matrix exponential and logarithm maps are diffeomorphisms between* $\text{Sym}(n, \mathbb{R})$ *and* $\text{Pos}(n, \mathbb{R})$, *and the power map* $P \mapsto P^\alpha$ *is smooth for any* $\alpha \in \mathbb{R}$, *since it can be expressed as:*

$$P^\alpha = \text{expm}(\alpha\text{logm}(P)), \quad \forall P \in \text{Pos}(n, \mathbb{R}) \tag{104}$$

As a reference for the following results on $\text{Pos}(n, \mathbb{R})$ and $\text{SPos}(n, \mathbb{R})$ see Förstner & Moonen (2003); Pennec (2020); Stegemeyer & Hüper (2021). The presentation here also follows (Rentmeesters et al., 2013, Section 3.5) and (Lezcano-Casado, 2021, Section 3.5.3).

$\text{Pos}(n, \mathbb{R})$ is a homogeneous space of the positive general linear group $\text{GL}^+(n, \mathbb{R})$. More precisely, $\text{GL}^+(n, \mathbb{R})$ has a smooth transitive action on $\text{Pos}(n, \mathbb{R})$ given by:

$$\lambda : \text{GL}^+(n, \mathbb{R}) \times \text{Pos}(n, \mathbb{R}) \to \text{Pos}(n, \mathbb{R}), \quad (A, P) \mapsto APA^T \tag{105}$$

Note that every SPD matrix $P \in \text{Pos}(n, \mathbb{R})$ can be written as $P = AA^T$ for some $A \in \text{GL}^+(n, \mathbb{R})$. The isotropy group of the identity matrix $I \in \text{Pos}(n, \mathbb{R})$ corresponding to this action is the special orthogonal group $\text{SO}(n)$ since $RIR^T = RR^T = I$ for $R \in \text{SO}(n)$.

---

[41](Hall, 2015, Lemma 2.18) or (Horn & Johnson, 2012, Theorem 7.2.6).

[42]See (Jost & Jost, 2008, Lemma 7.5.1) or (Horn & Johnson, 2012, Theorem 7.3.1).

[43](Arsigny et al., 2007, Theorems 2.6, 2.8 & Corollary 2.9) more specifically for the matrix exp/log and power map. For a review of algebraic properties and the Riemannian manifold structure of $\text{Pos}(n, \mathbb{R})$ see Arsigny et al. (2007); Pennec (2020).

Applying Theorems B.14 & B.15 we have that $\mathrm{GL}^+(n, \mathbb{R})/\mathrm{SO}(n) = \mathrm{Pos}(n, \mathbb{R})$. That is, we have a diffeomorphism:

$$\phi_I : \mathrm{GL}^+(n, \mathbb{R})/\mathrm{SO}(n) \to \mathrm{Pos}(n, \mathbb{R}), \quad A \cdot SO(n) \mapsto AA^T \tag{106}$$

And a smooth submersion of $\mathrm{GL}^+(n, \mathbb{R})$ onto $\mathrm{Pos}(n, \mathbb{R})$ given by:

$$\pi_I = \phi_I \circ \pi : \mathrm{GL}^+(n, \mathbb{R}) \to \mathrm{Pos}(n, \mathbb{R}), \quad A \mapsto AA^T \tag{107}$$

Let $\mathfrak{g} = \mathfrak{gl}(n, \mathbb{R})$ denote the Lie algebra of $\mathrm{GL}^+(n, \mathbb{R})$. The Lie algebra of $\mathrm{SO}(n)$ is the space $\mathfrak{so}(n) = \{X \in \mathrm{M}_n(\mathbb{R}) \mid X = -X^T\}$ of skew-symmetric matrices. $\mathrm{Pos}(n, \mathbb{R}) = \mathrm{GL}^+(n, \mathbb{R})/\mathrm{SO}(n)$ is a reductive homogeneous space (see Definition B.18) since $\mathrm{Ad}(\mathrm{SO}(n))(\mathrm{Sym}(n, \mathbb{R})) \subseteq \mathrm{Sym}(n, \mathbb{R})$ and we have the decomposition $\mathfrak{g} = \mathfrak{h} \oplus \mathfrak{m}$ given by:

$$\mathfrak{gl}(n, \mathbb{R}) = \mathfrak{so}(n) \oplus \mathrm{Sym}(n, \mathbb{R}) \tag{108}$$

Then $\mathfrak{h} = \mathfrak{so}(n)$ and $\mathfrak{m} = \mathrm{Sym}(n, \mathbb{R})$, and the bracket relations $[\mathfrak{h}, \mathfrak{h}] \subseteq \mathfrak{h}$, $[\mathfrak{h}, \mathfrak{m}] \subseteq \mathfrak{m}$, $[\mathfrak{m}, \mathfrak{m}] \subseteq \mathfrak{h}$ hold. We now choose an inner product on $\mathfrak{gl}(n, \mathbb{R})$ such that its restriction to $\mathrm{Sym}(n, \mathbb{R})$ is $\mathrm{Ad}(\mathrm{SO}(n))$-invariant, $\mathrm{SO}(n)^\perp = \mathrm{Sym}(n, \mathbb{R})$ and we can use it to define a left-invariant Riemannian metric on $\mathrm{GL}^+(n, \mathbb{R})$. We work with a scaled version of the canonical inner product $\langle X, Y \rangle := \mathrm{tr}(X^T Y)$:

$$B(X, Y) := 4\langle X, Y \rangle = 4\mathrm{tr}(X^T Y), \quad X, Y \in \mathfrak{gl}(n, \mathbb{R}) \tag{109}$$

The inner product respects the decomposition (108) into symmetric and skew-symmetric matrices, and the left-invariant metric on $\mathrm{GL}^+(n, \mathbb{R})$ (which is also right-$\mathrm{SO}(n)$-invariant) is:

$$g_A^{\mathrm{GL}^+(n, \mathbb{R})}(X, Y) = B(A^{-1} X, A^{-1} Y), \quad \forall A \in \mathrm{GL}^+(n, \mathbb{R}), \forall X, Y \in T_A \mathrm{GL}^+(n, \mathbb{R}) \tag{110}$$

To define a $\mathrm{GL}^+(n, \mathbb{R})$-invariant metric on $\mathrm{Pos}(n, \mathbb{R}) = \mathrm{GL}^+(n, \mathbb{R})/\mathrm{SO}(n)$, note that the differential of the projection (107) at $I$ is $d\pi_I(X) = X + X^T$ for any $X \in \mathfrak{gl}(n, \mathbb{R})$, with $\ker(d\pi_I) = \mathfrak{so}(n)$ and its restriction to $\mathfrak{m} = \mathrm{Sym}(n, \mathbb{R})$ gives the isomorphism:

$$d\pi_I : \mathfrak{m} \to T_I \mathrm{Pos}(n, \mathbb{R}), \quad X \mapsto 2X \tag{111}$$

We have $\lambda(P^{1/2}, I) = P$, and the differential with respect to the second argument at identity is $X \mapsto P^{1/2} X P^{1/2}$. The linear isomorphism $d(\pi_I \circ L_{P^{1/2}})_I : \mathfrak{m} \to T_P \mathrm{Pos}(n, \mathbb{R})$ is then given by:

$$d(\pi_I \circ L_{P^{1/2}})_I : X \mapsto 2P^{1/2} X P^{1/2}, \quad \forall X \in \mathfrak{m} \tag{112}$$

We denote its inverse by $\eta_P : T_P \mathrm{Pos}(n, \mathbb{R}) \to \mathfrak{m}$, such that $\eta_P : X \mapsto \frac{1}{2} P^{-1/2} X P^{-1/2}$. The induced (quotient) metric on $\mathrm{Pos}(n, \mathbb{R})$[44] is defined for any $P \in \mathrm{Pos}(n, \mathbb{R})$ and $X, Y \in T_P \mathrm{Pos}(n, \mathbb{R})$:

$$g_P^{\mathrm{Pos}(n, \mathbb{R})}(X, Y) := B(\eta_P(X), \eta_P(Y)) = \langle P^{-1/2} X P^{-1/2}, P^{-1/2} X P^{-1/2} \rangle = \mathrm{tr}(P^{-1} X P^{-1} Y) \tag{113}$$

Endowed with this metric the action of $\mathrm{GL}^+(n, \mathbb{R})$ is by isometries and $\pi_I$ is a Riemannian submersion. $(\mathrm{Pos}(n, \mathbb{R}), g^{\mathrm{Pos}(n, \mathbb{R})})$ is also a Riemannian symmetric space and $(\mathrm{GL}^+(n, \mathbb{R}), \mathrm{SO}(n), \Theta)$ is a Riemannian symmetric pair, with $\Theta$ the global Cartan involution:

$$\Theta : \mathrm{GL}^+(n, \mathbb{R}) \to \mathrm{GL}^+(n, \mathbb{R}), \quad \Theta : A \mapsto (A^T)^{-1} \tag{114}$$

In this case we have $G_0^\Theta = \mathrm{SO}(n) = G^\Theta$, and we have corresponding Lie algebra involution:

$$\theta := d\Theta_e : \mathfrak{gl}(n, \mathbb{R}) \to \mathfrak{gl}(n, \mathbb{R}), \quad \theta : X \mapsto -X^T \tag{115}$$

Analog results hold for $\mathrm{SPos}(n, \mathbb{R}) = \mathrm{SL}(n, \mathbb{R})/\mathrm{SO}(n)$, such that $(\mathrm{SL}(n, \mathbb{R}), \mathrm{SO}(n), \Theta)$ is a Riemannian symmetric pair. The group $\mathrm{SL}(n, \mathbb{R})$ has Lie algebra:

$$\mathfrak{sl}(n, \mathbb{R}) = \{X \in \mathfrak{gl}(n, \mathbb{R}) \mid \mathrm{tr}(X) = 0\} \tag{116}$$

$\mathrm{SL}(n, \mathbb{R})/\mathrm{SO}(n)$ is an example of a non-compact symmetric space[45]. We can reuse the metrics (110) and (113), restricting them to $\mathrm{SL}(n, \mathbb{R})$ and $\mathrm{SPos}(n, \mathbb{R})$, respectively.

---

[44]Known as the affine-invariant metric, see (Thanwerdas & Pennec, 2023, Proposition 3.1).

[45]For a classification of symmetric spaces see for example (Ziller, 2010, Chapter 6).

**Proposition B.29.** [46] $\mathrm{GL}^+(n, \mathbb{R})$ *can be represented as a product* $\mathrm{SL}(n, \mathbb{R}) \times \mathbb{R}_{>0}^\times$ *by the Lie group isomorphism:*

$$\mathrm{GL}^+(n, \mathbb{R}) \to \mathrm{SL}(n, \mathbb{R}) \times \mathbb{R}_{>0}^\times, \quad A \mapsto (\frac{A}{\det(A)^{\frac{1}{n}}}, \det(A)^{\frac{1}{n}}) \tag{117}$$

*Reusing the previously defined metrics,* $\mathrm{SL}(n, \mathbb{R})$ *and* $\mathrm{SPos}(n, \mathbb{R})$ *are totally geodesic submanifolds[47] of* $\mathrm{GL}^+(n, \mathbb{R})$ *and* $\mathrm{Pos}(n, \mathbb{R})$, *respectively. The decomposition (117) restricted to* $\mathrm{Pos}(n, \mathbb{R})$ *can be shown to induce a Riemannian isometry* $\mathrm{Pos}(n, \mathbb{R}) \cong \mathrm{SPos}(n, \mathbb{R}) \times \mathbb{R}_{>0}$. *The tangent space decomposition is* $\mathrm{Sym}(n, \mathbb{R}) = \mathrm{Sym}_0(n, \mathbb{R}) \oplus \mathfrak{d}$, *where* $\mathfrak{d}(n, \mathbb{R})$ *are scalar diagonal matrices.*

## B.5 PROOF OF THEOREM 4.2

As in the main text, we let $(G/H, M, \mathfrak{m})$ define our 'Lie group data', corresponding to $(\mathrm{GL}^+(n, \mathbb{R})/\mathrm{SO}(n), \mathrm{Pos}(n, \mathbb{R}), \mathrm{Sym}(n, \mathbb{R}))$ or $(\mathrm{SL}(n, \mathbb{R})/\mathrm{SO}(n), \mathrm{SPos}(n, \mathbb{R}), \mathrm{Sym}_0(n, \mathbb{R}))$.

**Theorem 4.2.** *Let* $(G/H, M, \mathfrak{m})$ *be as above, and denote by* $\mathfrak{g}$, $\mathfrak{h}$ *the Lie algebras of* $G$ *and* $H$.

1. *The matrix exponential and logarithm are diffeomorphisms between* $\mathfrak{m}$ *and* $M$, *respectively. For any* $P \in M$ *and* $\alpha \in \mathbb{R}$, *the power map* $P \mapsto P^\alpha$ *is smooth and can be expressed as:*
$$P^\alpha = \mathrm{expm}(\alpha \mathrm{logm}(P)), \quad \forall P \in \mathrm{Pos}(n, \mathbb{R}) \tag{19}$$

2. $G \cong M \times H$ *and* $G \cong \mathfrak{m} \times H$. *We have group-level diffeomorphisms:*
$$\chi : M \times H \to G, \quad \chi(P, R) \mapsto PR \tag{20}$$
$$\Phi : \mathfrak{m} \times H \to G, \quad \Phi : (X, R) \mapsto \mathrm{expm}(X)R = e^X R \tag{21}$$

3. *The above maps can be inverted in closed-form:*
$$\chi^{-1} : G \to M \times H, \ \chi^{-1} : A \mapsto (\sqrt{AA^T}, \sqrt{AA^T}^{-1} A) \tag{22}$$
$$\Phi^{-1} : G \to \mathfrak{m} \times H, \quad \Phi^{-1} : A \mapsto (\frac{1}{2}\mathrm{logm}(AA^T), \mathrm{expm}(-\frac{1}{2}\mathrm{logm}(AA^T))A) \tag{23}$$

*Proof.* The theorem is a collection of results related to the Cartan decomposition and the structure theory of Lie groups, which can be found for example in (Bridson & Haefliger, 2013, Chapter II.10) or (Abbaspour & Moskowitz, 2007, Chapter 6). Similar results apply to algebraic subgroups of $\mathrm{GL}(n, \mathbb{R})$ that are closed and stable under transposition (see (Abbaspour & Moskowitz, 2007, Prop. 6.3.3 & Definition 6.3.4) or (Bridson & Haefliger, 2013, Definition 10.56)).

1. The first result holds due to Proposition B.28, and the fact that for any $X \in \mathfrak{m}$ we have $\mathrm{expm}(tX) \in M$ for all $t \in \mathbb{R}$, see (Bridson & Haefliger, 2013, Lemma 10.52). The Riemannian exponential on $M$ is also a diffeomorphism at any point.

2. For the group-level Cartan/Polar decomposition see (Abbaspour & Moskowitz, 2007, Theorem 6.2.5 & 6.3.5). Given a tangent vector in $\mathfrak{m}$, the Riemannian exponential on $M$ and the matrix exponential are related by the diffeomorphism:
$$\mathrm{Exp}_e : \mathfrak{m} \to M, \quad X \mapsto \mathrm{expm}(X) \cdot I = \mathrm{expm}(X) I \mathrm{expm}(X)^T = \mathrm{expm}(2X) \tag{118}$$
$\mathrm{Exp}_e$ is obtained from applying Proposition B.21. The map $\Phi$ of (21) can be obtained from (20) and the fact that the matrix exponential is a diffeomorphism on $M$, or using Proposition B.26, such that (21) corresponds to (96).

3. The map $\chi^{-1}$ is simply the polar decomposition. To obtain $\xi^{-1}$ we use the fact that $AA^T \in M$ for $A \in G$ and the identities:
$$\mathrm{logm}(P^{1/2}) = \frac{1}{2}\mathrm{logm}(P), \ P^{-1/2} = \mathrm{expm}(-\frac{1}{2}\mathrm{logm}(P)), \quad \forall P \in M \tag{119}$$
The identities (119) can be obtained from (19).

$\square$

---

[46] A detailed treatment can be found in Dolcetti & Pertici (2015; 2019).

[47] $N$ is a totally geodesic submanifold of a Riemannian manifold $M$, if for any two points of $N$ and a geodesic $\gamma$ in $M$ that joins them, $\gamma$ is entirely contained in $N$.

## B.6 Integral factorizations for the Cartan/Polar decomposition

Consider again the notation $(G/H, M, \mathfrak{m})$ as in Theorem 4.2. Recall that we have $G = \text{GL}^+(n, \mathbb{R})$ ($M = \text{Pos}(n, \mathbb{R})$) or $G = \text{SL}(n, \mathbb{R})$ ($M = \text{SPos}(n, \mathbb{R})$), with $H = \text{SO}(n)$. From the proof of Thm. 4.2 the cross-section $\sigma = \text{expm} \circ \text{Exp}_e^{-1} : M \to G$, of Prop. B.26 is:

$$\sigma(S) = \text{expm}(\frac{1}{2}\text{logm}(S)), \quad \forall S \in M \tag{120}$$

which is smooth (Prop. B.28) and reduces simply to the square root $\sigma(S) = S^{1/2}$. Since symmetric positive definite matrices have a unique square root, $\sigma : M \to G$ is a diffeomorphism. We obtain a decomposition equivalent to the Polar/Cartan decomposition given by the map $\varphi : M \times H \to G$ defined as in Propositions B.16-B.26:

$$\varphi : M \times H \to G, \quad \varphi : (S, R) \mapsto \sigma(S)R = S^{1/2}R \tag{121}$$

With the inverse defined by:

$$\psi : G \to M \times H, \quad \psi : A \mapsto (\pi_I(A), (\sigma(\pi_I(A)))^{-1}A) = (AA^T, (AA^T)^{-1/2}A) \tag{122}$$

We have equivalent decompositions which allow us to represent $A \in G$ as $A = PR$ or $A = S^{1/2}R$ for $S, P \in M$, $R \in H$ and therefore $P = S^{1/2}$. The motivation behind presenting both decompositions is that for $\text{GL}(n, \mathbb{R})$, the decomposition $A = S^{1/2}R$, has a factorization of the Haar measure $\mu_{\text{GL}(n,\mathbb{R})}$ as a product of invariant measures on $\text{Pos}(n, \mathbb{R})$ and $\text{O}(n)$. The Haar measure on $\text{GL}(n, \mathbb{R})$ is given for any $A = (A_{ij}) \in \text{GL}(n, \mathbb{R})$ by[48]:

$$d\mu_{\text{GL}(n,\mathbb{R})}(A) = |\det(A)|^{-n}dA = |\det(A)|^{-n}\prod_{i,j=1}^{n} dA_{ij} \tag{123}$$

where $dA$ is the Lebesgue measure on $\mathbb{R}^{n^2}$ and $dA_{ij}$ is the Lebesgue measure on $\mathbb{R}$. $\text{GL}(n, \mathbb{R})$ has two homeomorphic connected components consisting of the group of invertible matrices with positive determinant $\text{GL}^+(n, \mathbb{R})$ and with negative determinant $\text{GL}^-(n, \mathbb{R})$[49]. Integrating the full group $\text{GL}^+(n, \mathbb{R})$ can be done by integrating each component separately, and we focus on constructing a solution for the identity component $\text{GL}^+(n, \mathbb{R})$. We use the shorter notation $\text{Pos}(n)$ and $\text{SPos}(n)$ going forward to denote $\text{Pos}(n, \mathbb{R})$ and $\text{SPos}(n, \mathbb{R})$. Using a similar notation scheme as in (123), the unique (up to scaling) $\text{GL}(n, \mathbb{R})$-invariant measure on $\text{Pos}(n)$ is[50]:

$$d\mu_{\text{Pos}(n)}(S) = |\det(S)|^{-(n+1)/2}dS = |\det(S)|^{-(n+1)/2}\prod_{1 \le i \le j \le n}^{n} dS_{ij}, \quad \forall S \in \text{Pos}(n) \tag{124}$$

The following result can be found in a more general setting, often expressed using the 'right' polar coordinates of the decomposition $A = RS^{1/2}$. Let $H = \mathcal{V}_{n,m} = \{R \in \text{M}_{nm}(\mathbb{R}) \mid R^TR = I_m\}$ for $n \ge m$ and $G = \text{M}_{nm}(\mathbb{R})^* = \{A \in \text{M}_{nn}(\mathbb{R}) \mid \text{rank}(A) = m\}$ the set of $n \times m$ matrices of rank $m$. $\mathcal{V}_{n,m}$ is the Stiefel manifold of orthonormal $m$-frames in $\mathbb{R}^n$, on which $\text{O}(n)$ acts transitively by left multiplication such that $\mathcal{V}_{n,m} = \text{O}(n)/\text{O}(n - m)$, with special cases $\mathcal{V}_{n,n} = \text{O}(n)$ and $\mathcal{V}_{n,n-1} = \text{SO}(n)$. The complement of $\text{M}_{nm}(\mathbb{R})^*$ in $\text{M}_{nm}(\mathbb{R})$ has Lebesgue measure zero[51], and $\text{M}_{nn}(\mathbb{R})^* = \text{GL}(n, \mathbb{R})$. In this case it will correspond to (Herz, 1955, Lemma 1.4) or (Muirhead, 2009, Theorem 2.1.14).

**Theorem 4.3.** *Denote $G = \text{GL}(n, \mathbb{R})$, $H = \text{O}(n)$, and let $\mu_G$ be the Haar measure on $G$ and $\mu_H$ the Haar measure on $H$ normalized by $\text{Vol}(H) = 1$. For $A \in G$, under the decomposition $A = S^{1/2}R$, $S \in \text{Pos}(n)$, $R \in H$, the measure on $G$ splits as $d\mu_G(A) = \beta_n d\mu_{\text{Pos}(n)}(S)d\mu_H(R)$, where $\beta_n = \frac{\text{Vol}(\text{O}(n))}{2^n}$ is a normalizing constant. Restricting to $G = \text{GL}^+(n, \mathbb{R})$ and $H = \text{SO}(n)$ and ignoring constants, we have:*

$$f \mapsto \int_G f(A)d\mu_G(A) = \int_{\text{Pos}(n)} \int_H f(S^{1/2}R)d\mu_H(R)d\mu_{\text{Pos}(n)}(S), \ \forall f \in C_c(G) \tag{24}$$

---

[48](Bourbaki & Berberian, 2004, VII, §3, No. 3, Example 1).

[49](Warner, 1983, Theorem 3.68).

[50](Bourbaki & Berberian, 2004, VII, §3, No. 3, Example 8). $\mu_{\text{Pos}(n,\mathbb{R})}$ is bi-$\text{GL}(n, \mathbb{R})$-invariant, as well as invariant under $P \mapsto P^{-1}$.

[51](Gross & Kunze, 1976, Example 5.5) or (Eaton, 1983, Proposition 7.1).

*Proof.* A proof is given in (Gross & Kunze, 1976, Prop. 5.6) for the decomposition of the form $A = RS^{1/2}$. In the form $S^{1/2}R$ it is proven for example in (Faraut & Travaglini, 1987, Section 4). In the context of multivariate statistics see Theorem 5.2.2 and Remark 5.2.3 of Farrell (2012). A recent reference is (Chirikjian, 2012, Section 16.7.2). □

Note that the constant $\beta_n$ is independent of $f \in C_c(G)$. From (Chirikjian, 2012, (16.36)):

$$\text{Vol}(\text{O}(n)) = 2 \cdot \text{Vol}(\text{SO}(n)) = \frac{2^n \pi^{n^2/2}}{\Gamma_n(n/2)} \tag{125}$$

Where $\Gamma_n(\cdot)$ denotes the multivariate Gamma function. From (Chirikjian, 2012, (16.55) & (16.56)), if $dA$ is the Lebesgue measure on $\mathbb{R}^{n^2}$, under the decomposition $A = S^{1/2}R$ we have:

$$dA = d(S^{1/2}R) = \beta_n |\det(S)|^{1/2} dO dS \tag{126}$$

Then considering that $S = AA^T$, the Haar measure $d\mu_{\text{GL}(n,\mathbb{R})} = |\det(A)|^{-n} dA$ can be expressed:

$$d\mu_{\text{GL}(n,\mathbb{R})}(dA) = |S^{1/2}R|^{-n} d(S^{1/2}R) = \beta_n |\det(AA^T)|^{-n/2} |\det(S)|^{1/2} dO dS \tag{127}$$

We use this decomposition treating $G$ ($\text{GL}^+(n,\mathbb{R})$ or $\text{SL}(n,\mathbb{R})$) as our sample space. From Section 5.2 of Farrell (2012), for the case $G = \text{M}_{nm}(\mathbb{R})^* \cong \mathcal{V}_{n,m} \times \text{Pos}(m)$ it can be shown that if a $A \in \text{M}_{nm}(\mathbb{R})^*$ is a random matrix with a $\text{O}(n)$-left invariant distribution, then for $\varphi(A) = RS^{1/2}$ the corresponding random variables $R \in \mathcal{V}_{n,m}$ and $S^{1/2} \in \text{Pos}(m)$ will be independent, and $R$ will have a uniform distribution on $\mathcal{V}_{n,m}$. Furthermore, there exists a relationship between the density function of $A = RS^{1/2} \in \text{M}_{nm}(\mathbb{R})^*$ with respect to the $\text{O}(n)$-invariant measure and that of $S \in \text{Pos}(n)$ with respect to (124). If $G = \text{GL}^+(n,\mathbb{R})$ or $G = \text{SL}(n,\mathbb{R})$, the Haar measure $\mu_G$ is bi-$\text{O}(n)$-invariant (respectively bi-$\text{SO}(n)$-invariant). We can then work with either decomposition [52] $S^{1/2}R$ or $RS^{1/2}$.

**Theorem 4.4.** *If a random matrix $A \in \text{GL}(n,\mathbb{R})$ has a left-$\text{O}(n)$ invariant density function relative to $|AA^T|^{-n/2}dA$, then $(AA^T)^{1/2} = S^{1/2}$ and $R = (AA^T)^{-1/2}A$ are independent random matrices and $R$ has a uniform probability distribution on $\text{O}(n)$. The uniform distribution on $\text{O}(n)$ will be the normalized Haar measure $\mu_{\text{O}(n)}$. Conversely, if $S \in \text{Pos}(n)$ has a density function $f : \text{Pos}(n) \to \mathbb{R}_{\geq 0}$ relative to $\mu_{\text{Pos}(n)}$ and $R \in \text{O}(n)$ is uniformly distributed with respect to the Haar measure $\mu_{\text{O}(n)}$, then $A = S^{1/2}R$ has a density function $\beta_n^{-1} f(AA^T)|\det(A)|^{-n}$ relative to $dA$.*

*Proof.* This theorem collects Lemma 5.2.4 & 5.2.8 of Farrell (2012) applied to the case where we are working with random matrices in $\text{GL}^+(n,\mathbb{R})$ and using the left polar decomposition. See also (Eaton, 1983, Proposition 7.4). □

Restricting only to the connected component $\text{GL}^+(n,\mathbb{R})$, the task is now to specify a probability distribution on $\text{Pos}(n)$ relative to the measure $d\mu_{\text{Pos}(n)}$. For the case of $\text{SL}(n,\mathbb{R})$, using the isomorphism (117), we can define a $\text{SL}(n,\mathbb{R})$-invariant measure on $\text{SPos}(n)$. More precisely, $P = (\det(P)^{1/n}I)\tilde{P}$ for $P \in \text{Pos}(n)$, $\tilde{P} = \det(P)^{-1/n}P \in \text{SPos}(n)$, which we write $P = t^{1/n}\tilde{P}$, $t > 0$, such that $d\mu_{\text{Pos}(n)}(P) = \frac{dt}{t}d\mu_{\text{SPos}(n)}(\tilde{P})$ and for $f \in L^1(\text{Pos}(n))$[53]:

$$\int_{\text{Pos}(n)} f(P) d\mu_{\text{Pos}(n)}(P) = \int_{t>0} \int_{\text{SPos}(n)} f(t^{1/n}\tilde{P}) \frac{dt}{t} d\mu_{\text{SPos}(n)}(\tilde{P}) \tag{128}$$

### B.6.1 SAMPLING ON THE SPD MANIFOLD

Following Said et al. (2017), a Riemannian Gaussian Distribution denoted as $G(\overline{P}, \sigma)$ depends on parameters $\overline{P} \in \text{Pos}(n)$ and $\sigma > 0$ to define a probability density function with respect to the volume element $d\mu_{\text{Pos}(n)}$ by:

$$p(P|\overline{P}, \sigma) = \frac{1}{Z(\sigma)} \exp[-\frac{d^2(P, \overline{P})}{2\sigma^2}], \ P \in \text{Pos}(n) \tag{129}$$

---

[52]In the second case, one considers the right action $A^T P A$, for $A \in \text{GL}(n,\mathbb{R})$ and $P \in \text{Pos}(n)$.

[53](Terras, 2016, (1.21) & (1.39)).

Here, $d : \text{Pos}(n) \times \text{Pos}(n) \to \mathbb{R}_{\geq 0}$ is the Riemannian distance corresponding to the affine-invariant metric (113). The metric plays a key role, as the measure (124) is the Riemannian volume element associated to it. The distance can be expressed by:

$$d^2(X, Y) = \text{tr}[\text{logm}(X^{-1/2} Y X^{-1/2})]^2, \quad \forall X, Y \in \text{Pos}(n) \tag{130}$$

and $Z(\sigma)$ is a normalization factor given in Said et al. (2017) by:

$$\int_{\text{Pos}(n)} \exp[-\frac{d^2(P, \overline{P})}{2\sigma^2}] \mathrm{d}\mu_{\text{Pos}(n)}(P) \tag{131}$$

For $n = 2$ an analytic expression of $Z(\sigma)$ exists, otherwise it can be approximated by Monte Carlo integration. (Said et al., 2017, Prop. 5 & 6) describe an algorithm for sampling from this distribution.

Alternatively, an approximate solution when sampling close to the identity is given by the Log-normal distribution defined in (Schwartzman, 2016, Sec. 4.4). From (Schwartzman, 2016, Def. 4.4.2), $X \in \text{Pos}(n)$ has a Log-normal distribution $X \sim LN(M, \Sigma)$ with mean $M \in \text{Pos}(n)$ and covariance $\Sigma$ if $\text{logm}(M^{-1/2} X M^{-1/2}) \sim N(0, \Sigma)$. This definition assumes that $M$ is the empirical Riemannian center of mass, corresponding to the random variable $X$.

### B.6.2 Alternative decomposition based on the QR factorization

There are several choices available for decomposing $\text{GL}^+(n, \mathbb{R})$ and $\text{SL}(n, \mathbb{R})$ such that invariant integration can be made easier while working with the smaller factors. The primary tools of interest are the Iwasawa and the Cartan decomposition, and one possibility is given by the Gram decomposition (QR factorization). Let $\text{T}(n, \mathbb{R}) = \{X \in \text{GL}(n, \mathbb{R}) \mid X_{ij} = 0 \text{ if } i > j\}$ be the group of real upper triangular matrices and $\text{T}(n, \mathbb{R})_+ \leq \text{T}(n, \mathbb{R})$ its subgroup whose diagonal entries are positive. Every matrix $A \in \text{GL}(n, \mathbb{R})$ has a unique decomposition as $A = RT$ or $A = TR$ for $T \in \text{T}(n, \mathbb{R})_+$ and $R \in \text{O}(n)$.

Under this decomposition, Theorem 4.1 (2) is applicable. The orthogonal factor becomes $R \in \text{SO}(n)$ if restricted to $A \in \text{GL}^+(n, \mathbb{R})$. For $A \in \text{SL}(n, \mathbb{R})$ the decomposition is given by replacing $\text{T}(n, \mathbb{R})_+$ with its subgroup $\text{ST}(n, \mathbb{R})_+ \leq \text{T}(n, \mathbb{R})_+$ of matrices with unit determinant.

### B.7 More details on the Lie algebra parametrization

Any $A \in G$ can be expressed uniquely as $A = e^X R$ for $x \in \mathfrak{m}$ and $R \in H$. Since $H = \text{SO}(n)$ in both cases, the fact that $\text{expm} : \mathfrak{so}(n) \to \text{SO}(n)$ is surjective[54], allows us to write it $A = e^X e^Y$, $Y \in \mathfrak{so}(n)$. The factors $X$ and $R = e^Y$ are obtained using $\Phi^{-1}$ (22). Then by taking the principal branch of the matrix logarithm on $H = \text{SO}(n)$, $Y = \text{logm}(R)$. A map $\xi^{-1} : G \to \mathfrak{g}$ as described in Section 4 is therefore constructed as $\xi^{-1} = (\text{id}_{\mathfrak{m}} \times \text{logm}) \circ \Phi^{-1}$. More precisely, for any $A = e^X e^Y \in G$, using $\xi^{-1}$ we obtain the horizontal/vertical tangent vectors $(Y, X) \in \mathfrak{so}(n) \times \mathfrak{m}$ and since $\mathfrak{g} = \mathfrak{so}(n) \oplus \mathfrak{m}$ we have a unique $Z = X + Y \in \mathfrak{g}$.

If $d$ is the dimension of $G$, the tangent space $\mathfrak{g}$ is a $d$-dimensional vector space isomorphic to $\mathbb{R}^d$, with basis elements denoted by $(E_1, \ldots, E_d)$. Once a basis is chosen we can concretely represent any element of $\mathfrak{g}$ (or $\mathfrak{h}$, $\mathfrak{m}$) as a linear combination of the 'generators' such that $v = \sum_{i=1}^{d} v_i E_i$ for any $v \in \mathfrak{g}$. The *vee* and *hat* functions (denoted $\vee$ and $\wedge$) are used to map tangent vectors to their coordinates in this basis and back:

$$\wedge : \mathbb{R}^d \to \mathfrak{g}, \quad \wedge : \mathbf{v} = (v_1, v_2, \ldots, v_d)^T \mapsto \mathbf{v}^\wedge = \sum_{i=1}^{k} v_i E_i \tag{132}$$

$$\vee : \mathfrak{g} \to \mathbb{R}^d, \quad \vee : \mathbf{v}^\wedge \mapsto (\mathbf{v}^\wedge)^\vee = \mathbf{v} \tag{133}$$

The basis $(E_i)_{i \in [d]}$ is chosen to be orthonormal with respect to the inner product (109) which is used to construct the invariant metric. Going forward it is understood that functions parametrized on the Lie algebra, such as the kernel $\tilde{k}_\theta : \mathfrak{g} \to \mathbb{R}$, take as input the vector of scalar coefficients of the tangent vector expressed in the chosen basis (the result of the $\vee$ map).

---

[54](Gallier & Quaintance, 2020, Theorem 2.6).

To summarize, the map $\xi^{-1} : G \to \mathfrak{g}$ is implemented for any $A \in G$ by[55]:

1. Mapping $A$ to its product space representation in $\mathfrak{m} \times \mathrm{SO}(n)$ using $\Phi^{-1}(A) = (X, R)$.
2. Using the matrix logarithm on $R = e^Y$ (which is available in closed form for the cases of interest $\mathrm{SO}(2)$ and $\mathrm{SO}(3)$) to obtain $(X, \mathrm{logm}(R)) = (X, Y)$.
3. Expressing the tangent vector $Z = X + Y$ using the chosen basis as $Z^\vee \in \mathbb{R}^d$.

## C Architecture & training details

All experiments will use the same ResNet-like architecture He et al. (2016), and it will consist of a lifting cross-correlation layer, a single residual block and a final cross-correlation layer. Finally, to achieve invariance global pooling is applied over the spatial and group dimensions. The (lifting) cross-correlation layers are always followed by normalization and non-linear activation layers. In the case of the affine robustness task, we use GeLU nonlinearities and 'LayerNorm' normalization[56]. The residual block contains 2 group cross-correlation layers and we apply max-pooling over the spatial dimension of the feature maps after each block to increase the robustness of the model. For all experiments, the kernels $k_\theta : \mathfrak{g} \to \mathbb{R}$ are parametrized using 'SIREN networks', introduced in Sitzmann et al. (2020). SIREN networks can be considered as one example of an Implicit Neural Representation (INR) model. These models have seen widespread use in various areas of computer vision and graphics, e.g. Mildenhall et al. (2021). INRs can be formalized as learned continuous function approximators based on MLPs. They can be described simply as MLP layers of the form:

$$\mathbf{y}_m = \sigma\left(W_m \mathbf{y}_{m-1} + \mathbf{b}_m\right) \tag{134}$$

where $\sigma$ is a non-linearity. In case of SIRENs we have $\sigma(x) = \sin(\omega_0 x)$, where $\omega_0 \in \mathbb{R}_{>0}$ is a multiplier controlling the frequency of the sinusoid. We emphasize again that the proposed methodology is not dependent on the specific parametrization of $k_\theta$, and have experimentally found that other activation functions such as the (complex) Gabor wavelet Saragadam et al. (2023) offer comparable results. We set $\omega_0 = 10$ for all experiments. We use 42 output channels in both the lifting and cross-correlation layers. Each SIREN network consists of 2 layers of size 60.

A key hyperparameter to consider is the number of group elements that will be sampled in the Monte Carlo approximation of each of the cross-correlation layers. Empirically, we have found that $10 - 12$ samples are enough to achieve a better performance compared to the previously described models. The models are trained for 100 epochs, with a batch size of 128, and the Adam optimizer of Kingma & Ba (2014) with a standard learning rate of 0.0001. Sampling from $\mathrm{Pos}(2, \mathbb{R})$ is done using the log-Normal distribution of Schwartzman (2016) centered at the identity while for $\mathrm{SO}(2)$ we work with a discretization of equi-distant points in $[0, 2\pi]$.

## D Equivariance error analysis

**Equivariance error** Since our models are only equivariant in expectation, we validate this property numerically by measuring their equivariance error following the same approach as Sosnovik et al. (2020), where we look to quantify:

$$\Delta := \|\mathcal{L}_g[\Phi(f)] - \Phi[\mathcal{L}_g(f)]\|_2^2 / \|\mathcal{L}_g[\Phi(f)]\|_2^2 \tag{135}$$

where $g \in G$ and $\Phi$ represents our network. We evaluate the equivariance error before training the network, i.e. $\Phi$ is a convolutional network with randomly initialized weights. We take $\Phi$ to be a simple convolutional network composed of a lifting map (5), a cross correlation (7) and a projection correlation mapping our data back to a scalar field (which transforms under the trivial representation). The same normalization and nonlinearities described previously are employed. In Figure 1 we plot the equivariance error of $\Phi$, for different choices of $k_\theta$, and compare our model to a standard CNN with the same input-output dimensionality for its layers. For the evaluation we use 100 samples from the Haar measure of $\mathrm{SL}(2, \mathbb{R})$, using the tools described in this thesis, and obtain an average estimate over 10 random seeds.

---

[55] Similar interpolation methods have been explored in the context of numerical linear algebra and computational mathematics. See Munthe-Kaas et al. (2001; 2014) and especially Gawlik & Leok (2018), as this method can in part be seen as an instance of their more general framework.

[56] See Hendrycks & Gimpel (2016) and Ba et al. (2016).

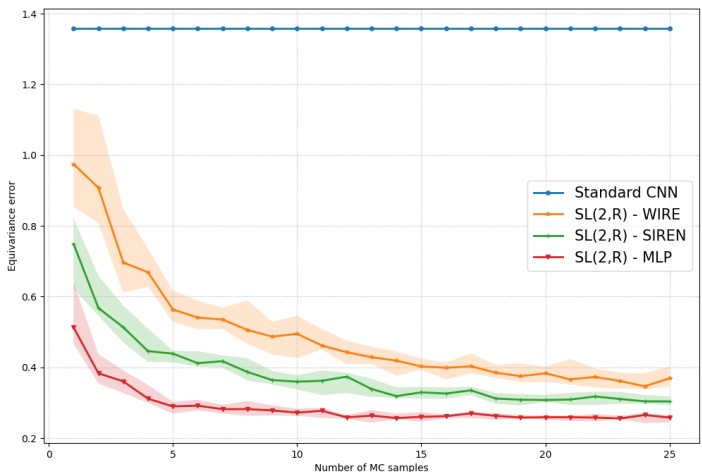

Figure 1: Equivariance error as a function of the number of MC samples.

We compare three possible choices of kernel parametrizations, namely a standard MLP with Swish non-linearities as employed by Finzi et al. (2020), as well as the Siren Sitzmann et al. (2020) and WIRE Saragadam et al. (2023) INRs. Note that this choice will also have an effect on the equivariance error, as we are working with a discrete pixel grid when representing images, and any symmetry breaking operations will propagate the loss of equivaraince through the network. Figure 2 quantifies the degree to which the performance of the model described in the previous section is affected by the number of MC samples used when approximating the convolution/cross-correlation integral. In general, we observe significant performance degradation when employing $\leq 6$ samples and as in previous work on integral approximations of continuous convolutions Knigge et al. (2022) observe no additional benefits beyond $12-14$ samples. However, an exact specification of the approximation bounds corresponding to the groups employed is missing in our presentation.

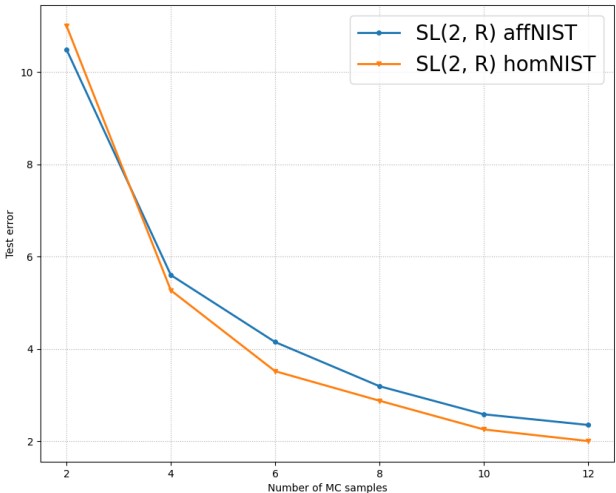

Figure 2: Test error on affNIST/homNIST as a function of MC samples.

