# OpenReview forum: "Lie Group Decompositions for Equivariant Neural Networks"
_ICLR.cc/2024/Conference — ICLR 2024 poster_

### Official Review · Reviewer_8ViQ · 2023-10-31

**Soundness:** 3 good
**Presentation:** 3 good
**Contribution:** 3 good
**Rating:** 8
**Confidence:** 3

**Summary:**

The authors propose to use Lie groups and Lie algebra to develop networks that can be equivariant with respect to affine transformations. They provide the theoretical framework to work with Lie groups that surpasses some of the issues encountered due to their surjective nature. \
The way they managed to enable invariant integration was achieved by decomposing the larger groups into subgroups and manifolds that can be treated individually. \

Main contributions are two-fold: \
a) they managed to develop a non-surjective method of the exponential map \
b) they achieved invariant integration with respect to the Haar measure in a principled manner\
----
Many thanks for you response

**Strengths:**

This is a robust and well-written paper that is mathematically sound (as far as I can tell given it is almost impossible to check everything in the absence of source code). The ablations studies are good, but am not sure whether the experiments on affnist are enough. Nevertheless affnist remains the gold standard - more or less - for evaluating such properties.

**Weaknesses:**

-- It does not matter too much to be frank, as SOTA is not everything, but please have a look at this paper which to my knowledge remains the SOTA on Affnist (figure 5 for instance)
https://aaai.org/ojs/index.php/AAAI/article/download/5785/5641 \

With 174K parameters (64,16,16,16,10) they have achieved 98.1% accuracy on Affnist when trained at 99.7% of MNIST accuracy.
So you might want to update that, considering the number of parameters entailed too. \

-- I do not think the paper needs to be that heavy in maths - too many derivations that could have given space to further experiments and ablations. \
-- I would expand the related work, and flesh out the contributions in the introduction.

**Questions:**

1) To what extent have you based the novelty of your paper on achieving SOTA performance on Affnist? \
2) Is there a scope that your method can be used in tandem with CapsNets?\
3) Would a dataset such as SmallNorb be relevant in this context?\

---

> ### Author Response · Authors · 2023-11-21
>
> Thank you for taking the time to review our paper and for the constructive feedback!  We have started incorporating the suggested changes to make the presentation more clear (see updated version), and will continue to incorporate further changes in the final revision. For the final revision we aim to incorporate further experiments and numerical analysis of the degree invariance/equivariance of our models. The current revision focuses on clarifying the proposed framework and tidying up our presentation. In the following we will focus on the reviewer's questions and address the weaknesses subsequently as they are related to the empirical validation of our paper and a more comprehensive review of related work.
>
> > To what extent have you based the novelty of your paper on achieving SOTA performance on Affnist?
>
> Not at all! We've updated the paper to make our contributions more clear.
> The affNIST dataset was simply the standard benchmark for robustness to "larger" transformations (beyond rotations) on image classification tasks. We want to thank the reviewer for bringing the particular Capsule paper mentioned in "Weaknesses" to our attention. We will incorporate it in the next revision of the paper. Our models' performance can be further improved by taking more Monte Carlo samples of the group, however we will focus on this once we revise the paper with further experiments.
>
> > Is there a scope that your method can be used in tandem with CapsNets?
>
> Thank you for the question, that is an interesting proposal! Capsule methods which rely on a discretization of the convolution integral such as [1] (see Section 4 of their paper) could in theory make use of the integral decomposition and sampling techniques proposed in our work. Further considering the framework of [1], one could make use of the Lie algebra parametrization to obtain tangent vectors which identify each specific group element, and allow for the calculation of the Frechet mean (in an equivariant manner) and the geodesic distance, which would play the role of the averaging operator $\mathcal{M}$ and the distance $\delta$ required by [1] (Section 2.1). We will consider this for possible future work.
>
> [1] - Jan Eric Lenssen, Matthias Fey, and Pascal Libuschewski. Group equivariant capsule networks. Advances in neural information processing systems, 31, 2018.

---

> > ### Comment · Reviewer_8ViQ · 2023-11-22
> > **thank you**
> >
> > Many thanks for your responses.
> > I am happy with that so I will increase my score

---

### Official Review · Reviewer_gz8n · 2023-11-01

**Soundness:** 2 fair
**Presentation:** 2 fair
**Contribution:** 2 fair
**Rating:** 5
**Confidence:** 3

**Summary:**

This paper presents a framework for the development of equivariant (regular) neural networks, specifically tailored to matrix Lie groups that may not be compact or abelian. The authors explain a strategy for the development of equivariant layers by subdividing larger groups into smaller subgroups or subspaces. This partitioning facilitates efficient sampling of group elements, conducted in accordance with the Haar measure, by allowing independent sampling from these defined subsets. Grounded in the theory of Lie group structure and the geometry of their homogeneous spaces, the paper focuses its analytical scope primarily on the groups $G = GL^+(n, R)$ or $G = SL(n, R)$.

**Strengths:**

1. While (regular) group convolution for non-abelian and non-compact groups have been proposed using basis functions on the Lie algebra. The exponential map, however, is only diffeomorphic locally around 0. In addition, the paper also aims the address the issue of efficient sampling of group elements according to the Haar measure for regular group convolutions. In this aspect, the paper is novel.
2. Preliminary experiments are conducted to verify the potential of the framework.

**Weaknesses:**

1. My main concern of the paper is its mathematical denseness. I would recommend the authors to focus on the main message, and defer unimportant details to the appendix. For example, the condition of part (2) of Theorem 4.1 is probably to general, and it is not helpful for understanding the paper.
2. The notation of the paper is also sometimes confusing. The (larger) group $G$ is sometimes refer to the affine group $G = R^d\rtimes H$ (mostly in Section 4). However, $G$ in Section 4 is mostly refer to the subgroup $H = GL^+(n, d)$.
3. One of the issue the paper aims to address is the global non-surjectiveness of the exponential map. However, since most (group) convolutional filters are only defined locally (on a neighborhood of the identity), does the existence of local diffeomorphism already suffice?
4. Page 5, line 4 after "Factorizing the Haar measure", for general subsets $P\subset G$ and $K\subset G$, what does it mean to have "G-invariant" Radom measure. Although the authors later say that $P$ will always be a $G$-homogeneous space (so the $G$-action on $P$ is essential an action on the quotient space), it would be much better to state this assumption in the beginning.
5. On page 4, the authors say $H = SO(n)$ is abelian. Is this the case for $n\ge 3$?
6. The theory seems to apply only to $H = GL^+(n, R)$ or $H = SL(n, R)$. Does it apply to a general $H\subset GL(n, R)$?
7. Is the Lie bracket of the Lie algebra used anywhere for the construction?
8. Can this construction be extended to representations beyond regular representations of $G$? For high-dimensional $G$, such as $E(n)$ with large $n$, sampling from $G$ will suffer from COD.

**Questions:**

See the previous section

---

> ### Author Response · Authors · 2023-11-21
>
> We thank the reviewer for the constructive comments. In the following, your comments are first stated and then followed by our point-by-point responses.
>
> > My main concern of the paper is its mathematical denseness. I would recommend the authors to focus on the main message, and defer unimportant details to the appendix. For example, the condition of part (2) of Theorem 4.1 is probably to general, and it is not helpful for understanding the paper.
>
> We agree that the paper presentation can be improved and have started incorporating the suggested changes to make the presentation more clear (see updated version), and will continue to incorporate further changes in the final revision.
> Part (2) of Theorem 4.1 is indeed a very general result, however we consider it an essential component of the overall framework as conditions (1) and (2) of this theorem categorize the major classes of decompositions that are employed in the context of invariant integration on topological/Lie groups. For the larger groups $\text{SL}(n, \mathbb{R})$ and $\text{GL}^{+}(n, \mathbb{R})$ which are decomposed both at the group and algebra level it is Theorem 4.1 that allows us to formalize the change of variables in the integral formula so as to eventually construct a sampling mechanism on these groups using their subcomponents.
> There are several ways these groups can be decomposed, with the Cartan decomposition being one such choice, which is what we've employed. The Cartan decomposition is a decomposition of the form $G = G/H \times H$ as in Theorem 4.1 (1), while the standard decomposition for the semi-direct product can be understood in terms of Theorem 4.1 (2). The reviewer is correct that presenting only one example for such a general theorem might not be well-motivated. In the revised manuscript we included included further discussion of possible decompositions which make use of Theorem 4.1 (2) (see Section B.6.1 of the Appendix), highlighting the fact that depending on the Lie group of interest, the integral decomposition can be realised in several ways. The decomposition into a product of subgroups (even if one of the subgroups is also a homogeneous space) will lead to a different set of tools being available when we go beyond the integral decompositions and try to understand what Lie algebra decomposition would correspond to the group decomposition.
>
> > The notation of the paper is also sometimes confusing. The (larger) group $G$ is sometimes refer to the affine group $G = R^d\rtimes H$ (mostly in Section 4). However, $G$ in Section 4 is mostly refer to the subgroup $H = GL^+(n, d)$.
>
> We agree! The notation for the groups can indeed be unclear and we have made an effort to change the presentation. Specifically, we have opted for a notation of the form $\textnormal{Aff}(G) \coloneqq = \mathbb{R}^{n} \rtimes G$ indicating that $G$ is the structure group of the affine group $\textnormal{Aff}(G)$. We believe the presentation is much more clear now.
>
> > Page 5, line 4 after "Factorizing the Haar measure", for general subsets $P\subset G$ and $K\subset G$, what does it mean to have "G-invariant" Radom measure.
>
> Thank you for highlighting this particular part of the presentation.
> What we should have made more clear (and have clarified in the revision) is that a topological space $X$ on which a group $G$ acts can have a $G$-invariant Radon measure even when $X$ itself is not a group. If $X=G$ (and we are describing the action of the group $G$ on itself), then the $G$-invariant Radon measure on $X = G$ is simply the Haar measure of $G$ (Thm. A.1). When $X$ is a topological space on which $G$ acts but it is not necessarily $G$ itself, an invariant measure on $X$ (invariant with respect to the action of $G$) may still exist, and it is simply referred to as a $G$-invariant (Radon) measure. For example the SPD matrices are a homogeneous space of $\text{GL}^{+}(n, \mathbb{R})$, with the $\text{GL}^{+}(n, \mathbb{R})$-invariant measure being described in Section A.10 of the appendix. In any case, this particular distinction has now been made clear in the paper explicitly (see Remark A.2 of the Appendix).
>
> > Although the authors later say that $P$ will always be a $G$-homogeneous space (so the $G$-action on $P$ is essential an action on the quotient space), it would be much better to state this assumption in the beginning.
>
> We have chosen to work with $P$ a homogeneous space, however this is not absolutely necessary as indicated in a previous response.
> We agree however that the role of $P$ can be made more clear and have attempted to address this in the revision by clarifying that the properties of the chosen decomposition at the Lie algebra level are also important for the construction of the map from the group to the Lie algebra.
>
> > On page 4, the authors say $H = SO(n)$ is abelian. Is this the case for $n\ge 3$?
>
> Thank you for pointing this out. We have clarified that the BCH simplification was employed in previous work for $n=2$.

---

> ### Author Response · Authors · 2023-11-21
>
> > One of the issue the paper aims to address is the global non-surjectiveness of the exponential map.
> However, since most (group) convolutional filters are only defined locally (on a neighborhood
> of the identity), does the existence of local diffeomorphism already suffice?
>
> Our work fits within the recent class of methods which rely on constructing
> the convolution kernel analytically using the group Lie algebra.
> Within this framework, to define a kernel taking in Lie algebra vectors
> corresponding to the group elements $g^{-1}\tilde{g}$ of the convolution, one
> has to construct a map \textbf{from the group to the algebra} (denoted by $\xi^{-1}: G \to \mathfrak{g}$ in our paper).
> One exception to this case is when the group is compact and Abelian, as in that
> case we can work only in the Lie algebra, as remarked in the paper.
> If the Lie group exponential map is not surjective, we cannot formally
> make the change of variables of Equation (8), and the map $\xi^{-1}$ might
> not be defined.
> The reviewer is correct that in some cases one could address this
> by simply restricting the integration domain
> to the injectivity radius of the group exponential, which for the Lie groups employed
> in the context of machine learning will be the matrix exponential,
> making $\xi^{-1}$ the matrix logarithm.
>
> To understand the motivation behind attacking the problem of non-surjectivty, first
> we must consider the degree to which the group exponential `covers' the
> group of interest, and for which group elements is $\xi^{-1}$ defined.
> This is formally addressed by Proposition A.4 in the Appendix. For many groups of interest
> (e.g. scaling transformation, $\text{SL}(n, \mathbb{R})$ or $\text{GL}^{+}(n, \mathbb{R})$) the complement of the image
> of the group exponential is not a set of measure zero.
> If one decides to opt for a restriction to the injectivity radius of the exponential,
> the expressivity of the model suffers, as the space of equivariant functions
> that can be approximated by the resulting method is reduced. Of course, the degree
> to which this is a problem is dependent on the particular application
> and the Lie group employed.
> Another issue to consider is that if one is employing a Monte Carlo approximation
> of the convolution/cross-correlation integral then it is possible that
> random group elements $g^{-1}\tilde{g}$ might be sampled outside of domain
> of definition of the matrix logarithm which would then require the
> sampling mechanism to reject these samples (incurring an additional computational
> cost), as otherwise we would have numerical errors introduced in the
> approximation independent of optimization process. In some sense, the sampling
> mechanism itself has to be designed with this restriction in mind.
> The reviewer is correct that in practice our filters/kernels are defined locally.
> We state this more precisely in the updated version of the paper and indicate that
> we can assume at least one of $f(\cdot)$ or $k(\cdot)$ in
> $\int_{G} f(\tilde{g})k(\tilde{g}^{-1}g) d\tilde{g}$ can be taken
> to have compact support. The advantage of our framework is that one does not have to
> assume that the support of the feature maps/kernels is tied in any way to the
> injectivity/surjectivity radius of the group exponential.
>
> > Is the Lie bracket of the Lie algebra used anywhere for the construction?
>
> The Lie bracket was presented as in the abelian case it leads to a simplified representation of the matrix logarithm via the BCH formula. However, in our construction we do not need to use the Lie bracket explicitly. We have simplified the presentation to record only the absolutely necessary details in the main text with further discussion in the Appendix.
>
> > The theory seems to apply only to $H = GL^+(n, R)$ or $H = SL(n, R)$. Does it apply to a general $H\subset GL(n, R)$?
>
> It is indeed the case that the tools are applicable in a more general setting, and the original purpose of the paper was to present a general framework that allows for one to deal with arbitrary groups in the context of integration/sampling/parametrization with the construction of conv. kernels being a primary application. We mentioned only briefly in the introduction that the methodology if applicable to any Lie group with finitely many connected components, however the degree of generality/applicability of our method was not clearly communicated. In the revised version of our paper we have made an attempt to clarify the 'framework' aspect of our approach and its components, as well as to point towards references which show that most matrix Lie groups $G \leq \textnormal{GL}(n, \mathbb{R})$ fulfill the conditions of this framework. Namely, non-compact groups can be decomposed into their maximal compact subgroup and some Euclidean factor. Affine groups can be decomposed/parametrised as indicated in the paper. Compact groups with surjective exp. maps can immediately be employed within this framework (and with tools employed by previous work).

---

> > ### Comment · Reviewer_gz8n · 2023-11-22
> >
> > I thank the authors for the detailed response. However, my concern on the extension beyond regular representations still remains. Moreover, I agree with the other reviewers that the sampling issues have not been sufficiently addressed. I have raised my rating accordingly.

---

> ### Author Response · Authors · 2023-11-22
>
> We thank the reviewer for considering the revised paper. We will try to address the last questions and are working to provide some numerical quantification of the equivariance/invariance error depending on the number of MC samples. Some initial remarks follow.
>
> > For high-dimensional $G$, such as $E(n)$ with large $n$, sampling from $G$ will suffer from COD.
>
> Sampling efficiency is actually one requirement we had in mind when developing our construction. Note that the cases $n = 2$, or $n = 3$ cover most machine learning applications of interest. In this case the largest spaces we would need to sample from would be GL(2, R) (4 free parameters) and GL(3, R) (9 free parameters). Of course as the reviewer points out, because our approach involves a numerical integration procedure the convergence is depends on the number of MC samples. The advantages of our methodology is that if one can find a change of variables via the product space decomposition and the subcomponents are independent (which occurs when the measure decomposes as a product of measures), then the simulation problem can be transferred from the larger space to the smaller-dimensional factors. Then, we can consider the problem of sampling on these spaces. Note that, if we fully embrace the idea that the problem has reduced to a MC integration of the convolution integral, we can make use of classical variance-reduction techniques to improve our sample efficiency. This is already shown to some degree in our experiments, when we decompose GL+(2,R) as SPD(2,R) x SO(2,R), on the rotation group we use a quasi-monte carlo [1] approach and sample equi-distant points in [0, 2\pi]. A similar procedure could be done for SO(3) where we could pre-compute the samples (using for example some discretization such as the icosahedron) and keep the SPD factor as being sampled randomly to obtain better convergence rates. Other methods for variance reduction could be applied, as shown for example in [2]. In any case, the solution in the paper is in some sense the most basic form of MC integration, and there is still space to improve sample efficiency.
>
> [1] - https://en.wikipedia.org/wiki/Quasi-Monte_Carlo_method
>
> [2] - Lagrave, Pierre-Yves, and Frédéric Barbaresco. "Adaptive Importance Sampling for Equivariant Group-Convolution Computation." Physical Sciences Forum. Vol. 5. No. 1. MDPI, 2022. - https://www.mdpi.com/2673-9984/5/1/17
>
> > Can this construction be extended to representations beyond regular representations of $G$
>
> A complete answer here is somewhat involved. What we would be looking for in some sense are subrepresentations of the regular representations since other representations (irreducible, quotient) are contained within it. First one need to consider when it is appropriate to use (or start from) the regular representation. This can be motivated if one desires to construct "dense" convolutional operators (via integral transforms) where we assume the input signal to be modelled as a (non-sparse) continuous representation, as can be done for 2D images (the input is then modelled as a function $f \in L^{2}(\mathbb{R}^{c}, \mathbb{R})$ since with this representation we assume no loss of information). For other types of inputs this assumption is not absolutely necessary e.g. if we are constructing (equivariant) MLPs between finite-dimensional vector spaces. Once we have decided that we want convolution operators, two common choices for their implementation are group convolutions or steerable convolutions. In [3] it is shown that there exists an equivalency between the approaches in that steerable kernels on a homogeneous space X are in correspondence with certain kernal operators on $L^{2}(X)$. The equivalence is established for $G$ compact by the application of the Peter-Weyl theorem (which will not hold for non-compact groups), and $L^{2}(X)$ is decomposed into a Hilbert sum of closed subspaces, and each subrepresentation of $G$ on these subspaces is equivalent to an irreducible representation. For certain non-compact non-abelian groups the irreducible representations are infinite-dimensional and the generalization of this result is not straightforward (see theorem 17.2 in [4]). In any case, we will try to discuss this in the paper. The decompositions explored in our paper (Iwasawa, Cartan) can be used to study for example finite-dimensional representations of Lie algebras of semi-simple Lie groups, and have application within the context of the representation theory of such groups [5]. One can see our approach as a first step towards constructing equivariant (convolutional) layers when working with such groups.
>
> [3] - Lang L, Weiler M. A wigner-eckart theorem for group equivariant convolution kernels. arXiv preprint arXiv:2010.10952. 2020 Oct 21.
>
> [4] - J. Gallier and J. Quaintance. “Aspects of Harmonic Analysis and Representation Theory”. In: (2023).
>
> [5] - Anthony Knapp. Lie Groups Beyond an Introduction, Second edition, volume 140. Jan 2002.

---

### Official Review · Reviewer_azQE · 2023-11-01

**Soundness:** 4 excellent
**Presentation:** 2 fair
**Contribution:** 3 good
**Rating:** 6
**Confidence:** 3

**Summary:**

This paper presents a method for constructing approximately Lie group equivariant neural nets. Like previous work, their method relies on Monte Carlo approximation of infinite integrals; however, they extend this methodology to Lie groups without surjective exponential maps. Their critical insight is to decompose certain Lie groups of practical interest into semidirect products, which simplifies the form of the Haar measure. They evaluate their method in modifications of MNIST and outperform baselines.

**Strengths:**

The framework is quite general, in that it applies to Lie groups with non-surjective exponential maps. The experimental performance is strong relative to baselines, including other work that handles groups with non-surjective exponential maps. The presentation of the underlying math is quite detailed, and is a good resource on its own (especially the appendix). The decomposition of a Lie group into groups via the semidirect product is novel in a machine learning context, as far as I know.

**Weaknesses:**

1. From what I can tell, the input and output spaces/representations are restricted to either the group itself under the regular or trivial representation, or a homogeneous space of the group. This does not seem as general as e.g. the two papers I mention in point (3), which can work with any finite-dimensional representation.

2. The advantage of this approach over MacDonald et al 2022 is not made sufficiently clear, since both approaches can also handle Lie groups with non-surjective exponential maps.
3. The related work is missing at least two important papers on Lie group equivariance: “A Practical Method for Constructing Equivariant Multilayer Perceptrons for Arbitrary Matrix Groups” by Finzi et al 2021, and “Lorentz Group Equivariant Neural Network for Particle Physics” by Bogatskiy et al 2020. It would be important to compare to these papers (both in theory, in terms of the Lie groups and input/output representations to which they apply, and in the experiments), since they are exactly equivariant.
4. To a similar point, this method is only approximately equivariant (due to the sampling in the integral computation), rather than exactly equivariant. This could be expressed more clearly in the introduction of the paper. It would also be helpful to have computational tests of equivariance, which assess to what degree the learned network is actually equivariant after integration error.
5. The paper’s presentation could be significantly improved. In addition to typos, some of which are noted below, the writing leaves something to be desired overall. For example, the main body of the paper is very dense in mathematical background and details. It would be helpful to clearly and concisely express the main ideas of the paper earlier on and to give only the important intuitions, and to defer many such details to later in the paper or the appendix.
6. Enforcing equivariance to Lie groups with non-surjective exponential maps could be much better motivated — applications are not discussed very often, and the datasets used are synthetic (e.g. affine transformations of MNIST). When does such equivariance arise in practice?

Here are several typos I encountered while reading:
* Page 1, abstract: backwards quotation mark on the word larger
* Page 2, fourth line: backwards quotation mark on the word larger
* Page 2, related work: “Recent proposal” —> “Recent proposals”
* Page 2, bottom line: “maps can be defined AS k-channel”
* Page 3, between equations 5 and 6: “ecompasses”
* Page 4, in Lie algebra parametrization: “approxiamtion” and “encompasses recent proposalS”
* Page 7 before Theorem 4.4: “to obtain GL(n,R)-sampleS”

**Questions:**

1. My understanding was that non-compact groups do not have finite Haar measures (i.e. the measure of the entire group is infinite). I am therefore confused how e.g. equation 13, and really all the integrals, are well-defined without additional assumptions on the bounds of integration or the decay of the function being integrated.
2. Could the authors elaborate on the primary advantages of this integration method over that of MacDonald et al 2022, which also handles the case of Lie groups with non-surjective exponential maps? The related work mentions memory requirements, but it is not clear to me how this method circumvents these issues (or what the precise issues are).
3. Assuming the main advantage over MacDonald et al 2022 is in memory requirements, why then do the authors think that MacDonald et al had slightly worse performance in the experiments?
4. The papers by Finzi et al and Bogatskiy et al, noted under “weaknesses” above, are not integration methods. Are they suited for the input and output representations that this task considers? If so, can the authors compare to these methods as well?
5. At the top of page 8, the authors note that they sample the translation factor from $[-1,1]^n \subset \mathbb{R}^n$. How is this justified, when the integral is defined over all of $\mathbb{R}^n$? Doesn’t this induce some error, which is compounded in subsequent layers?
Is this construction universal? I.e. ignoring the Monte Carlo integral approximation error, can any continuous equivariant function be represented?

---

> ### Author Response · Authors · 2023-11-22
>
> Thank you for taking the time to review our paper and for the constructive feedback! We have updated the paper, with the current revision focusing on clarifying our contributions, streamlining the presentation and incorporating as much feedback as possible from the reviewer's comments. We plan on soon adding another revision which focuses on further experimental validation and numerical equivariance error analysis, while also providing a broader comparison with related work. The typos have also been address (with the exception of the abstract, as it can only be modified in case of acceptance for the Camera Ready version).
>
> > My understanding was that non-compact groups do not have finite Haar measures (i.e. the measure of the entire group is infinite). I am therefore confused how e.g. equation 13, and really all the integrals, are well-defined without additional assumptions on the bounds of integration or the decay of the function being integrated.
>
> Thank you for pointing this out. Due to the heavy mathematical language in the paper we were unsure whether to include further discussion related to the existence and range of the conv. operators. In the revised manuscript, one can find in Appendix A.2.1 several assumptions and propositions which clarify sufficient conditions for the convolution operators to be well defined. Note that it is possible to work with even weaker assumptions, and as the reviewer points out beyond the support of the functions themselves their decay rates could be used to characterize the existence of these operators. Note that, under our framework if one decides to make a restriction of the integration domain it is not tied in any way to the group exponential (of course some restriction already occurs in practice for most types of data signals). One could for example take some norm-bounded set of matrices {$\{A \in \textnormal{GL}(n, \mathbb{R}) \mid a <= |A| <= b \}$} (where we can take $\|\cdot\|_{F}$ to be e.g. Frobenius norm, and $a, b \in \mathbb{R}_{>0}$) and because Lie groups are Polish and completely metrizable (by the Heine-Borel property) this set is compact. One could then construct a uniform distribution on this set. In our approach we work with the non-orthogonal factor (the SPD manifold) sampled from a Gaussian distrubtion. We can include further discussion on this case aswell in a further revision of the manuscript, and we point the interested reader towards Section 1.3.4 of [1].
>
> [1] Audrey Terras. Harmonic analysis on symmetric spaces—higher rank spaces, positive definite matrix space and generalizations. Springer, 2016.
>
> > Assuming the main advantage over MacDonald et al 2022 is in memory requirements, why then do the authors think that MacDonald et al had slightly worse performance in the experiments?
>
> In the revised paper we have added Sec. A.3 (in the appendix) which discusses this work in more detail! We mostly focus on analyzing the expressiveness and theoretical guarantees of their method while also pointing to numerical errors that can appear in practice as a consequence of their approach (the memory limitations having been already mentioned). Of course it is possible that aspects related to the implementation of their model also affect the sample efficiency, however the approach is theoretically clear where we can at least point out were these issues can arise from.
>
> > Could the authors elaborate on the primary advantages of this integration method over that of MacDonald et al 2022, which also handles the case of Lie groups with non-surjective exponential maps? The related work mentions memory requirements, but it is not clear to me how this method circumvents these issues (or what the precise issues are).
>
> Besides the comments in the previous response (which point to Appendix A.3), we can make some remarks. Both methods employ a change of variables in the integration procedure. In our case, this happens at the group-level, and we reduce the numerical simulation problem from a larger space to a product of (independent!) factor spaces. This means that we can construct independent sampling mechanisms on the individual factors to obtain a sample on the original space. In general the smaller spaces will be easier to deal with, and for example the orthogonal groups SO(2) and SO(3) (which would appear in the most common applications of our approach for $n \in \{2,3\}$) already have a large amount of literature dedicated to methods which can be used to sample efficiently on these spaces with respect to their Haar measure. Without the decomposition tools specific to $SO(n)$ could not be employed. The same is true for the case of SPD matrices, which are also a manifold which has a wide variety of applications.
> The reason we avoid exponential memory constraints comes down to not having to reorder the convolution operation (as described in A.3). We only need to sample effectively from 1 layer before the current one, similar to previous proposals on regular group cnns.

---

> ### Author Response · Authors · 2023-11-22
>
> > At the top of page 8, the authors note that they sample the translation factor from $[-1,1]^n \subset \mathbb{R}^n$. How is this justified, when the integral is defined over all of $\mathbb{R}^n$? Doesn’t this induce some error, which is compounded in subsequent layers? Is this construction universal? I.e. ignoring the Monte Carlo integral approximation error, can any continuous equivariant function be represented?
>
> The presentation here is very much lacking and we thank the reviewer for bringing attention to this part of the manuscript. The framework itself does not require this discretization. The assumption here was actually done in the context of image classification where we consider that the underlying sampling grid is restricted to $[-1,1]^n \subset \mathbb{R}^n$. Further restriction can also appear in the context of point cloud data when we normalize our inputs. But the restriction itself is application dependent, and does not influence the generality of the framework. The idea is that if the data is in $\mathbb{R}^{n}$ we often make a simplifying assumption about the range of this data to be able to construct convolution operators more efficiently. Yes, any continuous equivariant function can be represented, with the complexity and accuracy of the representation being influenced by the number of MC samples and the dimension of the space, as with other numerical integration procedures.
>
> > Enforcing equivariance to Lie groups with non-surjective exponential maps could be much better motivated — applications are not discussed very often, and the datasets used are synthetic (e.g. affine transformations of MNIST). When does such equivariance arise in practice?
>
> We agree that further motivation for this class of transformations would benefit the paper and intend on discussing this in our next revision. There are several ways to motivate the applicability of such groups, one of them being as the reviewer points out specific problems where more exotic symmetries are present. Another way the applicability of these groups could be motivated is if on desires additional out-of-distribution robustness, independent of the degree of symmetry in the dataset. For example, if one is building a computer vision system and the geometric variability in the dataset is low, lets say only rotations, it does not mean that the robustness of the system has to be limited only to those transformations that are found in the dataset (consider the case of a self-driving car system trying to detect lanes, pedestrians, etc.). In some cases neither the training set nor the test set can indicate exactly what the symmetry group of the underlying real-world object is. An equivariant network is advantageous even in this case, as the entire network can be made invariant to all affine transformations, i.e. we can obtain a form of out-of-distribution robustness irrespective of the symmetry found in the present form of the dataset. We agree that there are objects especially in physics and chemistry which are assumed to be governed by rigid-body motions. However, even in this case deformations can occur for example in molecule interactions/dynamics. Other areas of application are pose estimation with full affine degrees of freedom, image registration (especially in the context of medical imaging) and robotic motion/manipulation.
>
> > The paper’s presentation could be significantly improved. In addition to typos, some of which are noted below, the writing leaves something to be desired overall. For example, the main body of the paper is very dense in mathematical background and details. It would be helpful to clearly and concisely express the main ideas of the paper earlier on and to give only the important intuitions, and to defer many such details to later in the paper or the appendix.
>
> Thank you for the feedback! We have incorporated the reviewer's feedback in the current updated version of the paper streamlining the presentation and only presenting the essential concepts in the main text, while also making more clear the our contributions.

---

> > ### Comment · Reviewer_azQE · 2023-11-22
> > **Thank you for the response**
> >
> > I thank the authors for their response. Although the reviewers did not address weakness 1 or question 4, and I still feel that the presentation of the paper leaves room for improvement (e.g. the conditions from appendix A.2.1 should at least be cited in the main body near equation 13, the comparison to MacDonald et al should perhaps be in the main body as well, and the draft remains mathematically dense in a way that perhaps obscures its primary contributions/ideas), I am reasonably satisfied with the novelty of their approach relative to MacDonald et al and think this paper would serve as a useful mathematical resource for the community. I will increase my rating by 1.

---

### Official Review · Reviewer_kmML · 2023-11-01

**Soundness:** 4 excellent
**Presentation:** 3 good
**Contribution:** 3 good
**Rating:** 6
**Confidence:** 3

**Summary:**

This paper introduces a way to build group convolutional (Gconv) layers for groups in which the exponential map from the Lie algebra to the group is not surjective (e.g. $GL(n,mathbb{R})$). The problem is that Gconv is an integral over the group, which is often approximated by discrete sampling done using the exp map. When the exp map is not surjective, we can't rely on it to get good coverage in the sampling.
To mitigate this, this paper suggests to find a decomposition of the group in the form $G=PK$ such that the integration (Haar) measure over the group also factorizes $d\mu_G = d\mu_P d\mu_K \delta(\cdot)$. In certain cases, such as the Affine group, this decomposition can yield familiar subgroups for which we have either a surjective exp (e.g. $SO(n)$) or efficient sampling strategies (e.g. diffeomorphic to $\mathbb{R}^k$).
After deriving the theory behind such decomposition, they introduce a Monte Carlo sampling scheme for it. Then they conduct a couple of toy experiments to show potential benefit of their design.

**Strengths:**

1. Mathematical rigor: goes through the details of the structure of Gconv to pinpoint the issue arising from non-surjective expm.
2. Has extensive derivations, appendices and concrete calculations of the integration measures, rarely stated in other works.
3. The Monte Carlo sampling scheme seems to work well with few samples and may alleviate memory issues of others.
4. The experiments, though limited, are useful.

**Weaknesses:**

1. Mathematical rigor: very dense, difficult to read, too much re-derivation of previous results. A good part of sec. 3, Background, in __continuous group equivariance__ and some of __Lie algebra parametrization__ can be cut and moved to the appendix. Sec. 4 can also be shortened, like derivations in __layer definition__ in 4.2. The density of the math makes the paper hard to read.
2. The paper calls $SO(n)$ abelian, which is incorrect for $n>2$ (page 4, sec 4).
3. Other literature: Many Gconv approaches do not sample the group this way (ses question): Clebsch-Gordon Nets (Kondor et al NeurIPS 2018), Lorentz group (Bogatskiy, ICML 2020), EMLP (Finzi 2021) or Lie algebra conv (Dehmamy, NeurIPS 2021).

**Questions:**

1. How are methods relying on irreps affected by the expm not being surjective? Does it matter in those cases?
2. For the Haar measures to factorize to $\mu_K \otimes \mu_P$, do we need $G=PK$ or semidirect or direct product? A number of different decompositions are used in 4.1 and 4.2 and the requirements are not fully clear.
3. I see that $G=PK$ can be done when $P$ or $K$ is a normal subgroup. In the $SL(n)$ to $\mathrm{SPos} \times SO(n)$, is $SO(n)$, is one of the factors also a normal subgroup?
4. Even when expm is not surjective, the product of multiple $g_i = \mathrm{expm}(X_i)$ can produce any arbitrary group element, as I understand it. Doesn't this mean that multi-layer Gconv using expm should in principle be sufficient to get good sample coverage on the group? In other words, is your decomposition really necessary?

---

> ### Author Response · Authors · 2023-11-22
>
> Thank you for reviewing our paper and for the constructive feedback! We have updated the paper incorporating reviewer feedback and are working to address further empirical validation concerns, as well as to provide numerical equivariance erorr analysis.
>
> > For the Haar measures to factorize to $\mu_K \otimes \mu_P$, do we need $G=PK$ or semidirect or direct product? A number of different decompositions are used in 4.1 and 4.2 and the requirements are not fully clear.
>
> Both cases are possible! In the semi-direct product case the decomposition is somewhat trivial in the euclidean space case, as this is simply the translation factor of the convolutions. The idea of the paper is that we have a general framework where we can perform a decomposition G = PK for multiple cases : P is homogeneous space, P is a subgroup (and we have a direct product decomposition) and P is a subgroup that is normal (and we have a semi-direct product decomposition). All cases are covered by this framework. The specific decomposition described by the reviewer is an application of the more general theorem, and a specified decomposition of the groups G = GL(n) and G = SL(n). The decomposition leading to a decomposition of the Haar measure into a product of invariant measures is actually a stronger condition than just a decomposition at the group or algebra level. The objective was to not only find a decomposition, but one which leads to a product space decomposition of independent factors, so as to be able to sample on the individual space and produce a sample on the product space and therefore on the original group. We have tried to make this more clear in the revision as well as provide further examples of possible decompositions (see Appendix B.6.1).
>
> > I see that $G=PK$ can be done when $P$ or $K$ is a normal subgroup. In the $SL(n)$ to $\mathrm{SPos} \times SO(n)$, is $SO(n)$, is one of the factors also a normal subgroup?
>
> The original presentation in the paper was indeed not clear and the exact representation of each of the factor spaces might be confused. It is not necessary for one of the components to be a normal subgroup (this is the case when we are working with the semi-direct product $\mathbb{R}^{n} \rtimes G$ for example). Actually in the decompositions employed one of the components is not even a group (the SPD factor), it is a homogeneous space of the original group which we decompose. So one the P factor can be a group (normal or not) or a homogeneous space. In the case of the Cartan Decomposition we should also mention that the SPD factor is indeed diffeomorphic to another group (upper triangular), which could be used and would lead to another decomposition of GL(n) (as shown in B.6.1).
>
> > Even when expm is not surjective, the product of multiple $g_i = \mathrm{expm}(X_i)$ can produce any arbitrary group element, as I understand it. Doesn't this mean that multi-layer Gconv using expm should in principle be sufficient to get good sample coverage on the group? In other words, is your decomposition really necessary?
>
> Thank you for the question! We have attempted to clarify this more precisely in the revision of the paper. Considering each question separately:
>
> > Even when expm is not surjective, the product of multiple $g_i = \mathrm{expm}(X_i)$ can produce any arbitrary group element, as I understand it.
>
> In some sense this is already happening. The reviewer is correct that when the group exponential is not surjective, it is still possible to take a **product** of exponentials to represent an element. The problem is that this simple representation will in general not offer any way for the parametrization map $\xi^{-1}: G \to \mathfrak{g}$ to be constructed, and will not simplify the integration problem (see for example Remark B.11 in the updated manuscript which explains that a first alternative that was considered for the group exponential is the Riemannian exponential. However the issues lies with the fact that the Riemannian logarithm does not have a close form expression and we would then require an optimization process to happen at each layer for each forward pass). The Cartan/Polar decomposition is actually used to represented elements $g \in G$ as $g = e^{X}e^{Y}$ (see Section B.7 for the full details), only that X, Y are not just elements of $\mathfrak{g}$, but rather we decompose $\mathfrak{g}$ into a direct sum of vector spaces and represent each element separately, which allows for the construction of the map $\xi^{-1}$ and the decomposition corresponds to the group-level decomposition which also realizes a decomposition of the integral!
>
> > Doesn't this mean that multi-layer Gconv using expm should in principle be sufficient to get good sample coverage on the group? In other words, is your decomposition really necessary?
>
> The issue is that once one single layer is not invariant/equivariant the property is lost by induction for the later parts of the network.

---

> ### Author Response · Authors · 2023-11-22
>
> > The paper calls $SO(n)$ abelian, which is incorrect for $n>2$ (page 4, sec 4).
>
> Thank you for pointing this out. We have clarified that the BCH simplification was employed in previous work for $n=2$.
>
> > Mathematical rigor: very dense, difficult to read, too much re-derivation of previous results. A good part of sec. 3, Background, in **continuous group equivariance** and some of **Lie algebra parametrization** can be cut and moved to the appendix. Sec. 4 can also be shortened, like derivations in **layer definition** in 4.2. The density of the math makes the paper hard to read.
>
> We have updated the manuscript based on the reviewer's feedback. We agree that the original presentation was too dense in non-essential details and made an effort to streamline the presentation.
>
> > Other literature: Many Gconv approaches do not sample the group this way (ses question): Clebsch-Gordon Nets (Kondor et al NeurIPS 2018), Lorentz group (Bogatskiy, ICML 2020), EMLP (Finzi 2021) or Lie algebra conv (Dehmamy, NeurIPS 2021).
>
> We will discuss these works in our next revision.
>
> > How are methods relying on irreps affected by the expm not being surjective? Does it matter in those cases?
>
> Generally irrep methods are used in the context of compact groups where the exponential map is surjective. In the work of EMLP (Finzi 2021) which you mention, the authors construct equivariant MLPs by solving a system of constraints where the resulting linear system is solved using the singular value decomposition (if the representation is finite dimensional, whereas we work with an infinite-dimensional representation). From their paper, one can review Appendix H where they also employ a form of product of exponentials in the case where the group of interest $G$ is "larger" (and have non-surjective exp maps).

---

### Official Review · Reviewer_yeU2 · 2023-11-02

**Soundness:** 4 excellent
**Presentation:** 3 good
**Contribution:** 3 good
**Rating:** 8
**Confidence:** 3

**Summary:**

The paper considers the problem of designing equivariant neural networks when the group may not be compact or abelian, and when the exponential map for the Lie group may not be surjective. In an earlier work which makes use of the Lie algebra to construct the kernels, when the exponential map is not surjective, the log map is only applicable around the identity in the lie algebra, and is thus restrictive. Furthermore, this method has exponential memory complexity in the group dimension. The focus of the paper is two such groups: $GL^+(n,\mathbb{R})$ and $SL(n,\mathbb{R})$. The key idea is to decompose these groups into subgroups and submanifolds where sampling the group elements is easier and well defined. Experiments on small standard datasets shows improvements over earlier methods.

**Strengths:**

1. The paper is mathematically well-grounded.

2. The contributions in the paper clearly improve on existing works in more than one way: applicable to groups like $GL^+(n,\mathbb{R})$ and $SL(n,\mathbb{R})$

3. The ideas of group decomposition as a way to build equivariance to such groups is an interesting and general idea, and will hopefully be useful in other cases.

4. Experimental results show improvements over state-of-the-art methods.

**Weaknesses:**

1. The number of samples for Monte Carlo integration is not clear to me. When the authors say 10 samples are used for the affine group, do they mean 10 samples for both rotations and scales combined? How many rotations and scales?

2. Although clear theoretically, experiments don't seem to have tested the settings that were setting this apart from the others, particularly, by MacDonald et al. 2022. Perhaps at least one experiment with n=3 with larger scale changes as part of the test set could be interesting. Even a different version of affNIST with larger scale changes could be interesting.

3. Having numerical values of equivariance/invariance error, and as a function of the number of Monte Carlo samples should also be useful here.

3. I don't fully understand why the proposed group decomposition technique is able to estimate the correlation integral well with only 10 samples while the work by MacDonald et al. need 100.

**Questions:**

No additional questions.

---

> ### Author Response · Authors · 2023-11-22
>
> Thank you for reviewing our paper and for the positive feedback! We have updated the paper and are looking to make a final revision which also contains a numerical analysis of the equivariance error. In the following we discuss clarification/theoretical questions about our approach.
>
> > The number of samples for Monte Carlo integration is not clear to me. When the authors say 10 samples are used for the affine group, do they mean 10 samples for both rotations and scales combined? How many rotations and scales?
>
> The number of rotation and SPD matrices sampled are the same in this sampling scheme. For example, let $n = 2$ and suppose we want to produce 10 samples. To produce a sample $A \in G$ ($G = GL+(2, R)$ or $G = SL(2, R)$), we sample on the individual factors and have a 10 x 2 x 2 tensor R of 10 random SO(2) matrices and a 10 x 2 x 2 tensor S of 10 random SPD matrices which we multiply (S^{1/2}R = A) to get a 10 x 2 x 2 tensor A of G-matrices, i.e. 10 samples of GL+(2, R) or SL(2, R) matrices distributed according to the Haar measure. This works if the rotation matrices are sampled uniformly with respect to the Haar measure and the SPD matrices are sampled with respect to the GL(n, R) or SL(n, R) invariant measure on this manifold, as described in detail in Appendix B.6. In this case the individual factors of the product space are independent random variables and the joint distribution factorizes, which allows for this sampling scheme. We will try to make this more in clear in the next revision.
>
> > I don't fully understand why the proposed group decomposition technique is able to estimate the correlation integral well with only 10 samples while the work by MacDonald et al. need 100.
>
> Thank you for the question! We have tried to make this more clear in the revision. Section A.3 in the appendix has a more in-depth analysis of this approach, focusing on the theoretical presentation of their method (as we cannot now if further implementation issues exist which lead to their reduced sample efficiency).

---

> > ### Comment · Reviewer_yeU2 · 2023-11-22
> > **Thank you for author response**
> >
> > I thank the authors for their response and for addressing my concerns.
> >
> > I am assuming that the authors will add numerical results for equivariance error as a function of samples.

---

### Official Review · Reviewer_gQEm · 2023-11-02

**Soundness:** 4 excellent
**Presentation:** 2 fair
**Contribution:** 3 good
**Rating:** 8
**Confidence:** 3

**Summary:**

This paper proposes a new approach to efficiently generalize group equivariant convolutions for the  $\text{GL}^{+}(n, \mathbb{R})$ and $\text{SL}(n, \mathbb{R})$ groups. While the Haar measure for such non-compact groups is ill-defined, the approach proposes to work with the decomposition of these groups into factors that have well-behaved Haar measures. Factorizing the convolution integral enables the computation of group equivariant convolutions on these groups. By using results from random matrix theory to efficiently sample elements of these groups, the authors propose an efficient way to compute these integrals via Monte Carlo (MC) integration. This results in an order of magnitude improvement in the number of samples needed for a converged integral. The method is tested on two image classification tasks under affine and homographic transformations. In both benchmarks, the method is shown to outperform previous approaches.

**Strengths:**

- The paper is clearly written, and the contributions are made clear.
- The mathematical concepts are well-introduced, precise, and effectively utilized.
- The approach is novel and original, making elegant use of existing results in classical group theory and random matrix theory within the context of geometric deep learning.
- The two groups covered in the paper ($\text{GL}^{+}(n, \mathbb{R})$, $\text{SL}(n, \mathbb{R})$), are two important groups for image-related tasks.
- The sample efficiency gains might prove to be a key element for the future adoption of group equivariant convolutions in image-related tasks.
- The benchmark results are convincing, comparing favorably to strong baselines in the field.

**Weaknesses:**

- The analysis of the numerical stability of the method is largely missing. In particular, methods that rely on approximating integrals over groups will always be approximately equivariant. While increasing the number of samples could reduce this error below floating-point precision, such an analysis is not present in the paper. A curve illustrating the equivariance error as a function of the number of Monte Carlo samples would be pertinent in this context. Although sizeable equivariance errors might be manageable in the context of image classification tasks, this is not the case for all types of application tasks where group convolutions are relevant. Specifically, in physics simulations, equivariance errors are related to the non-conservation of physical quantities. The proposed approach should be contrasted with approaches that use linear operators to evaluate the group integral, such as [1] and [2], which are exactly equivariant (even for non-compact groups) but require large tensorial operations (for [1]) or ad hoc representation theory for each group (for [2]). I encourage the authors to briefly discuss these points in the text and add an analysis of numerical stability in the form of a plot to the appendix.

- In the paper, only the symmetrization to the canonical group action is explicitly mentioned. Extending the matrix elements to other finite-dimensional group representations should be straightforward, provided that the correct basis is determined. I am curious why this was not presented in full generality.

[1] "A Practical Method for Constructing Equivariant Multilayer Perceptrons for Arbitrary Matrix Groups," by Finzi et al.

[2] "A General Framework for Equivariant Neural Networks on Reductive Lie Groups," by Batatia et al.

**Questions:**

- See my point regarding numerical stability.

- This might be a standard result already, but I would like to have a formal view of the convergence of your approach. Do you think it would be possible to obtain an explicit bound on the equivariance error as a function of the number of Monte Carlo samples, the sizes of the factors in the group decomposition, and the size of the representation? To be more precise, I am looking for something akin to this bound: $|| \int_{H} \int_{K} f(\rho(h) \rho(k) \cdot x) d\mu_{H}(\rho(h)) d\mu_{K}(\rho(k)) - \frac{V}{N} \sum_{i} f(\rho(h_{i}) \rho(k_{i}) \cdot x), \rho(h_{i}) \sim \mu_{H}, \rho(k_{i}) \sim \mu_{K}|| \leq f(N, dim(\rho), dim(H), dim(K) )$
Using that, one could ask what kind of factorization gives the best bound.

---

> ### Author Response · Authors · 2023-11-22
>
> Thank you for the positive feedback! We have made an update to the manuscript focused on clarifying general details and plan to make another revision where the equivariance error analysis is presented as a factor of the number of samples, and related work is discussed more in-depth (addressing the other questions in your review which are not mentioned here).
>
> > In the paper, only the symmetrization to the canonical group action is explicitly mentioned. Extending the matrix elements to other finite-dimensional group representations should be straightforward, provided that the correct basis is determined. I am curious why this was not presented in full generality.
>
> Because of the mathematical denseness already existing in the paper we decided to limit its scope and leave possible applications of these tools in the context of finite-dimensional representations for these groups to future work.
>
> > This might be a standard result already, but I would like to have a formal view of the convergence of your approach. Do you think it would be possible to obtain an explicit bound on the equivariance error as a function of the number of Monte Carlo samples, the sizes of the factors in the group decomposition, and the size of the representation? To be more precise, I am looking for something akin to this bound: $|| \int_{H} \int_{K} f(\rho(h) \rho(k) \cdot x) d\mu_{H}(\rho(h)) d\mu_{K}(\rho(k)) - \frac{V}{N} \sum_{i} f(\rho(h_{i}) \rho(k_{i}) \cdot x), \rho(h_{i}) \sim \mu_{H}, \rho(k_{i}) \sim \mu_{K}|| \leq f(N, dim(\rho), dim(H), dim(K) )$ Using that, one could ask what kind of factorization gives the best bound.
>
> We will try to include a discussion on this aspect in the next revision. For certain groups [1] much more explicit bounds can be obtained, however we are not aware of a result of this form for the groups employed in the paper.
>
> [1] - https://en.wikipedia.org/wiki/Quantum_t-design#Unitary_t-Designs

---

### Meta-Review · Area_Chair_vQVn · 2023-12-05

**Metareview:**

The paper presents a method for constructing layers (approximately) equivariant to Lie groups whose exponential maps are not surjective, emphasizing $\mathrm{GL}^+(n, \mathbb{R})$ and $\mathrm{SL}(n, \mathbb{R})$ . A key idea used in the work concerns the decomposition of the Lie group under consideration into factors that have well-behaved Haar measure.

Several issues were discussed and largely resolved, primarily concerning the positioning of the contributions relative to related work, the use of Monte Carlo samples, and the writing, the mathematical density in particular. Overall, the paper received positive feedback with the reviewers appreciating the novelty of the approach and the mathematical rigor. Please make sure to incorporate the important feedback given by the knowledgeable reviewers in the revised version.

**Justification For Why Not Higher Score:**

Scope of applicability has not been made sufficiently convincing.

**Justification For Why Not Lower Score:**

Good to see an execution of the idea to form equivariant layers using decomposition of a Lie group.

---

### Decision · Program_Chairs · 2024-01-16

Accept (poster)